# The WNK-OXSR1 osmosensing pathway mediates intestinal regeneration via Hippo-YAP signaling

Heming Cao[1,3], Xiawei Huang [ID][1,3], Xiaobing Jiang[1,3], Jingrong Deng[1], Jiahui Wang [ID][1], Chengfang Wu[1], Minhuang Hu[1], Bei Zeng[1], Zhihao Hu[1], Huimin Pan[1], Yuxia Yang[1], Kewei Zheng[1], Rui Shen[1], Mingqing Zhang[2] & Bo Liu [ID][1✉]

## Abstract

**Animals activate regenerative processes to repair injuries and restore homeostasis following tissue damage. A central question in regeneration is how damage signals are sensed and translated into regenerative growth. Tissue injuries lead to the release of intracellular contents and bodily fluids and disturb the osmotic balance. However, the role of osmolarity in regeneration remains largely unexplored. Using *Drosophila* and mouse intestine, as well as samples from inflammatory bowel disease (IBD) patients, we identify a key role for the osmolarity-sensing WNK-OXSR1 kinase cascade in intestinal regeneration. Mechanistically, OXSR1 phosphorylates the RhoB GTPase at threonine 37 upon intestinal injury, thereby disrupting its interaction with ARHGAP17 and increasing the levels of GTP-bound RhoB. RhoB activation in turn leads to enhanced F-actin polymerization and YAP activation, thus promoting tissue regeneration. We further show that pharmacological inhibition of WNK or OXSR1 reduces the oncogenic potential of intestinal regeneration. These findings reveal osmolarity as a critical damage signal in regeneration and position WNK-OXSR1 as a potential therapeutic target for stimulating intestinal repair.**

Subject Categories Cancer; Cell Adhesion, Polarity & Cytoskeleton; Signal Transduction

## Introduction

Regeneration is the biological process by which lost cells are replenished, and tissue structure and function are restored following injury (Goldman and Poss, 2020; Poss and Tanaka, 2024; Schafer and Werner, 2008). Understanding the molecular mechanisms that regulate tissue regeneration is crucial for developing effective therapeutic strategies in regenerative medicine. A central question in the field is how tissue damage signals are sensed and leveraged to initiate regenerative growth. Decades of research have found that tissues rely on both transcription-dependent and transcription-independent signals to detect injuries and initiate regeneration (Cordeiro and Jacinto, 2013). Regeneration begins with transcription-independent damage signals, such as $Ca^{2+}$, hydrogen peroxide ($H_2O_2$), and ATP, that are rapidly released at the injury site and diffuse through the surrounding tissue microenvironment, ultimately triggering downstream gene expression and regenerative responses. One potentially important but underexplored transcription-independent damage signal is the change in osmolarity by injury. Disruption of cell membrane integrity and tissue barrier function during damage can cause leakage of intracellular contents and bodily fluids rich in ions and solutes into the extracellular space, potentially disturbing local osmotic balance. However, the role of osmolarity disturbances in regulating tissue regeneration remains largely unknown.

The WNK-SPAK/OXSR1 signaling cascade is a key pathway that mediates cell volume and ionic homeostasis in response to osmolarity change. Hyperosmolarity and cell shrinkage activate the with-no-lysine (WNK) kinase (Boyd-Shiwarski et al, 2022; Zagorska et al, 2007), which further phosphorylates and activates two downstream Ste20 kinases, Ste20-related proline/alanine-rich kinase (SPAK) and oxidative stress response kinase (OXSR1) (Anselmo et al, 2006; Moriguchi et al, 2005; Richardson et al, 2008; Vitari et al, 2005). Activated SPAK/OXSR1 in turn phosphorylates and activates ion-importing co-transporters, including $Na^+/Cl^-$ co-transporter (NCC) (Richardson et al, 2008) and $Na^+/K^+/2Cl^-$ co-transporters (NKCC1 and NKCC2) (Gimenez and Forbush, 2005; Moriguchi et al, 2005; Ponce-Coria et al, 2008) [collectively, N(K)CC], triggering ion influx. Meanwhile, SPAK/OXSR1 inactivates ion-extruding $K^+/Cl^-$ co-transporters (KCCs) (de Los Heros et al, 2014), blocking ion efflux. Echoing the pivotal roles of the WNK-SPAK/OXSR1-N(K)CC axis in ionic homeostasis, gene mutations in this axis are associated with various human diseases characterized by abnormal blood pressure and ion levels (Gordon and Hodsman, 1986; Simon et al, 1996a; Simon et al, 1996b; Wilson et al, 2001). In addition to their roles in adult physiology, mice homozygous for mutations in *Wnk1* or *Oxsr1* exhibit angiogenesis defects and die at the embryonic stage (Xie et al, 2009; Xie et al, 2013), highlighting their importance in embryonic development. It has been reported that SPAK and OXSR1 exhibit broad tissue expression (Gagnon and Delpire, 2012), including in cells that experience minimal osmotic stress. However, their physiological roles in these tissues remain largely underexplored.

[1]State Key Laboratory of Cellular Stress Biology, Innovation Center for Cell Signaling Network, School of Life Sciences, Xiamen University, Xiamen, China. [2]Department of Gastroenterology, The 909th Hospital, School of Medicine, Xiamen University, Zhangzhou, China. [3]These authors contributed equally: Heming Cao, Xiawei Huang, Xiaobing Jiang. ✉E-mail: bliu23@xmu.edu.cn

The evolutionarily conserved Hippo pathway is a key regulator of tissue growth by coordinating cell proliferation and cell death across diverse animal species (Davis and Tapon, 2019; Halder and Johnson, 2011; Harvey and Hariharan, 2012; Yu et al, 2015; Zheng and Pan, 2019). At the core of this signaling pathway is a kinase cascade in which the Hpo-Sav complex (MST1/2-SAV1 in mammals) phosphorylates and activates the Wts-Mats complex (LATS1/2-MOB1A/B in mammals). This activated complex subsequently phosphorylates the transcriptional coactivator Yki (YAP/TAZ in mammals). Phosphorylation of Yki at S168 (corresponding to YAP S127 and TAZ S89) promotes its interaction with cytoplasmic 14-3-3 proteins, resulting in the exclusion of Yki (YAP/TAZ) from the nucleus (Dong et al, 2007; Oh and Irvine, 2008; Zhao et al, 2007). Under normal conditions, Yki (YAP/TAZ) acts in the nucleus as a coactivator of the TEAD family transcription factor Sd (TEAD1/2/3/4 in mammals) to promote the expression of Hippo pathway target genes that regulate cell proliferation and survival. In addition, LATS-mediated YAP S381 or TAZ S311 phosphorylation results in the recruitment of β-TRCP, a subunit of the SCF ubiquitin E3 ligase, and subsequently the degradation of YAP/TAZ (Liu et al, 2010; Zhao et al, 2010). The Hippo signaling pathway is regulated by diverse upstream signals, including various G-protein-coupled receptor ligands and mechanical forces (Dupont et al, 2011; Wada et al, 2011; Yu et al, 2012; Zhao et al, 2012; Zheng and Pan, 2019). These upstream cues act through the Rho GTPase-ROCK (Rho-associated protein kinase)-MLC (non-muscle myosin II light chain) pathway, influencing the integrity and dynamics of the contractile actomyosin cytoskeleton. This modulation subsequently affects Yki/YAP/TAZ activity through Wts/LATS-dependent or independent mechanisms. Crucially, the Hippo pathway plays a pivotal role in tissue regeneration through its downstream effectors, YAP and TAZ (Driskill and Pan, 2023; Juan and Hong, 2016; Moya and Halder, 2019; Patel et al, 2017). In mouse intestinal epithelium, the levels of YAP were strongly induced after dextran sulfate sodium (DSS)-induced colitis and regeneration (Cai et al, 2010; Taniguchi et al, 2015). Furthermore, mice with intestinal epithelial cell-specific conditional knockout of *Yap* exhibited more severe loss of the crypt compartment and a higher lethality rate than wild-type mice upon DSS-induced intestinal injury (Cai et al, 2010). Despite these findings, how YAP is activated during the intestinal regeneration process remains incompletely understood. Discovering new regulators and understanding their mechanisms of action in YAP/TAZ regulation holds significant potential for advancing regenerative medicine.

In this study, we demonstrate that the WNK-OXSR1 signaling axis is a crucial mediator of tissue regeneration through its regulation of the Hippo signaling pathway. Genetic or pharmacological inhibition of the WNK-OXSR1 axis significantly impairs tissue regeneration in a DSS-induced intestinal regeneration model. Mechanistically, OXSR1 directly phosphorylates the RhoB small GTPase at Threonine 37 (T37), inhibiting its interaction with the negative regulator ARHGAP17. This promotes the active form of RhoB, leading to increased F-actin assembly, which subsequently inhibits YAP phosphorylation and enhances YAP activity. Moreover, targeting WNK or OXSR1 with chemical inhibitors markedly reduced the oncogenic potential associated with intestinal regeneration in the AOM/DSS-induced mouse model. Overall, our findings elucidate an OXSR1-RhoB-F-actin-Hippo-YAP signaling cascade that is essential for intestinal tissue regeneration. These results not only identify OXSR1 as a novel upstream regulator of Hippo signaling but also expand the physiological roles of the WNK-OXSR1 pathway beyond the maintenance of ionic homeostasis. Furthermore,

our study provides proof-of-principle evidence that manipulating the WNK-OXSR1 pathway may serve as a potential therapeutic strategy to improve tissue repair and mitigate the oncogenic risks during intestine regeneration.

## Results

### Wnk–Fray axis is essential for intestinal regeneration in *Drosophila*

To investigate the influence of osmolarity on tissue regeneration, we examined the WNK-SPAK/OXSR1 pathway, given its well-established role in osmolarity regulation. We began by assessing the necessity of this pathway in tissue regeneration using a *Drosophila* intestinal regeneration model, chosen for its ease of genetic manipulation and high conservation of cellular composition and regenerative mechanisms with mammals. Similar to the vertebrate gut, *Drosophila* gut epithelium is predominantly composed of absorptive enterocytes (EC), interspersed with hormone-producing enteroendocrine (EE) cells. Intestinal stem cells (ISC) are located along the basement membrane and are the only cell types capable of proliferation. In response to tissue damage such as those induced by feeding with DSS (Amcheslavsky et al, 2009), ISCs divide to produce both renewed ISCs and transient progenitor cells, including enteroblasts (EB) or enteroendocrine (EE) progenitor cells (EEP). These progenitors further differentiate into EC or EE cells to replenish the damaged gut epithelium (Biteau et al, 2011; Jiang and Edgar, 2012; Jiang et al, 2016; Lucchetta and Ohlstein, 2012; Micchelli and Perrimon, 2006; Ohlstein and Spradling, 2006). To investigate the role of the WNK-SPAK/OXSR1 axis in fly intestinal regeneration, we employed the ISC/EB-specific *escargot-Gal4*; *UAS-mCD8GFP* (*esg* > GFP) driver/reporter line (Micchelli and Perrimon, 2006) to genetically manipulate the expression of *frayed* (*fray*), the sole *Drosophila* homolog of OXSR1/SPAK. In this system, ISC/EB cells are simultaneously marked by GFP. We observed that flies with *fray* knockdown in ISC/EB developed normally, suggesting that Fray is dispensable for normal gut development and homeostasis. Consistent with this, genetic manipulation of Fray expression in the wing or eye did not affect organ size (Fig. EV1A,B), further supporting a minimal role for Fray in developmental organ growth. In contrast, following DSS treatment, flies with ISC/EB-specific *fray* knockdown showed significantly higher lethality, while flies with ISC/EB-specific Fray overexpression exhibited significantly lower lethality compared to control flies (Figs. 1A,B and EV1C,D). These results suggest that Fray is essential for gut regeneration following injury.

To further support this, we performed a Smurf assay to evaluate the integrity of the intestinal barrier (Rera et al, 2011). In this assay, adult flies were fed food containing Bromophenol Blue, which remains confined to the digestive tract when the gut is intact. If the intestinal epithelial barrier is compromised, the dye leaks into the abdominal cavity. Our results showed that *fray* knockdown flies exhibited increased blue dye leakage, while Fray overexpression flies showed reduced leakage into the abdominal cavity compared to control flies following DSS treatment (Figs. 1C and EV1E). These findings suggest compromised gut regeneration in the *fray* knock-down group and improved gut regeneration in the Fray overexpression group.

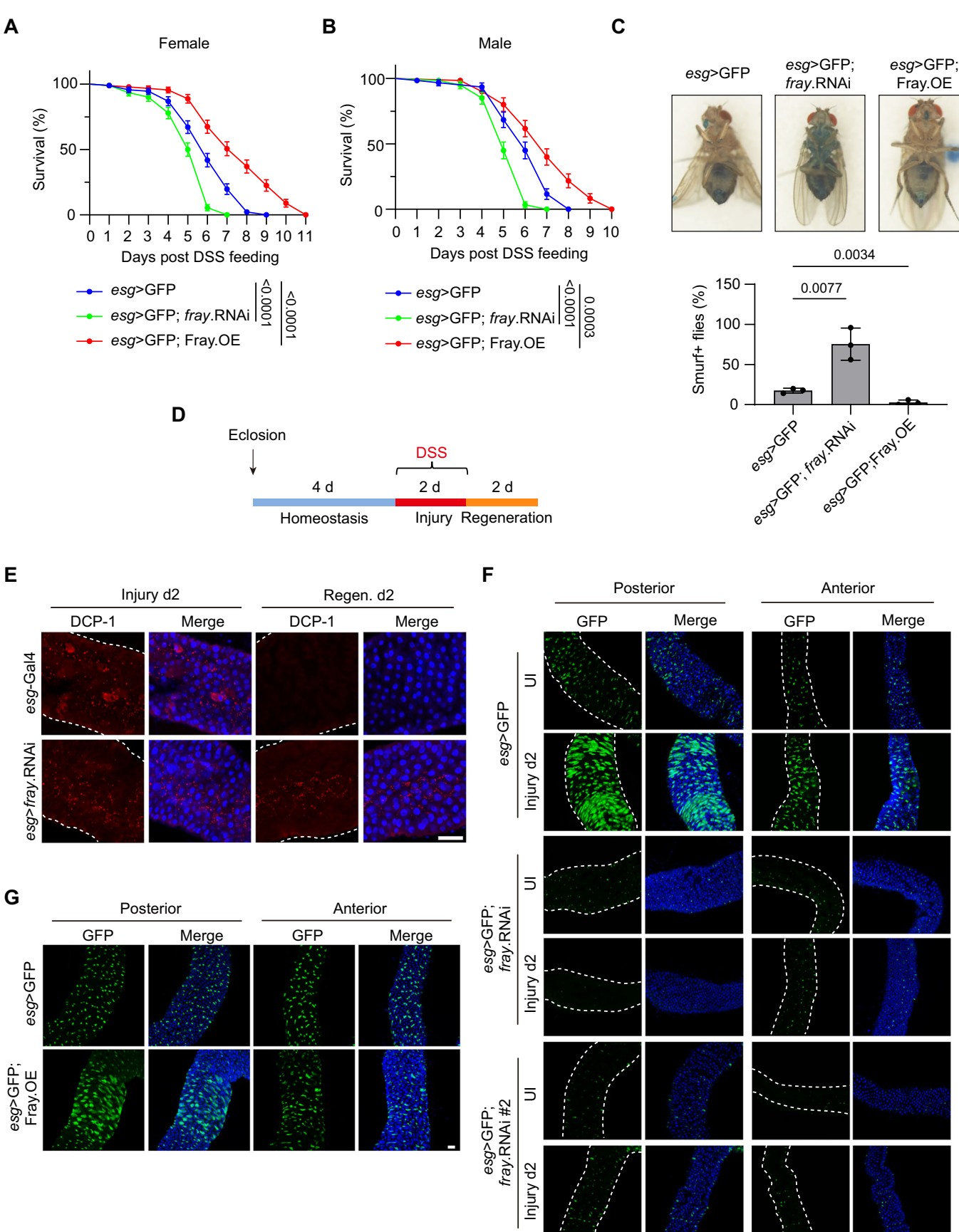

**Figure 1. Wnk–Fray axis regulates intestinal regeneration in *Drosophila*.**

(A, B) Virgin female (A) and male (B) flies with *fray* knockdown (BDSC #41587) or Fray overexpression (OE) were exposed to 5% DSS, and survival was monitored daily. Survival curves were analyzed using the log-rank (Mantel–Cox) test and are presented as mean ± s.e.m. For (A), n = 90 flies from three independent experiments. For (B), n = 60 flies from three independent experiments. (C) Smurf assays were performed on flies with the indicated genotypes. Flies were treated with 3% DSS for 5 days, followed by feeding with 0.5% Brilliant Blue in 5% sucrose for 12 h before harvest. Top: the ventral views of adult flies after 5-day DSS treatment. Bottom: quantification of Smurf-positive flies. Data were analyzed using two-tailed Student's t test and are presented as mean ± s.d., n = 3 independent experiments (50 flies per experiment). (D) Schematic representation of the DSS treatment regimen in *Drosophila*. (E) Midguts from flies with the indicated genotypes were stained with anti-DCP-1 antibody (red) and counterstained with DAPI (blue). The gut boundary is marked by white dashed curves. Scale bar: 50 µm. (F) ISC/EB cells in fly midguts with the indicated genotypes and treatments were visualized using esg > GFP reporter (green). Cell nuclei were counterstained with DAPI (blue). Note that knockdown of *fray* inhibited the DSS-induced increase in the number of ISC/EB cells. Data shown are representative images of the anterior (R2) and posterior (R4) regions of the midgut. The gut boundary is marked by white dashed curves. UI, uninjured. Scale bar: 50 µm. (G) Similar to (F) except for the fly genotypes. Note the increased number of ISC/EB cells in uninjured midguts with Fray overexpression. Scale bar: 50 µm. The microscopy images shown are representative of at least three independent experiments.

Next, we subjected flies to DSS treatment for 2 days (injury), followed by 2 days of standard food (regeneration; Fig. 1D). We observed that gut damage resulted in increased levels of effector *Drosophila* caspase-1 (DCP-1) in both control and esg>*fray* RNAi guts (Fig. 1E), indicating enhanced cell death. By day 2 of regeneration, DCP-1 expression was barely detectable in control guts, while it remained prominent in esg>*fray* RNAi guts (Fig. 1E), suggesting impaired gut repair in the latter. Enhanced ISC division is a key feature of gut regeneration. Consistent with this, DSS treatment led to a dramatic increase in the number of ISCs/EBs, as visualized by the esg > GFP reporter in control guts (Fig. 1F). In contrast, this increase was barely detectable in guts from two independent esg>*fray* RNAi lines targeting non-overlapping regions of the *fray* gene (Fig. 1F). Similarly, the increase in ISC/EB numbers following DSS treatment was largely inhibited by Rafoxanide (Rafo), a chemical inhibitor of SPAK/OXSR1 (AlAmri et al, 2017), in wild-type flies (Fig. EV1F). On the other hand, overexpression of Fray promoted an increase in ISC/EB numbers even in the absence of DSS treatment (Fig. 1G). These results collectively suggest that Fray is not only essential but also sufficient for ISC division and gut regeneration.

Having established an important role for Fray in fly intestinal regeneration, we next investigated whether this process also requires its upstream kinase, Wnk. We found that DSS-induced increases in ISC/EB numbers were largely suppressed by WNK463, a chemical inhibitor of WNK (Yamada et al, 2016), in wild-type guts (Fig. EV1F). Consistently, by day 2 of regeneration, DCP-1 levels in WNK463-treated guts were dramatically higher compared to untreated controls (Fig. EV1G). Together, these findings suggest that the Wnk–Fray pathway is essential for ISC proliferation and intestinal regeneration in *Drosophila*.

## WNK–OXSR1 axis is a conserved regulator of cell proliferation and intestinal regeneration in mammals

Having established a crucial role for the Wnk–Fray axis in cell proliferation and tissue regeneration in *Drosophila*, we next investigated whether this function is conserved in mammals. To this end, we first examined the roles of mammalian homologs of Fray, OXSR1, and SPAK in cell proliferation and cell death using cultured mammalian cell lines. Interestingly, via standard assays such as clonogenic assay, wound-healing assay, and cleaved caspase-3 assay, we observed that only *Oxsr1*-null HEK293T cells displayed a significant reduction in colony formation (Appendix Fig. S1A,B), wound healing (Appendix Fig. S1C,D) and an increase in cleaved caspase-3 levels (Appendix

Fig. S1E). Conversely, HEK293T cells stably overexpressing OXSR1 exhibited increased colony numbers (Appendix Fig. S1A,B), enhanced wound healing (Appendix Fig. S1F,G), and reduced cleaved caspase-3 levels compared to wild-type cells (Appendix Fig. S1H). Notably, *Spak*-null or SPAK-overexpressing cells did not exhibit similar deficits (Appendix Fig. S1A–H). These results prompted us to further explore the role of OXSR1 in mammalian tissue regeneration in vivo.

For this purpose, we utilized a well-established DSS-induced mouse model of colitis and regeneration (Okayasu et al, 1990). We crossed a conditional knockout allele of *Oxsr1* with *Villin-Cre-ERT2* (*VilCre^ERT2*) (el Marjou et al, 2004; Madison et al, 2002), which resulted in *Oxsr1* depletion in the epithelium of the small intestine and colon after feeding mice with Tamoxifen (TAM) (hereafter referred to as *Oxsr1* cKO mice) (Fig. 2A,B). *Oxsr1* cKO mice exhibited no overt intestinal abnormalities under steady-state conditions (Fig. EV2A), suggesting that OXSR1 does not play an essential role in intestinal developmental growth. However, following DSS exposure, *Oxsr1* cKO mice exhibited a dramatically increased mortality rate and shortened colon (a biological marker of colonic inflammation) compared to the control littermates (Fig. 2C,D). Histological analysis demonstrated severe damage with substantial crypt loss (Fig. 2E), reduced cell proliferation (Fig. 2F), and a pronounced depletion of colonic stem cells (Fig. 2G) in the *Oxsr1* cKO mice compared to the control littermates. Similarly, administration of the OXSR1 inhibitor, Rafo, resulted in regeneration and proliferation defects akin to those observed in *Oxsr1* cKO mice (Fig. EV2B–E). These results suggest that OXSR1 is essential for DSS-induced tissue regeneration.

To further support this conclusion, we examined the levels of active OXSR1 in the process of intestinal regeneration via immunohistochemistry, immunofluorescence, and immunoblotting. The results consistently showed significantly elevated OXSR1 phosphorylation levels during the early stages of regeneration (Regen. day 4), which returned to baseline levels by the later stages (Regen. day 9) (Fig. 2H–J). Importantly, this temporal pattern paralleled the restoration of the intestinal stem cells, as indicated by Lrig1 immunostaining (Fig. 2K) and crypt *Lgr5* mRNA levels (Fig. EV2F). Notably, colonic osmolality was markedly elevated in the early regeneration phase and returned to homeostatic levels at later stages (Fig. EV2G). This temporal pattern closely mirrors OXSR1 phosphorylation dynamics, supporting a potential functional link between gut osmolality and OXSR1 activation during regeneration. Consistent with the findings from the DSS-induced ulcerative colitis mouse model, biopsy specimens from patients with inflammatory bowel disease (IBD), characterized by intestinal

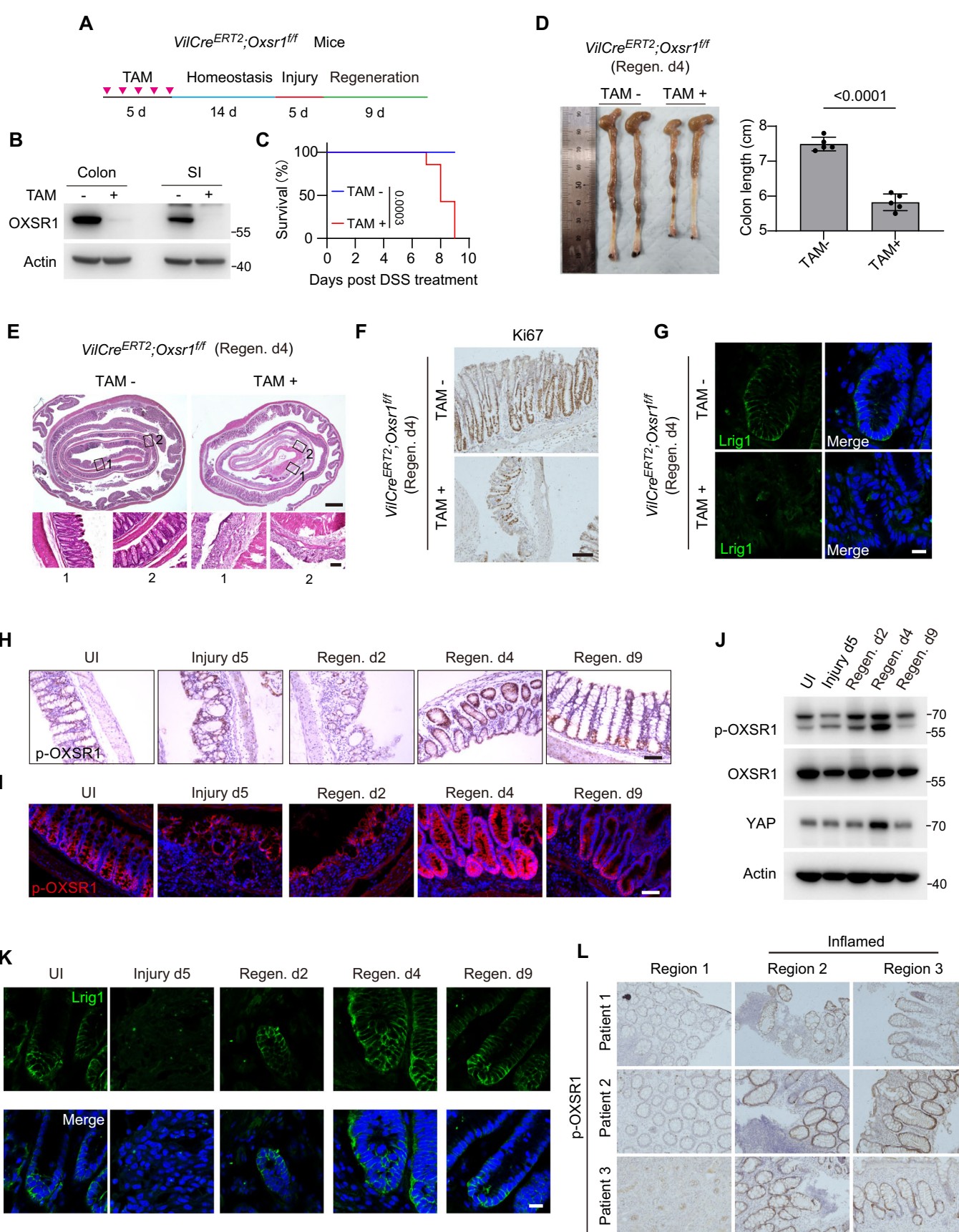

**Figure 2.  WNK–OSXR1 axis is a conserved regulator of cell proliferation and intestinal regeneration in mammals.**

(A) Schematic representation of the DSS treatment regimen in mice. (B) OXSR1 protein levels in the crypts of the small intestine (SI) and colon from *VilCre^ERT2; Oxsr1^f/f* mice, with or without Tamoxifen (TAM) treatment, were assessed via western blot. (C) *VilCre^ERT2; Oxsr1^f/f* mice, with or without TAM treatment, were administered 2.5% DSS for 5 days, and their survival was subsequently monitored daily. Survival curves were analyzed using log-rank (Mantel–Cox) test. *n* = 7 mice. Note the increased mortality of *Oxsr1* conditional knockout (cKO) mice after DSS treatment. (D–G) Colons from TAM-untreated or treated *VilCre^ERT2; Oxsr1^f/f* mice were collected on day 4 of the regeneration phase following DSS treatment and subjected to further analyses. Data shown are representative results of the colon gross morphology (D, left panel), quantification of colon length (D, right panel), hematoxylin and eosin (H&E) staining of colon Swiss roll (E), immunohistochemical staining for Ki67 (F), and immunostaining for Lrig1 (G) in colonic crypts. For (D), data were analyzed using two-tailed Student's *t* test and are presented as mean ± s.d. (*n* = 5 colons). Scale bars in (E): 500 μm (upper panel) and 100 μm (lower panel). Scale bar in (F): 100 μm. Scar bar in (G): 20 μm. (H–J) Phosphorylation levels of OXSR1 in the colonic crypts during the indicated stages of regeneration were assessed by immunohistochemistry (H), immunofluorescence (I), and immunoblotting (J). Note the elevated phosphorylation of OXSR1 at the early regeneration stage (Regen. d4), which returned to baseline levels at the late regeneration stage (Regen. d9). UI, uninjured. Scale bars in (H, I): 100 μm. (K) Dynamic changes in the Lrig1⁺ stem-cell population within colonic crypts at the indicated stages of injury and regeneration were assessed by immunostaining. Note that Lrig1 signals were reduced during injury and progressively reappeared during regeneration. Scale bar: 20 μm. (L) Representative immunohistochemistry images showing OXSR1 phosphorylation levels in colonic tissue from human IBD patients. Note the increased OXSR1 phosphorylation in the inflamed regions. Scale bar: 100 μm. The gel and microscopy images shown are representative of at least two independent experiments.

inflammation and injury, also showed increased OXSR1 phosphorylation levels in the inflamed regions (Fig. 2L).

To further substantiate the importance of OXSR1 activation in intestinal regeneration, we performed genetic rescue experiments by re-expressing OXSR1 variants in *Oxsr1* cKO mice using an adeno-associated virus (AAV) vector. The results showed that reintroduction of the constitutively active OXSR1-T185E mutant (Xie et al, 2013) effectively restored the regeneration defects observed in *Oxsr1* cKO mice, whereas reconstitution of the kinase-dead mutant OXSR1-T185A (Moriguchi et al, 2005; Rafiqi et al, 2010; Vitari et al, 2005) failed to rescue this defect (Fig. EV2H–J).

Building on these findings, we further investigated whether WNK, the upstream kinase of OXSR1, plays a role in this process. Similar to Rafo treatment, the administration of the WNK inhibitor, WNK463, significantly exacerbated the tissue damage induced by DSS, as evidenced by shortened colon length, increased crypt loss, and a reduction in proliferating cells compared to the vehicle control (Fig. EV2B–E). Together, these results suggest that the WNK-OXSR1 axis is essential for intestinal regeneration in mammals.

## OXSR1/Fray promotes cell proliferation and tissue regeneration through YAP/Yki

To elucidate the underlying mechanisms by which OXSR1 regulates tissue regeneration, we performed RNA-Seq to compare the transcriptomes of wild-type and *Oxsr1*-null cells. Notably, KEGG pathway analysis of differentially expressed genes revealed significant enrichment of the Hippo signaling pathway, which is critically involved in cell proliferation and tissue regeneration (Fig. 3A). Consistently, GO biological process (GO-BP) analysis showed that genes downregulated upon *Oxsr1* depletion were associated with DNA damage response, cell division, and regulation of apoptosis, all of which represent canonical processes modulated by Hippo signaling, further suggesting that OXSR1 functions as a potential upstream modulator of this pathway (Fig. 3B). Further analysis showed that multiple YAP target genes were downregulated in *Oxsr1*-null cells, and this finding was independently validated by RT-qPCR (Fig. 3C–E). Consistent with the decrease in YAP target genes following *Oxsr1* loss, we observed a significant increase in YAP phosphorylation at canonical sites S127 and S381, accompanied by a mobility shift of YAP in *Oxsr1* knockout cells (Fig. 3F). Moreover, the YAP protein levels were reduced in *Oxsr1* knockout cells (Fig. 3F), consistent with the notion that YAP S381

phosphorylation promotes its degradation (Zhao et al, 2010). In line with its increased phosphorylation levels, YAP demonstrated a more pronounced cytoplasmic localization in *Oxsr1* knockout cells than in wild-type cells (Fig. 3G). On the other hand, we observed that overexpression of OXSR1 resulted in reduced levels of YAP phosphorylation (Fig. EV3A). These results indicate that the osmolarity-sensing kinase OXSR1 is a positive regulator of YAP and suggest that hyperosmotic stress may promote YAP activity.

Previous studies using cultured cells have reported that hyperosmotic stress activates Hippo signaling, leading to enhanced YAP phosphorylation (Hong et al, 2020; Hong et al, 2017; Wang et al, 2022). Paradoxically, despite this enhancement, YAP still showed increased nuclear accumulation under hyperosmotic conditions (Hong et al, 2017). We note that these studies applied relatively high osmolarity levels (≥0.2 M sorbitol), whereas the osmolarity increase observed in the regenerating gut (Fig. EV2G) was milder (comparable to ~0.1 M sorbitol in cultured cells). This discrepancy prompted us to investigate whether hyperosmotic stress exerts biphasic effects on YAP phosphorylation depending on its intensity, with mild hyperosmolarity reducing and strong hyperosmolarity increasing YAP phosphorylation. Indeed, we found that mild hyperosmotic stress, mimicking conditions observed during intestinal regeneration, resulted in decreased YAP phosphorylation (Fig. EV3B). Because TAZ is a YAP paralog that often responds similarly to upstream cues, we also assessed its behavior and found that mild hyperosmolarity decreased TAZ phosphorylation as well (Fig. EV3B). Consistent with enhanced YAP/TAZ activity, expression of their target genes was increased under these conditions (Fig. EV3C). In contrast, more intense hyperosmotic stress led to elevated YAP phosphorylation (Fig. EV3D). Importantly, the decrease in YAP/TAZ phosphorylation and the upregulation of their target gene expression under mild hyperosmotic stress conditions were abolished in *Oxsr1*-null cells (Fig. EV3B,C), underscoring a critical role for OXSR1 in mediating YAP/TAZ response to a modest osmolarity increase. Notably, Nemo-like kinase (NLK) has been reported to regulate YAP under conditions of high osmolarity ( ≥0.2 M sorbitol) or high cell density (Hong et al, 2017; Moon et al, 2017). To explore whether NLK is also involved in YAP regulation under mild osmotic stress conditions, we examined YAP phosphorylation (p-S127/S381) in cells with stable *Nlk* knockdown. As shown in Fig. EV3E,F, the reduction in YAP phosphorylation caused by 0.1 M sorbitol treatment or OXSR1 overexpression was unaffected

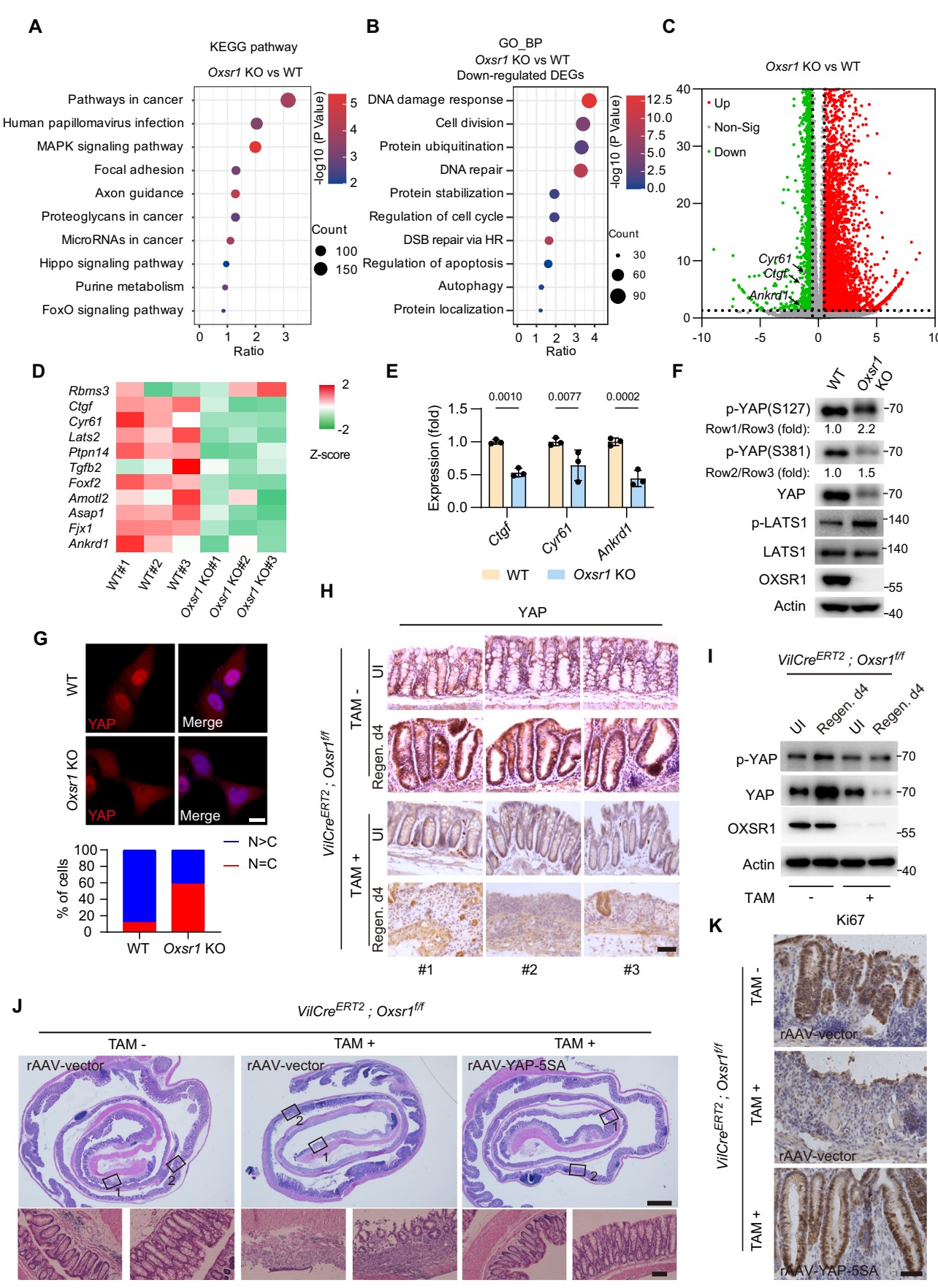

**Figure 3.  OXSR1 promotes cell proliferation and tissue regeneration through YAP.**

(A–D) RNA sequencing (RNA-seq) was performed on wild-type (WT) and *Oxsr1* knockout (KO) HEK293T cells. (A) Differentially expressed genes (DEGs) were defined as those with |log$_2$ fold change| >0.5 and *P* < 0.05, and were subjected to KEGG pathway enrichment analysis using the modified Fisher's exact test. (B) GO-BP analysis (modified Fisher's exact test) of the downregulated DEGs in *Oxsr1* knockout cell. (C) The volcano plot displays upregulated (red) and downregulated (green) DEGs, with YAP target genes (*Ctgf*, *Cyr61*, and *Ankrd1*) highlighted in purple. (D) Heatmap showing the expression of selected YAP target genes across biological replicates of WT and *Oxsr1* KO cells. (E) RT-qPCR analysis of *Ctgf*, *Cyr61*, and *Ankrd1* mRNA levels in WT and *Oxsr1* KO HEK293T cells. Note that knockout of *Oxsr1* significantly reduced expression of these YAP target genes. Data were analyzed using two-tailed Student's *t* test and are presented as mean ± s.d., *n* = 3 independent experiments. (F) Western blot analysis of WT and *Oxsr1* KO HEK293T cells. Phosphorylation of YAP and LATS1 was detected using phospho-specific antibodies against YAP (S127 and S381) and LATS1 (T1077). The p-YAP over total YAP levels were quantified and indicated below the blots. Note that *Oxsr1* knockout led to elevated phosphorylation levels of both YAP and LATS1. (G) Immunostaining of endogenous YAP in WT and *Oxsr1* KO HeLa cells. Quantification of YAP subcellular localization is shown in the bottom panel. Note that *Oxsr1* knockout reduced the nuclear localization of YAP. Scale bar: 10 μm. (H, I) Colon sections from *Oxsr1* cKO (TAM +) and control (TAM −) mice were harvested on day 4 of the regeneration phase following DSS treatment, and analyzed by immunohistochemistry (H) and western blot (I) to assess YAP expression. Note that the elevated YAP levels observed in regenerating crypts of control mice were markedly diminished in *Oxsr1* cKO mice. UI, uninjured. Scale bar: 100 μm. (J, K) Recombinant adeno-associated virus carrying a constitutively active form of YAP (rAAV-YAP-5SA) was delivered to *Oxsr1* cKO and control mice. Colon sections were subjected to H&E (J) and Ki67 (K) staining. Note that overexpression of YAP-5SA restored crypt regeneration and cell proliferation impaired by *Oxsr1* deletion. Scale bars in (J): 500 μm (upper panel) and 100 μm (lower panel). Scale bars in (K): 50 μm. Data shown are representative of at least two independent experiments.

by *Nlk* knockdown, suggesting that NLK is dispensable in this context.

Next, we further investigated whether YAP is involved in OXSR1-mediated cell proliferation. Strikingly, the enhanced cell proliferation and viability led by OXSR1 overexpression were completely abrogated by *Yap* knockout (Fig. EV3G,H). Conversely, the increased cleavage of caspase-3 observed upon *Oxsr1* deletion was rescued by the overexpression of a constitutively active form of YAP with all five phosphorylation sites mutated (YAP-5SA) (Zhao et al, 2010) (Fig. EV3I). Collectively, these results suggest that OXSR1 promotes cell proliferation and inhibits apoptosis through YAP.

In addition to Hippo signaling, our RNA-seq analysis also revealed enrichment of the MAPK signaling pathway, which has been differentially implicated in intestinal regeneration (Assi et al, 2006; Hollenbach et al, 2004; Hommes et al, 2002; Jiang et al, 2011; Wu et al, 2021; Zhang et al, 2020). To explore whether MAPK signaling is also involved in OXSR1-mediated intestinal regeneration, we examined MAPK signaling activation during regeneration in *Oxsr1* cKO mice and their control littermates. We found that MAPK signaling was activated in control colons at regeneration day 4, as evidenced by increased phosphorylation of p38, JNK, and ERK (Fig. EV3J). In contrast, this activation was largely abolished in *Oxsr1* cKO colons (Fig. EV3J), suggesting that the MAPK pathway may also contribute to OXSR1-mediated intestinal regeneration. Besides MAPK signaling, we investigated the possible involvement of Wnt signaling in OXSR1-mediated regeneration, given its essential role in intestinal regeneration (Clevers et al, 2014; Gehart and Clevers, 2019; Nusse and Clevers, 2017). RT-qPCR analysis showed that the expression of multiple canonical Wnt target genes was induced in control colons at regeneration day 4, whereas this induction was markedly impaired in *Oxsr1* cKO colons (Fig. EV3K). Together, these results suggest that OXSR1 may coordinate multiple regeneration-associated signaling pathways to promote intestinal regeneration.

Having identified YAP as an essential downstream effector of OXSR1-mediated cell proliferation in cell lines, we next explored its role in OXSR1-driven tissue regeneration in vivo. As previously reported (Cai et al, 2010), in the regenerating crypts of wild-type mice following DSS treatment, YAP protein levels were dramatically increased (Fig. 3H,I). Notably, this increase in YAP levels was largely absent in *Oxsr1* cKO mice (Fig. 3H,I), indicating its dependence on OXSR1. To directly test whether YAP activation

could rescue the regenerative defects caused by *Oxsr1* loss, we delivered a recombinant adeno-associated virus encoding a constitutively active YAP mutant (rAAV-YAP-5SA) via intraperitoneal injection into *Oxsr1* cKO mice and confirmed efficient intestinal transduction by immunostaining (Fig. EV4A). Remarkably, enforced expression of YAP-5SA significantly rescued regeneration defects (Fig. 3J,K), attenuated body weight loss (Fig. EV4B), and reduced lethality (Fig. EV4C) in DSS-treated *Oxsr1* cKO mice. To further enhance tissue specificity and more closely mimic physiological YAP activation, we obtained an AAV vector expressing a mir30-based short-hairpin RNA targeting *Sav1* under the control of the *Vil1* promoter (rAAV-*Vil1*-mir30-shSav1). Consistent with YAP-5SA overexpression, *Sav1* knockdown markedly alleviated crypt loss, restored Ki67-positive proliferative activity in *Oxsr1* cKO colons, and significantly improved survival during DSS-induced injury (Fig. EV4D–F).

In *Drosophila*, overexpression of a constitutively active form of Yorkie (Yki-3SA) (Oh and Irvine, 2009) markedly restored the number of ISC/EB cells following DSS treatment in *fray* knockdown flies (Fig. EV4G). Together, these results suggest that OXSR1/Fray mediates tissue regeneration through YAP/Yki in vivo.

## OXSR1 functions upstream of MST1/2 and MAP4Ks to regulate YAP

Having identified OXSR1 as a positive regulator of YAP, we proceeded to explore the underlying mechanisms. To this end, we first examined whether the traditional downstream ion transporter targets of OXSR1, NKCC, and KCC are involved in this process. Interestingly, we found that neither the NKCC inhibitors Bumetanide and Chlorothiazide nor the KCC activator CLP-290 affected the reduction in YAP phosphorylation observed with OXSR1 overexpression (Fig. 4A), suggesting that OXSR1 regulates YAP phosphorylation independently of its role in ion transport. Since LATS1/2 are the key kinases that mediate YAP S127 and S381 phosphorylation, we examined whether OXSR1 regulates YAP phosphorylation through LATS1/2. Consistent with the increased YAP S127/S381 phosphorylation, we observed enhanced LATS1 phosphorylation in *Oxsr1* knockout cells as well (Fig. 3F). More importantly, *Oxsr1; Lats1/2* triple knockout (TKO) cells, similar to *Lats1/2* double knockout (DKO) cells, exhibited abolished YAP phosphorylation (Fig. 4B). Consistently, the *Oxsr1; Lats1/2* TKO

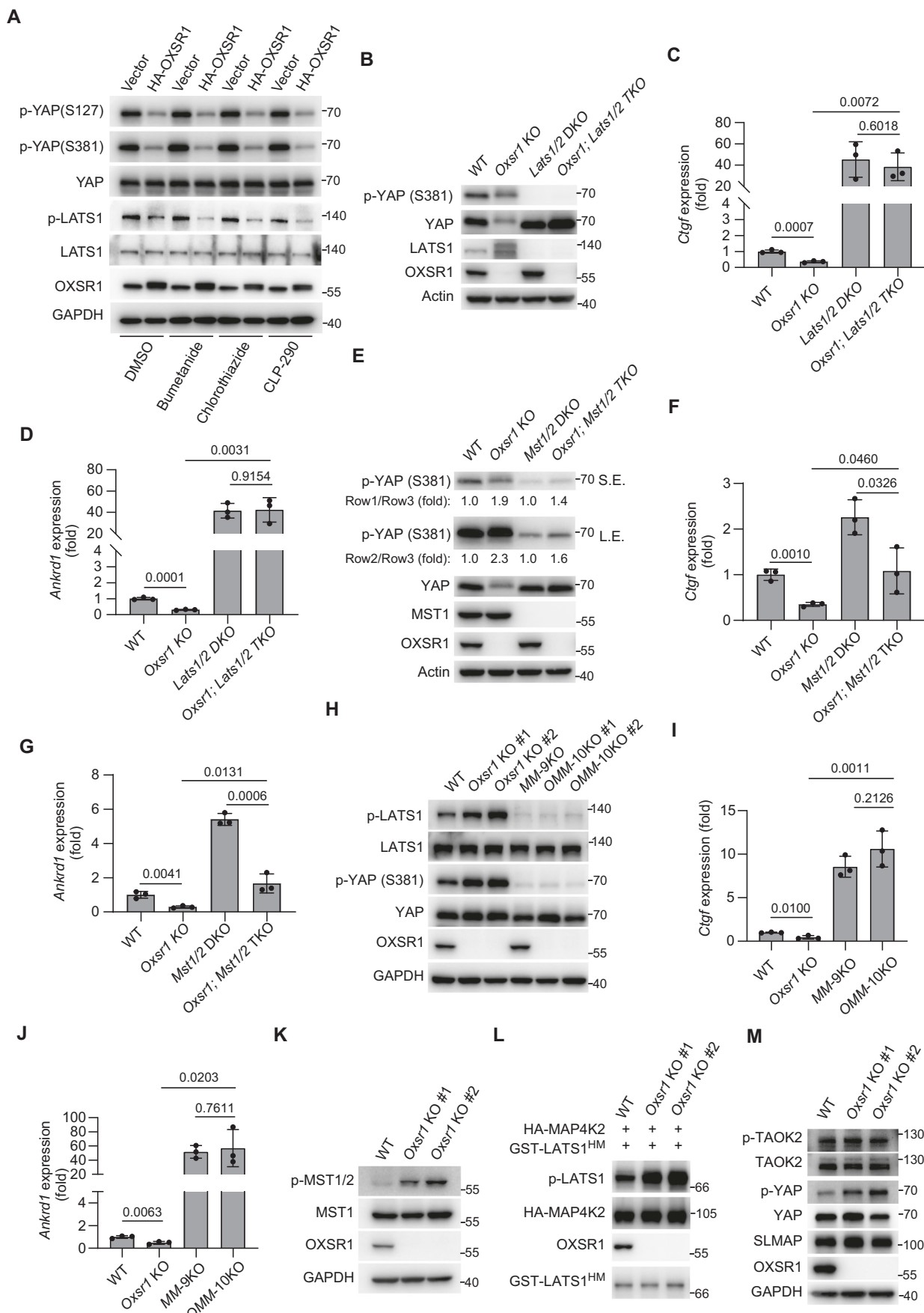

◄ **Figure 4.  OXSR1 functions upstream of MSTs and MAP4Ks to regulate YAP.**

(**A**) HEK293T cells were treated with the NKCC inhibitors Bumetanide (10 μM) and Chlorothiazide (10 μM), or the KCC activator CLP-290 (20 μM) for 2 h. Phosphorylation levels of LATS1 and YAP were analyzed by western blot. Note that OXSR1-mediated suppression of LATS1 and YAP phosphorylation is independent of NKCC, NCC, or KCC transporter activity. (**B**) Western blot was performed in the indicated HEK293T cell lines. Note that the *Oxsr1; Lats1/2* triple knockout (TKO) cells, like *Lats1/2* double knockout (DKO) cells, showed abolished YAP phosphorylation. (**C, D**) RT-qPCR was performed in HEK293T cells with the indicated genotypes. Note that the *Oxsr1; Lats1/2* TKO cells, like *Lats1/2* DKO cells, showed comparably increased expression of YAP target genes such as *Ctgf* (**C**) or *Ankrd1* (**D**). Data were analyzed using two-tailed Student's *t* test and are presented as mean ± s.d., $n = 3$ independent experiments. (**E**) Similar to (**B**) except for the genotypes of the cells. Note that *Oxsr1; Mst1/2* TKO cells still showed higher YAP phosphorylation levels compared to *Mst1/2* DKO cells. The p-YAP over total YAP levels were quantified and indicated below the blots. S.E., short exposure, L.E., long exposure. (**F, G**) Similar to (**C, D**) except for the genotypes of the cells. Note that the upregulation of YAP target genes induced by *Mst1/2* double knockout was still suppressed by *Oxsr1* knockout. Data were analyzed using two-tailed Student's *t* test and are presented as mean ± s.d., $n = 3$ independent experiments. (**H**) Western blot was performed in the indicated HEK293A cell lines. Note that simultaneous loss of *Oxsr1*, *Mst1/2*, and *MAP4K1/2/3/4/5/6/7* (*OMM*-10KO) led to markedly reduced phosphorylation of LATS and YAP, to a level similar to that of *Mst1/2; MAP4K1/2/3/4/5/6/7* knockout (*MM*-9KO) cells. (**I, J**) RT-qPCR was performed in HEK293A cells with the indicated genotypes. Note that *OMM*-10KO cells, like *MM*-9KO cells, showed comparably increased expression of YAP target genes *Ctgf* (**I**) or *Ankrd1* (**J**). Data were analyzed using two-tailed Student's *t* test and are presented as mean ± s.d., $n = 3$ independent experiments. (**K**) Phosphorylation of MST1/2 was analyzed by western blot in wild-type (WT) and *Oxsr1* knockout (KO) HeLa cells. *Oxsr1* KO cells generated from two independent sgRNAs were included. Note the increased phosphorylation levels of MST1/2 in *Oxsr1* KO cells. (**L**) In vitro kinase assay was performed using HA-MAP4K2 immunoprecipitated from wild-type (WT) or *Oxsr1* knockout (KO) HEK293A cells, and a bacterially purified GST-tagged LATS1 fragment containing its hydrophobic motif (GST-LATS1$^{HM}$). Phosphorylation of LATS1$^{HM}$ was assessed by western blot using a phospho-specific antibody against LATS1 at Threonine 1079 (p-LATS1 T1079). Note that HA-MAP4K2 isolated from *Oxsr1* KO cells induced markedly higher phosphorylation of LATS1$^{HM}$ compared to that from WT cells. (**M**) Phosphorylation of TAOK2 was analyzed by western blot in wild-type (WT) and *Oxsr1* knockout (KO) HeLa cells. *Oxsr1* KO cells generated from two independent sgRNAs were included. Note that knockout of *Oxsr1* did not affect TAOK2 phosphorylation. Gel images shown are representative of at least two independent experiments.

cells, like *Lats1/2* DKO cells, showed comparably increased expression of YAP target genes such as *Ctgf* or *Ankrd1* (Fig. 4C,D). These data collectively suggest that OXSR1 functions upstream of LATS1/2 to inhibit YAP phosphorylation.

Next, we further analyzed the dependence of OXSR1-induced YAP phosphorylation on kinases upstream of LATS1/2, including MST1/2 and the MAP4K subfamily of kinases. The MAP4K subfamily of kinases, including MAP4K1/2/3/4/5/6/7, function redundantly with MST1/2 to directly phosphorylate LATS1/2 in response to cues such as serum starvation, contact inhibition, and F-actin disruption (Li et al, 2014; Meng et al, 2015; Plouffe et al, 2016; Zheng et al, 2015). Interestingly, overall YAP phosphorylation was strongly reduced in *Oxsr1; Mst1/2* TKO cells, indicating that the MST1/2 are required for the elevated YAP phosphorylation in *Oxsr1* KO cells. However, *Oxsr1; Mst1/2* TKO cells still showed increased YAP phosphorylation compared to *Mst1/2* DKO cells (Fig. 4E), suggesting that additional kinases, most likely MAP4Ks, function redundantly with MST1/2 to promote YAP phosphorylation in response to *Oxsr1* KO. Consistent with this view, the upregulation of YAP target gene *Ctgf* and *Ankrd1* induced by *Mst1/2* double knockout was still suppressed by *Oxsr1* knockout (Fig. 4F,G), presumably due to these additional kinases. In contrast, as shown in Fig. 4H, a cell line with simultaneous loss of *Oxsr1*, *Mst1/2*, and *MAP4K1/2/3/4/5/6/7* (*OMM*-10KO) showed markedly reduced phosphorylation of LATS and YAP, to a level similar to that of *Mst1/2; MAP4K1/2/3/4/5/6/7* knockout (*MM*-9KO) cells. Consistently, the *OMM*-10KO cells, like *MM*-9KO cells, showed comparably increased expression of YAP target genes *Ctgf* or *Ankrd1* (Fig. 4I,J). Furthermore, we found that MST1/2 phosphorylation was increased (Fig. 4K), and MAP4K kinase (e.g., MAP4K2) activity was enhanced (Fig. 4L), in *Oxsr1*-null cells compared with wild-type cells, suggesting that OXSR1 inhibits the intrinsic kinase activity of these kinases. In contrast, phosphorylation of the upstream kinase TAOK2 remained unchanged in *Oxsr1*-knockout cells (Fig. 4M). Altogether, these data suggest that OXSR1 acts as an upstream inhibitor of MST1/2 and MAP4Ks, independently of TAOK kinases, thereby suppressing Hippo signaling.

## OXSR1 suppresses the Hippo signaling via promoting F-actin assembly

A previous study has suggested a potential role for the OXSR1 paralog SPAK kinase in regulating F-actin assembly (Tsutsumi et al, 2000). Moreover, given that F-actin has been shown to inhibit the Hippo pathway through both MST1/2 and MAP4Ks (Meng et al, 2018; Yu et al, 2012; Zheng et al, 2015), we therefore tested the possibility that OXSR1 regulates the Hippo signaling through promoting F-actin assembly. We first investigated the effects of OXSR1 on F-actin assembly by performing immunostaining with FITC-conjugated phalloidin, a highly selective bicyclic peptide that binds to F-actin. The results revealed that *Oxsr1* knockout markedly reduced the levels of F-actin stress fibers in HeLa cells (Fig. 5A) or SW480 colorectal epithelial cells (Fig. 5B), suggesting a broad role of OXSR1 in regulating F-actin levels. This conclusion was further supported by an alternative method (Qiao et al, 2017) in which F-actin in cultured cells or primary colonic crypts was collected using ultracentrifuge and analyzed via western blot (Fig. 5C,D). On the other hand, cells with stable OXSR1 overexpression displayed enhanced F-actin stress fibers (Fig. 5E,F). Together, these data suggest that OXSR1 is both necessary and sufficient for F-actin assembly. Notably, reintroducing the constitutively active OXSR1-T185E mutant, but not the kinase-dead OXSR1-T185A mutant, rescued the diminished F-actin levels in *Oxsr1*-null cells, underscoring the critical role of OXSR1 kinase activity in this process (Fig. 5G).

To further demonstrate that OXSR1 suppresses the Hippo pathway by promoting F-actin formation, we treated *Oxsr1*-null cells with Jasplakinolide (JAS), a drug that enhances F-actin polymerization (Bubb et al, 1994). We then assessed the phosphorylation levels of YAP, along with the expression of YAP target genes. The results indicated that the enhanced phosphorylation of YAP in *Oxsr1*-null cells was fully blunted by JAS treatment (Fig. 5H). Consistently, the reduction in expression of YAP target genes such as *Ctgf* and *Cyr61* in *Oxsr1*-null cells were abolished by JAS treatment (Fig. 5I). On the other hand, the reduced phosphorylation of LATS and YAP (Fig. 5J), as well as enhanced

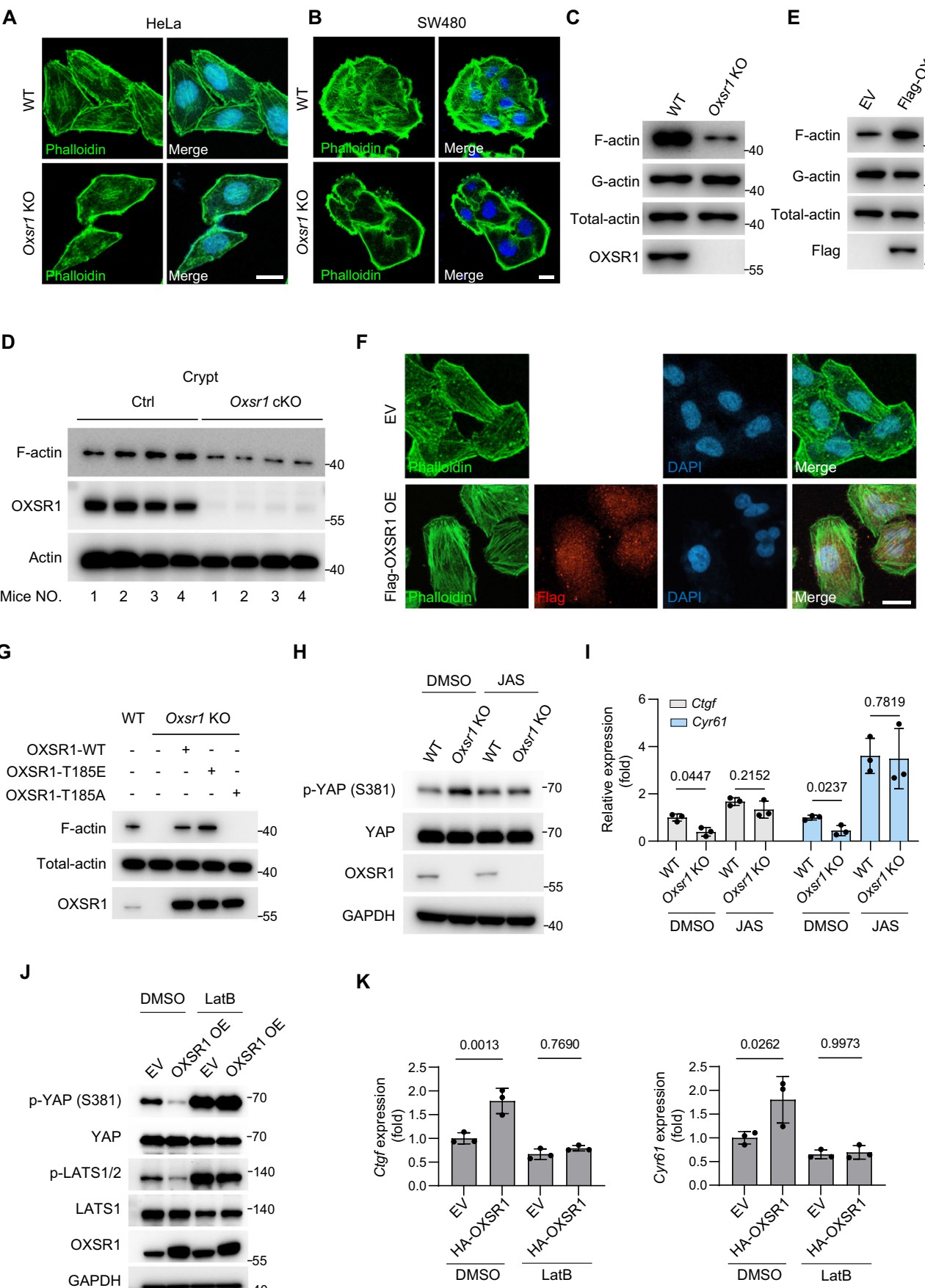

**Figure 5.   OXSR1 suppresses the Hippo signaling via promoting F-actin assembly.**

(A, B) Phalloidin staining of F-actin in wild-type or *Oxsr1* knockout HeLa (A) or SW480 (B) cells. Note that knockout of *Oxsr1* suppressed the abundance of Phalloidin-positive F-actin fibers. Scale bars: 10 µm. (C) Western blot was performed to examine the levels of detergent-insoluble F-actin and total actin in HEK293T cells of the indicated genotypes. Note the reduced F-actin levels in *Oxsr1* knockout cells. (D) Similar as in (C), except that primary colonic crypts of the indicated genotypes collected at regeneration day 4 were analyzed. For each genotype, crypts isolated from four independent animals were included. (E) Western blot analysis of F-actin in wild-type HEK293T cells and cells stably expressing OXSR1. Note that OXSR1 overexpression enhanced F-actin levels. (F) Phalloidin staining in wild-type HeLa cells and cells stably expressing OXSR1. Note that OXSR1 overexpression enhanced F-actin levels. Scale bar: 10 µm. (G) Similar to (C). Note that the diminished F-actin levels in *Oxsr1*-null HEK293T cells were rescued by reconstitution with OXSR1-WT or OXSR1-T185E, but not by OXSR1-T185A. (H) Western blot was performed in wild-type or *Oxsr1* knockout HEK293A cells treated with DMSO or Jasplakinolide (JAS, 300 nM, 2 h). Note that the increased phosphorylation of YAP observed in *Oxsr1* knockout cells was fully blocked by JAS treatment. (I) RT-qPCR was performed in wild-type or *Oxsr1* knockout HEK293T cells treated with DMSO or JAS (300 nM, 3 h). Note that the downregulation of YAP target genes *Ctgf* and *Cyr61* in *Oxsr1* knockout cells was fully rescued by JAS treatment. Data were analyzed using two-way ANOVA followed by Sidak's multiple comparisons test and are presented as mean ± s.d., $n = 3$ independent experiments. (J) HeLa cells stably expressing empty vector (EV) or OXSR1 were treated with DMSO or Latrunculin B (LatB, 500 ng/mL, 1.5 h). Note that the reduced phosphorylation of endogenous LATS and YAP caused by OXSR1 overexpression was rescued by LatB treatment. (K) HEK293T cells stably expressing empty vector (EV) or OXSR1 were treated with DMSO or Latrunculin B (LatB, 500 ng/mL, 1.5 h). The expression of the YAP target genes *Ctgf* and *Cyr61* was analyzed by RT-qPCR. Note that the elevated expression of *Ctgf* and *Cyr61* caused by OXSR1 overexpression was blocked by LatB treatment. Data were analyzed using two-way ANOVA followed by Sidak's multiple comparisons test and are presented as mean ± s.d., $n = 3$ independent experiments. The gel and microscopy images shown are representative of at least two independent experiments.

expression of YAP target genes (Fig. 5K), caused by OXSR1 overexpression was restored by the actin polymerization inhibitor latrunculin B (LatB). These data suggest that OXSR1 suppresses the Hippo signaling via promoting F-actin assembly.

## OXSR1 promotes F-actin assembly by boosting the amount of GTP-bound RhoB

After identifying an important role for OXSR1 in F-actin assembly, we further investigated the mechanisms underlying this process. By searching the BioGRID database (Oughtred et al, 2021), we found that multiple small GTPase such as RhoB and Rac1, well-known regulators of actin dynamics (Tapon and Hall, 1997), as well as Rho GTPase-activating protein (RhoGAP) ARHGAP17 (Harada et al, 2000; Richnau and Aspenstrom, 2001), were identified in the OXSR1 interactome. This result prompted us to investigate the roles of RhoB and Rac1 in OXSR1-mediated F-actin assembly.

Rho GTPase acts as a molecular switch that cycles between an inactive GDP-bound state and an active GTP-bound state (Etienne-Manneville and Hall, 2002). The GTP-bound Rho activates two different kinds of molecules, WASP/WAVE proteins and Diaphanous-related formins, to stimulate actin polymerization (Ridley, 2006). To investigate the roles of RhoB and Rac1 in OXSR1-mediated F-actin assembly, we first examined whether OXSR1 regulates the activity of these small GTPases. We compared the levels of GTP-bound RhoB and Rac1 in wild-type and *Oxsr1*-null cells using pull-down assays with GST-tagged Rho-binding domains (GST-RBD) from the Rho downstream effector Rhotekin and GST-tagged Rac/p21-binding domain (GST-PBD) from the Rac/p21 downstream effector PAK1, respectively (Reid et al, 1996). As shown in Fig. 6A,B, the levels of GTP-bound RhoB were markedly reduced in *Oxsr1*-null HEK293T cells compared with wild-type cells, whereas GTP-bound Rac1 showed only a modest decrease. Similarly, reduced levels of GTP-bound RhoB were also observed in SW480 colorectal epithelial cells (Fig. 6C). In addition, we also examined the activity of two closely related Rho GTPases, RhoA and RhoC, which were not identified in the OXSR1 interactome. However, their GTP-bound levels remained unchanged in *Oxsr1* knockout cells (Fig. 6D,E). These data suggest that RhoB is a primary target of OXSR1. Importantly, reintroducing the constitutively active OXSR1-T185E mutant, but not the kinase-

dead OXSR1-T185A mutant, rescued the diminished RhoB-GTP levels in *Oxsr1*-null cells, highlighting the critical role of OXSR1 kinase activity in this process (Fig. 6F). Furthermore, we assessed the effects of RhoB and Rac1 on the Hippo pathway by examining the phosphorylation levels of both exogenous and endogenous YAP. Our results showed that YAP phosphorylation was reduced only when co-expressed with RhoB, not with Rac1 (Fig. 6G,H). These findings suggest that OXSR1 may promote F-actin assembly by increasing the active form of RhoB, thereby further suppressing the Hippo signaling.

To corroborate these findings, we next examined whether inhibiting RhoB activity can inhibit the increased F-actin assembly and the reduced LATS/YAP phosphorylation caused by OXSR1 overexpression. As shown in Fig. 6I,J, inhibition of RhoB with the C3 toxin, a potent Rho inhibitor, suppressed the enhanced F-actin assembly by OXSR1 overexpression. Similarly, blocking RhoB activity with RhoB[T19N], a dominant-negative form of RhoB, also abolished the increased F-actin assembly caused by OXSR1 overexpression (Fig. 6J,K). Consistently, the reduced LATS1/YAP phosphorylation (Fig. 6L) and the increased nuclear localization of YAP (Fig. 6M) by OXSR1 overexpression were both rescued by C3 treatment. Altogether, these data indicate that OXSR1 increases the active form of RhoB to promote F-actin assembly, thereby suppressing Hippo signaling.

## OXSR1 phosphorylates RhoB and inhibits its interaction with ARHGAP17

We next explored how OXSR1 boosts the levels of active GTP-bound RhoB (hereafter referred to as 'RhoB-GTP'). Interestingly, we observed a strong direct interaction between OXSR1 and RhoB in an in vitro pull-down assay using bacterially purified GST-OXSR1 and His-RhoB (Fig. 7A). Given that phosphorylation is a key post-translational modification regulating Rho activity (Olson, 2018), we reasoned that OXSR1 may directly phosphorylate RhoB. To test this possibility, we conducted an in vitro kinase assay using bacterially purified RhoB as the substrate. In this assay, ATP-γS was used as a phosphate donor to generate a thiophosphorylated substrate, which was subsequently alkylated with p-nitrobenzyl mesylate to form a thiophosphate ester that can be detected using a thiophosphate ester-specific antibody (Allen et al, 2007).

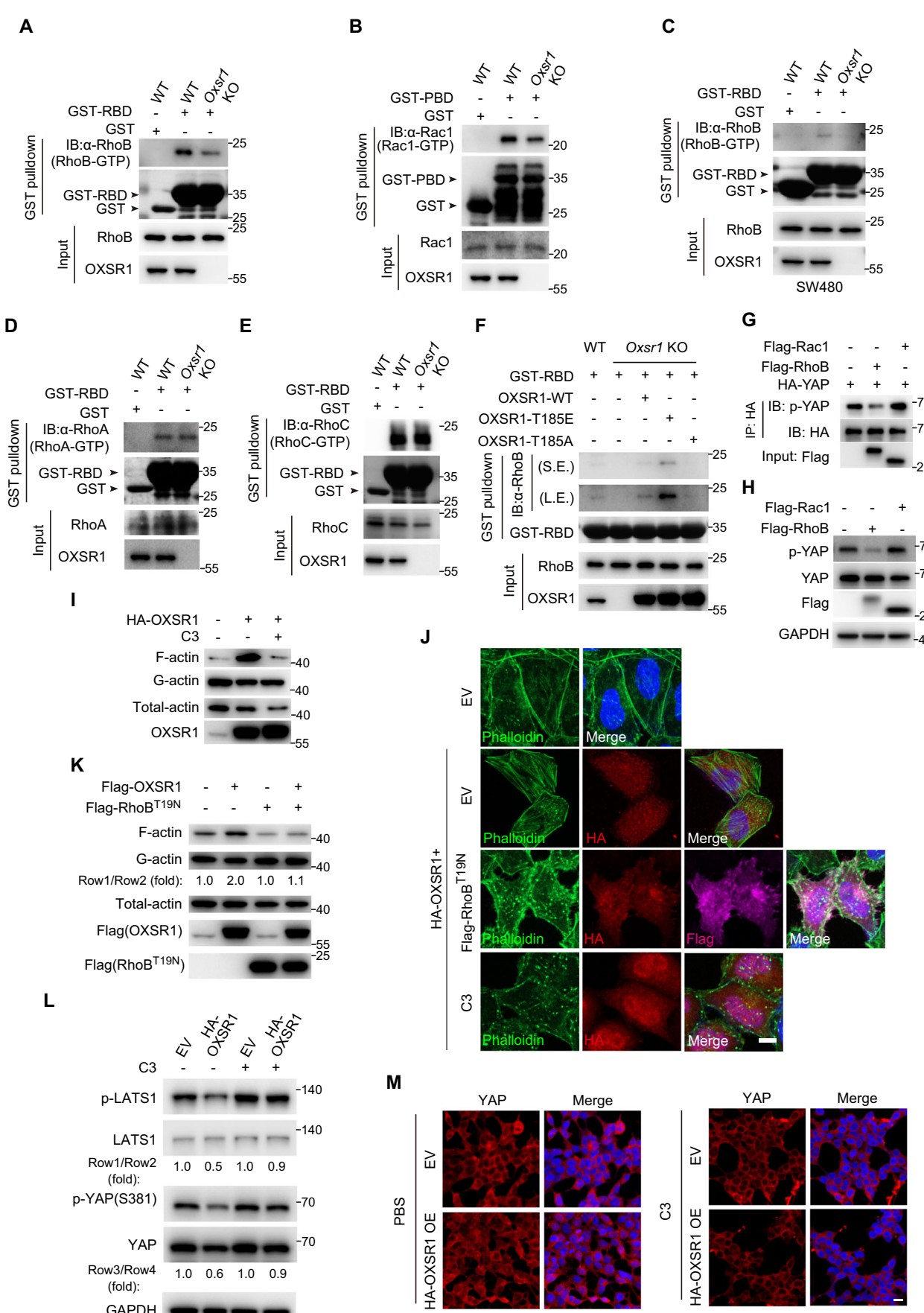

**Figure 6. OXSR1 promotes F-actin assembly by boosting the amount of GTP-bound RhoB.**

(A–E) GTP-bound RhoB (A, C), Rac1 (B), RhoA (D), and RhoC (E) in wild-type and *Oxsr1* knockout HEK293T cells (A, B, D, E) or SW480 cells (C) were pulled down using GST-tagged Rho-binding domain of human Rhotekin (GST-RBD) or GST-tagged Rac/p21-binding domain of PAK (GST-PBD). Note that the *Oxsr1* knockout cells showed a marked reduction in GTP-bound RhoB and a moderate decrease in GTP-bound Rac1, while levels of GTP-bound RhoA and RhoC remained unchanged. (F) Similar to (A–E) except for the cell genotypes. Note that reintroducing OXSR1-T185E, but not OXSR1-T185A, rescued the diminished RhoB-GTP levels in *Oxsr1*-null HEK293T cells. S.E., short-exposure, L.E., long-exposure. (G) Co-IP was performed in HEK293T cells transfected with the indicated plasmids. YAP phosphorylation levels were assessed by western blot. Note the reduced phosphorylation of exogenous YAP upon co-expression with Flag-RhoB, but not with Flag-Rac1. (H) Western blot was performed in HEK293T cells transfected with the indicated plasmids. Note the reduced phosphorylation of endogenous YAP by RhoB, but not Rac1. (I) Western blot was performed to examine the levels of detergent-insoluble F-actin in HEK293T cells with the indicated treatment. Note that the elevated levels of F-actin led by OXSR1 overexpression were blocked by C3 treatment (0.1 mg/mL, 6 h). (J) Phalloidin staining in HeLa cells with the indicated treatment. Note that the OXSR1-mediated increase in F-actin fibers was abolished by either co-expression of the dominant-negative Flag-RhoB[T19N] mutant or treatment with C3 toxin (0.1 mg/mL, 6 h). EV, empty vector. Scale bar: 10 μm. (K) Similar to (I) except for the transfected plasmids. The F-actin over G-actin levels were quantified and indicated below the blots. Note that the OXSR1-induced increase in F-actin levels was abolished by co-expression of the dominant-negative Flag-RhoB[T19N] mutant. (L, M) HEK293T cells that stably express HA-OXSR1 were treated with C3 (0.1 mg/mL, 6 h). LATS1/YAP phosphorylation levels and YAP subcellular localization were assessed by western blot (L) and immunostaining (M), respectively. For (L), the p-LATS1 over total LATS1 levels and p-YAP over total YAP levels were quantified and indicated below the blots. Note that the reduced levels of LATS1 and YAP (S381) phosphorylation caused by OXSR1 overexpression were rescued by C3 treatment (L). Also note that the increased nuclear localization of YAP induced by HA-OXSR1 overexpression was blocked by C3 treatment (M). Scale bar: 10 μm. Data shown are representative of at least two independent experiments.

Notably, we observed that bacterially purified GST-tagged wild-type OXSR1 (OXSR1-WT), but not the kinase-dead variant OXSR1-K46R (Chen et al, 2004), strongly phosphorylated RhoB (Fig. 7B). These data indicate that RhoB is a direct substrate of OXSR1. Furthermore, OXSR1's kinase activity is essential for promoting RhoB-GTP levels, as the reintroduction of OXSR1-WT, but not OXSR1-K46R, into *Oxsr1*-null HEK293T cells rescued the decreased RhoB-GTP levels resulting from *Oxsr1* knockout (Fig. 7C).

The activation of Rho proteins is catalyzed by RhoGEFs (guanine nucleotide exchange factors), which stimulate the release of GDP, thus allowing GTP to bind. On the contrary, the inactivation of Rho proteins is mediated by RhoGAPs (GTPase-activating proteins), which catalyze GTP hydrolysis, converting Rho proteins into the GDP-bound inactive conformation (Etienne-Manneville and Hall, 2002; Hodge and Ridley, 2016). Therefore, RhoB phosphorylation by OXSR1 may enhance the RhoB/RhoGEF interaction or suppress the RhoB/RhoGAP interaction to promote RhoB-GTP levels. Given that ARHGAP17 has been identified in the OXSR1 interactome in the BioGRID database, we first examined this RhoGAP. To this end, we first verified the association between bacterially purified His-ARHGAP17 and GST-RhoB via in vitro pulldown assay. Indeed, we observed that ARHGAP17 is strongly associated with RhoB (Fig. EV5A). Furthermore, a pull-down assay performed with GST-RBD revealed that cells with ARHGAP17 overexpression had reduced levels of RhoB-GTP (Fig. 7D). This data confirmed that ARHGAP17 indeed functions as a RhoGAP for RhoB.

Based on these findings, including (1) OXSR1 directly phosphorylates RhoB, and its kinase activity is essential for promoting RhoB-GTP levels; (2) ARHGAP17 negatively regulates RhoB-GTP levels; and (3) both RhoB and ARHGAP17 are identified as OXSR1 binding partners in the BioGRID database, we proposed that OXSR1 directly phosphorylates RhoB, disrupting the association between RhoB and ARHGAP17, ultimately enhancing RhoB-GTP levels. To test this model, bacterially purified GST-RhoB was subjected to an in vitro kinase assay with bacterially purified OXSR1 kinase, in the presence or absence of ATP. The phosphorylated RhoB was then incubated with HA-ARHGAP17 immunoprecipitated from HEK293T cell lysates to assess the interaction between RhoB and ARHGAP17 (Fig. EV5B). The results demonstrated that adding ATP to the in vitro kinase assay

significantly attenuated the association between RhoB and ARHGAP17 (Fig. 7E). Furthermore, OXSR1-WT, but not OXSR1-K46R, markedly reduced RhoB binding to ARHGAP17 (Fig. 7F). These results suggest that OXSR1 phosphorylates RhoB, inhibiting its interaction with ARHGAP17.

To further corroborate these findings, we aimed to identify the OXSR1 phosphorylation site on RhoB. Given that OXSR1-mediated phosphorylation of RhoB disrupts its interaction with ARHGAP17, we postulated that the phosphorylation site resides within the ARHGAP17 interaction region of RhoB. To test this, we first mapped the RhoB region that interacts with ARHGAP17 by dividing RhoB into two fragments and examining their interactions with ARHGAP17. The results showed that the RhoB fragment spanning amino acids 1–68 exhibited a strong interaction with ARHGAP17 (Fig. 7G). Further domain analysis revealed that this fragment contains three conserved signatures of Rho proteins: the P-loop (aa 12–19), Switch I (aa 27–42), and Switch II (aa 57–68). Each of these functional domains contributes to the RhoGAP interaction interface characterized in a previous study (Amin et al, 2016) (Fig. EV5C). This interface was further confirmed in a 3D protein structure model of RhoB/ARHGAP17 interaction predicted by AlphaFold (Fig. EV5D). Strikingly, only one Ser/Thr residue (T37) was identified within this interaction interface (Fig. EV5C). Since OXSR1 is a Ser/Thr kinase, we then tested whether this residue serves as the OXSR1 phosphorylation site. By performing in vitro kinase assay, we found that mutating T37 to Alanine (RhoB-T37A) strongly diminished the OXSR1-induced phosphorylation of RhoB (Fig. 7H), suggesting that T37 is a primary phosphorylation site for OXSR1 on RhoB. To further demonstrate that T37 phosphorylation disrupts the interaction between RhoB and ARHGAP17, we compared the interactions of ARHGAP17 with wild-type RhoB, the non-phosphorylatable RhoB-T37A, and the phosphomimetic RhoB-T37E. Co-IP results indicated that the interaction between ARHGAP17 and RhoB-T37A was stronger than that with wild-type RhoB, while RhoB-T37E exhibited a dramatically weaker interaction (Fig. 7I). Moreover, OXSR1-mediated phosphorylation effectively disrupted the interaction between ARHGAP17 and wild-type RhoB, but not RhoB-T37A (Fig. 7J).

To further substantiate that phosphorylation of RhoB at T37 impairs its interaction with ARHGAP17, we directly assessed the effect of ARHGAP17 on the GTPase activity of different RhoB variants using an in vitro GTPase assay. In this assay, bacterially purified GST-tagged RhoB variants were incubated with His-tagged

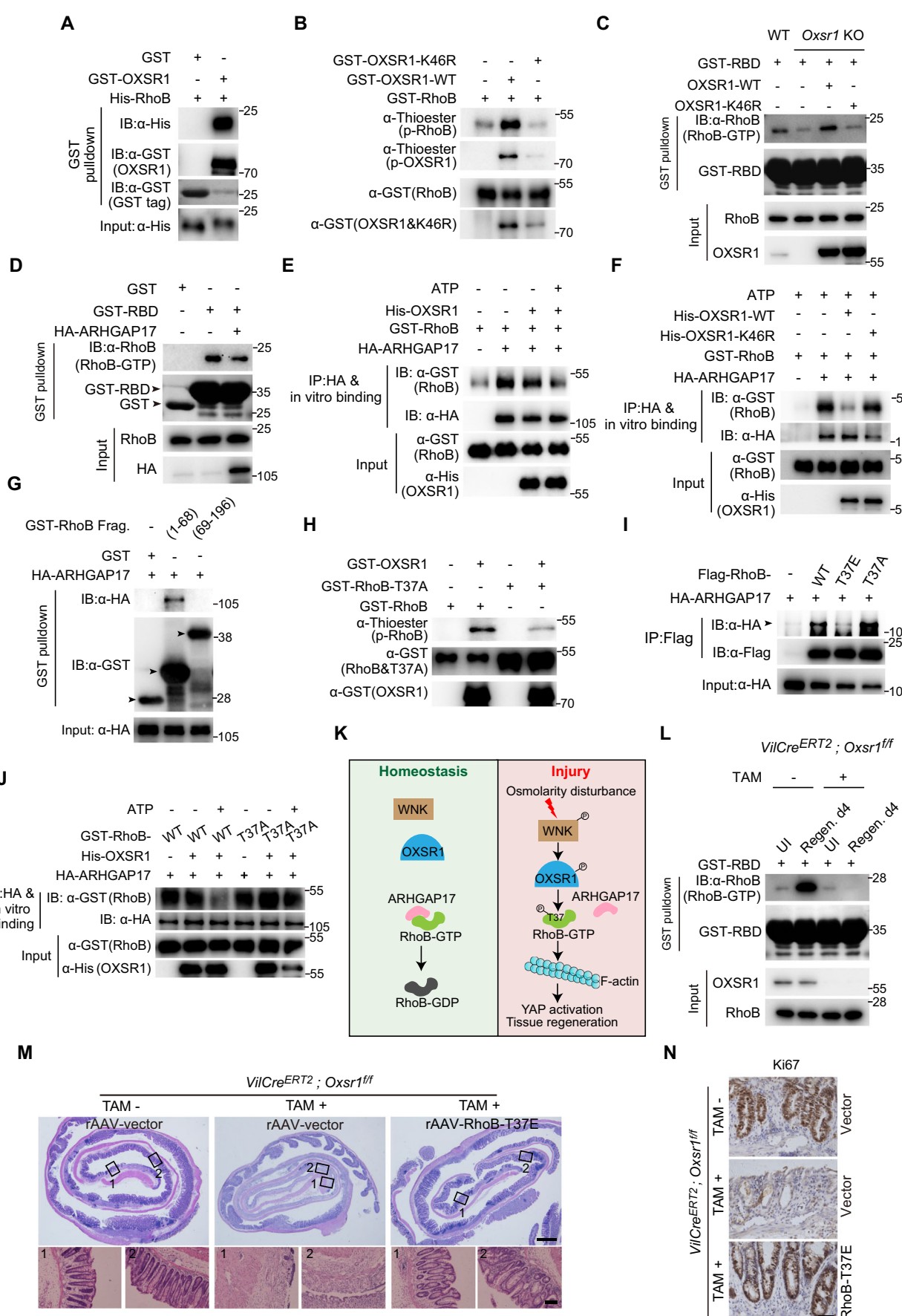

**Figure 7. OXSR1 phosphorylates RhoB and inhibits its interaction with ARHGAP17.**

(A) GST pull-down assay was performed using bacterially purified His-RhoB and GST-OXSR1. Note the direct interaction between RhoB and OXSR1. (B) An in vitro kinase assay was performed using bacterially purified GST-RhoB, GST-OXSR1-WT, and GST-OXSR1-K46R. Note that OXSR1-WT, but not the kinase-dead mutant OXSR1-K46R, phosphorylated GST-RhoB. (C) OXSR1-WT and OXSR1-K46R were reintroduced into *Oxsr1*-null HEK293T cells. GTP-bound RhoB was pulled down using GST-RBD and assessed by western blot. Note that re-expression of OXSR1-WT, but not OXSR1-K46R, restored the reduced levels of RhoB-GTP caused by *Oxsr1* knockout. (D) GTP-bound RhoB in HEK293T cells transfected with HA-ARHGAP17 was pulled down using GST-RBD and assessed by western blot. Note that overexpression of HA-ARHGAP17 reduced the level of RhoB-GTP. (E) Bacterially purified recombinant His-OXSR1 and GST-RhoB were subjected to in vitro kinase assay, then incubated with HA-ARHGAP17 immunoprecipitated from HEK293T cell lysates. Note that adding ATP to the in vitro kinase assay significantly attenuated the association between RhoB and ARHGAP17. (F) Similar to (E) except for the bacterially purified proteins. Note that OXSR1-WT, but not OXSR1-K46R, markedly reduced RhoB binding to ARHGAP17. (G) Bacterially purified GST-RhoB truncations were used to pull down HA-ARHGAP17 in the HEK293T cell lysates. Note the interaction between RhoB (1–68 aa) and ARHGAP17. Arrowheads indicated the GST-tagged RhoB fragments. (H) In vitro kinase assay was performed using bacterially purified GST-RhoB, GST-RhoB-T37A, and GST-OXSR1. Note that mutating T37 to Alanine (RhoB-T37A) significantly reduced the OXSR1-induced phosphorylation of RhoB. (I) Co-IP assay was performed in HEK293T cells transfected with the indicated plasmids. Arrowhead indicated HA-ARHGAP17 band. Note that compared to wild-type RhoB, the RhoB-T37A mutant exhibited enhanced interaction with ARHGAP17, while the phosphomimetic RhoB-T37E mutant showed a markedly reduced interaction. (J) Similar to (E) except for the bacterially purified proteins. Note that OXSR1 disrupted the interaction between ARHGAP17 and wild-type RhoB, but did not affect the interaction between ARHGAP17 and the RhoB-T37A mutant. (K) A schematic model depicting the WNK-OXSR1-RhoB-F-actin signaling axis in the regulation of YAP activation and tissue regeneration. Upon osmotic disturbance caused by tissue damage, the WNK–OXSR1 axis is activated and phosphorylates RhoB at T37, disrupting its interaction with ARHGAP17. This increases GTP-bound RhoB levels, promoting F-actin polymerization and YAP activation to facilitate intestinal regeneration. (L) Crypts from *Oxsr1* cKO mice and their control littermates were isolated at the indicated time points following DSS treatment. GTP-bound RhoB was pulled down using GST-RBD and assessed by western blot. Note that the elevated RhoB-GTP levels observed during gut regeneration in control mice were abolished in *Oxsr1* cKO mice. (M, N) Recombinant adeno-associated virus carrying RhoB-T37E (rAAV-RhoB-T37E) was delivered to *Oxsr1* cKO and control mice. Colon sections were collected on day 4 of the regeneration stage and subjected to H&E (M) and Ki67 (N) staining. Note that overexpression of the phosphomimetic RhoB-T37E restored crypt regeneration and cell proliferation impaired by *Oxsr1* deletion. Scale bars in (M): 500 μm (upper panel) and 100 μm (lower panel). Scale bar in (N): 100 μm. Data shown are representative of at least two independent experiments.

ARHGAP17, and the inorganic phosphate (Pi) released from GTP hydrolysis was quantified via a colorimetric method. As shown in Fig. EV5E, the GTPase activities of wild-type RhoB and the nonphosphorylatable RhoB-T37A were robustly enhanced by ARHGAP17, whereas the phosphomimetic RhoB-T37E showed only a modest increase in activity. These results not only confirm ARHGAP17 as a GAP for RhoB by targeting the T37 residue but also underscore the essential role of T37 phosphorylation in regulating this process. Altogether, we propose a working model in which OXSR1 phosphorylates RhoB at T37, disrupting its association with ARHGAP17 and increasing the levels of GTP-bound RhoB. The elevated GTP-bound RhoB promotes F-actin assembly and YAP activation, facilitating the regeneration process (Fig. 7K). In support of this model, the reduction in YAP activity observed upon *Oxsr1* knockout was partially rescued by *Arhgap17* depletion (Fig. EV5F). Consistently, we observed a marked increase in RhoB-GTP levels in the crypts of control mice during intestinal regeneration, highlighting its essential role in this process (Fig. 7L). Notably, this increase was completely absent in *Oxsr1* cKO mice, suggesting that RhoB activation during intestinal regeneration is OXSR1-dependent (Fig. 7L). Furthermore, intraperitoneal delivery of rAAV encoding the phosphomimetic RhoB mutant, T37E, (rAAV-RhoB-T37E) significantly rescued the regeneration and proliferation defects observed in *Oxsr1* cKO mice (Figs. 7M,N and EV5G–I). In addition, sorbitol treatment induced a substantial increase in both RhoB-GTP and F-actin levels in wild-type SW480 intestinal epithelial cells, but these effects were abolished in *Oxsr1*-null SW480 cells (Fig. EV5J,K), further emphasizing the crucial role of the OXSR1-RhoB-actin cascade in sensing osmolarity disturbances.

## Targeting the WNK–OXSR1 axis restricts the oncogenic potential of the intestinal regeneration program

Excessive and sustained regenerative proliferation can lead to tumorigenesis (Beachy et al, 2004; Terzic et al, 2010), and

hyperactivation of YAP has been associated with widespread early-onset polyp formation following DSS treatment (Cai et al, 2010). To further highlight the clinical relevance of our findings, we tested whether inhibiting WNK or OXSR1 kinase activity could reduce the risk of tumorigenesis during intestinal regeneration. To this end, we utilized the well-established Azoxymethane (AOM)/DSS model of colitis-associated cancer (CAC) (Parang et al, 2016), in which mice receive the procarcinogen AOM, followed by DSS to induce colonic polyp formation (Appendix Fig. S2A). Interestingly, immunohistochemical analysis showed elevated levels of phosphorylated OXSR1, YAP protein, and the proliferation marker Ki67 in colonic polyps compared to adjacent non-tumorous tissue (Appendix Fig. S2B), underscoring the potential roles of OXSR1 and YAP in promoting cell proliferation and polyp formation. Consistently, the administration of Rafo or WNK463, inhibitors of OXSR1 or WNK, respectively, dramatically inhibited the number of AOM/DSS-induced polyps (Appendix Fig. S2C,D). Altogether, these findings provide proof-of-principle evidence that targeting the WNK-OXSR1-YAP signaling axis may serve as a potential therapeutic strategy to mitigate oncogenic potential during tissue regeneration.

## Discussion

Regenerative responses are critical in the gastrointestinal tract, which is constantly subjected to damage and requires frequent renewal. The microenvironment plays a pivotal role in intestinal regeneration, with several factors such as mesenchymal cells, immune cells, bacteria, enteric neuronal cells, the extracellular matrix, and nutritional status working in concert to regulate this process (Hageman et al, 2020). Damaged cells and epithelial barrier function can cause the release of intracellular contents and bodily fluids rich in ions and solutes into the extracellular microenvironment, potentially leading to osmolarity disturbance. This suggests that osmolarity could be an important damage-associated signal

influencing intestinal repair. However, the specific role of osmolarity changes in intestinal regeneration remains poorly defined. In this study, we find that the osmolarity-sensing WNK-OXSR1 signaling pathway is essential for intestinal regeneration. Mechanistically, we show that OXSR1 phosphorylates RhoB at T37, disrupting its association with ARHGAP17, leading to increased levels of GTP-bound RhoB. The elevated GTP-bound RhoB enhances F-actin assembly and promotes YAP activation, which in turn drives the regeneration response. These findings not only highlight osmolarity as a key damage-associated signal in intestinal regeneration but also reveal the molecular mechanisms by which osmolarity influences this process.

It should be noted that tissue damage elicits a complex and multifaceted cellular response, as injured cells can release a variety of molecules that can activate multiple signaling pathways simultaneously. In the present study, our data support a prominent role for osmotic changes in mediating the observed signaling responses following tissue damage. However, a more detailed characterization of regional intestinal osmolality dynamics during regeneration would be valuable to further substantiate these findings, pending the development of more tractable experimental systems and methodologies. Moreover, other damage-associated signals, such as released ATP, cytokines, growth factors, or ions, as well as changes in mechanical tension and cell-matrix interactions, may also contribute to downstream pathway activation. Given the inherent complexity of injury responses, it remains challenging to fully disentangle the relative contributions of these parallel inputs using the current experimental system. Future studies employing more refined approaches to selectively manipulate individual damage-associated signals will be important to further delineate their respective roles. Moreover, although our results identify Hippo-YAP signaling as a key downstream effector of OXSR1 in intestinal regeneration, other regeneration-associated pathways, such as MAPK and Wnt signaling, are also activated during regeneration in an OXSR1-dependent manner. The relative contributions of these pathways to OXSR1-mediated intestinal regeneration, as well as their potential interplay in this process, warrant further investigation.

The WNK-SPAK/OXSR1 pathway is well-known for its role in maintaining cellular ion and volume homeostasis in response to osmotic stress. Interestingly, WNK, SPAK, and OXSR1 are expressed in a variety of tissues, some of which experience minimal osmotic stress, suggesting that these proteins may have additional functions beyond regulating cellular volume. Although there is currently no direct evidence linking the WNK-SPAK/OXSR1-N(K) CC axis to tissue regeneration, this signaling pathway has been implicated in critical cellular processes such as proliferation, migration, cytoskeleton remodeling, and angiogenesis (Algharabil et al, 2012; Dbouk et al, 2014; Gallolu Kankanamalage et al, 2018; Jung and Cobb, 2023; Jung et al, 2022; Shiozaki et al, 2014; Xie et al, 2013; Zhang et al, 2023). It is therefore not unexpected that this signaling axis contributes to the regulation of tissue homeostasis. Our results provide direct evidence for a role of WNK and OXSR1 in tissue regeneration, thereby extending the functional significance of the WNK-OXSR1 pathway to broader physiological contexts. Further mechanistic analysis identifies YAP as a key downstream effector of OXSR1 in intestinal regeneration. YAP is expressed in most intestinal cells, except for Paneth cells (Gregorieff et al, 2015).

Previous studies have demonstrated that YAP is strongly induced during intestinal regeneration, and that its deletion from either the whole intestinal epithelium or specifically from intestinal stem cells (ISC) results in impaired ISC proliferation and compromised regeneration (Cai et al, 2010; Gregorieff et al, 2015). Our results show that epithelial-wide inactivation of OXSR1 abolished YAP induction and severely disrupted intestinal regeneration. However, whether OXSR1 functions cell-autonomously to promote ISC proliferation and intestinal regeneration requires further investigation using ISC-specific Oxsr1 deletion. In addition, while our study primarily employs Oxsr1 conditional knockout mice, which clearly establish OXSR1 as essential for intestinal regeneration, it remains unclear whether OXSR1 activation alone is sufficient to drive intestinal regeneration. This question is of particular importance, as it could offer valuable insights into potential therapeutic strategies for inflammatory bowel disease (IBD) through targeted OXSR1 activation. Future studies using OXSR1 transgenic mice could help address this issue.

SPAK and OXSR1, despite being closely related, have distinct phenotypic outcomes in knockout mice. Spak-null mice do not show overt developmental defects (Delpire and Gagnon, 2008; Yang et al, 2010), whereas Oxsr1-null mice are embryonic lethal (Delpire and Gagnon, 2008; Lin et al, 2011), suggesting that SPAK and OXSR1 perform different roles in animal development. Indeed, a previous study has demonstrated a functional separation between SPAK and OXSR1 in angiogenesis in vitro (Dbouk et al, 2014). Our results, showing that OXSR1, but not SPAK, promotes cell proliferation, further reinforce the distinct functions of these homologs. Moreover, while hypertonicity is known to induce rapid F-actin polymerization, the underlying mechanisms remain complex and poorly understood (Bustamante et al, 2003; Di Ciano et al, 2002; Lewis et al, 2002; Thirone et al, 2009; Yamamoto et al, 2006). Our study provides new insights into this process by showing that OXSR1 phosphorylates RhoB, inhibiting its interaction with ARHGAP17 and enhancing the levels of active RhoB, which in turn promotes F-actin assembly. It is worth noting that, although our results implicate an important role for ARHGAP17 in the regulation of RhoB by OXSR1, the potential involvement of other RhoGAPs remains to be determined, particularly given the promiscuity and relatively low substrate selectivity of RhoGAPs (Muller et al, 2020). Our findings reveal that the phosphorylation of the RhoB T37 residue is essential for disrupting the RhoB-ARHGAP17 association. Notably, this residue is highly conserved across the Rho GTPase family (Amin et al, 2016), suggesting its functional significance. Interestingly, the equivalent residue (T35) in Ras GTPase family proteins mediates a so-called 'loaded-spring mechanism', which is critical for GTPases to bind their regulators (Vetter and Wittinghofer, 2001). Phosphorylation of this residue may disrupt this mechanism, thereby impairing the GTPase-GAP interaction. Further structural studies are needed to explore this possibility.

The Hippo signaling pathway plays a key role in intestinal regeneration (Gregorieff et al, 2015; Hong et al, 2016). Previous studies have shown that YAP, the nuclear effector of this pathway, is essential for intestinal regeneration after tissue injury (Cai et al, 2010). While an increase in YAP protein levels is observed 2–5 days after DSS-induced colitis and during regeneration, the upstream signals that mediate YAP activation following intestinal injuries remain

incompletely understood. Previous studies have demonstrated that a gp130-Src-YAP axis mediates YAP activation and that group 3 innate lymphoid cells (ILC3s) are essential for this process in response to inflammatory cues (Romera-Hernandez et al, 2020; Taniguchi et al, 2015). Our findings reveal an increase in intestinal osmolarity and suggest hyperosmolarity as an additional physiological cue for YAP induction during intestinal regeneration. Further mechanistic investigations identify the osmosensing WNK-OXSR1 axis as a key player connecting the damage-associated osmolarity changes to pro-regeneration Hippo-YAP signaling. In keeping with our findings, a recent study demonstrated that hyperosmotic stress promotes the growth of intestinal organoids (Li et al, 2021).

Consistent with the role of hyperosmotic stress in promoting YAP activation, a previous study in cultured cell lines has shown that hyperosmotic stress reduces phosphorylation of LATS and YAP (S127), promotes YAP nuclear localization, and enhances YAP transcription activity under serum-starvation conditions, in which basal YAP phosphorylation levels are substantially elevated (Hong et al, 2017). Paradoxically, previous studies have also reported increased phosphorylation of YAP at S127 following hyperosmotic stress (Hong et al, 2020; Hong et al, 2017; Wang et al, 2022). One possible explanation for this discrepancy is that the effects of hyperosmotic stress on YAP phosphorylation are intensity-dependent, with mild stress decreasing YAP phosphorylation while strong stress increasing it. Whereas strong hyperosmotic stress engages multiple mechanisms that activate Hippo signaling and elevate YAP phosphorylation (Hong et al, 2020; Wang et al, 2022), we show that mild hyperosmotic stress activates the WNK-OXSR1 axis, which in turn suppresses Hippo signaling and reduces YAP phosphorylation. An additional explanation may relate to differences in the basal phosphorylation state of YAP, which is known to be highly sensitive to variables such as cell density and serum concentration. When basal YAP phosphorylation levels are high, decreases are more readily detectable, whereas when basal levels are already low, further reductions may fall below the detection threshold. Moreover, pharmacological inhibition of the WNK-OXSR1 axis mitigates the oncogenic risk associated with intestine regeneration, further highlighting the clinical relevance of our findings. Collectively, these results provide deeper mechanistic insight into YAP activation during intestinal regeneration and offer potential therapeutic avenues for IBD and intestinal tumors. Future studies investigating the interplay among diverse signaling pathways and their relative contributions to YAP activation and intestinal regeneration will be essential to fully elucidate the complex regulatory networks governing tissue repair and regeneration.

# Methods

### Reagents and tools table

| Reagent/resource | Reference or source | Identifier or catalog number |
| --- | --- | --- |
| **Experimental models** | | |
| Human HeLa cells | ATCC | CCL-2 |
| Human HEK293T | ATCC | CRL-11268 |
| Human HEK293A | ATCC | CRL-1573 |

| Reagent/resource | Reference or source | Identifier or catalog number |
| --- | --- | --- |
| Human HEK293FT | ThermoFisher | R70007 |
| Human SW480 | ATCC | CCL-228 |
| *Mst1/2; MAP4K1/2/3/4/5/6/7 knockout (MM-9KO) HEK293A* | Provided by Drs. Kun-Liang Guan and Fa-Xing Yu | |
| Mouse: *Oxsr1^fl/fl* | Cyagen | S-CKO-17935 |
| Mouse: Villin-CreERT2 | Cyagen | C001433 |
| *D. melanogaster*: UAS-Fray: w[*]; M{RFP[3xP3.PB] w[+mC]=UAS-fray.R}ZH-22A | Bloomington Drosophila Stock Center | RRID: BDSC_99477 |
| *D. melanogaster*: UAS-Yki: w[*]; P{y[+t7.7] w[+mC] =UAS-yki.V5.O}attP2 | Bloomington Drosophila Stock Center | RRID: BDSC_28819 |
| *D. melanogaster*: UAS-Yki-3SA: w[*]; P{y[+t7.7] w[+mC]=UAS-yki.S111A.S168A.S250A.V5} attP2 | Bloomington Drosophila Stock Center | RRID: BDSC_28817 |
| *D. melanogaster*: Fray RNAi: y[1] v[1]; P{y[+t7.7] v[+t1.8] =TRiP.GL00704}attP40/ CyO | Bloomington Drosophila Stock Center | RRID: BDSC_41587 |
| *D. melanogaster*: Fray RNAi: y[1] v[1]; P{y[+t7.7] v[+t1.8] =TRiP.HMJ02228}attP40 | Bloomington Drosophila Stock Center | RRID: BDSC_42569 |
| *D. melanogaster*: UAS-Fray RNAi | Vienna Drosophila Resource Center | RRID: VDRC#:106919 |
| *D. melanogaster*: escargot-Gal4; UAS-mCD8GFP (esg>GFP) | Provided by Dr. Xianjue Ma | |
| **Recombinant DNA** | | |
| PCDNA3.1-FLAG-RhoB | This paper | |
| PCDNA3.1-FLAG-RAC1 | This paper | |
| PCDNA3.1-HA-MAP4K2 | This paper | |
| PCDNA3.1-HA-OXSR1 | This paper | |
| PCDNA3.1-FLAG-OXSR1 | This paper | |
| PCDNA3.1-HA-ARHGAP17 | This paper | |
| PCDNA3.1-HA-MST2 | This paper | |
| PCDNA3.1-HA-YAP | This paper | |
| PCDNA3.1-V5-SLMAP | This paper | |
| PCDNA3.1-FLAG-RhoB-T19N | This paper | |
| PCDNA3.1-FLAG-RhoB-T37E | This paper | |
| PCDNA3.1-FLAG-RhoB-T37A | This paper | |
| PCDNA3.1-HA-OXSR1-K46R | This paper | |
| PGEX-6P-1-GST-OXSR1 | This paper | |
| PGEX-6P-1-GST-RhoB | This paper | |
| PGEX-6P-1-GST-OXSR1-K46R | This paper | |
| PGEX-6P-1-GST-RhoB-37A | This paper | |

| Reagent/resource | Reference or source | Identifier or catalog number |
|---|---|---|
| PGEX-6P-1-GST-RhoB-T37E | This paper | |
| PGEX-6P-1-GST-RhoB(1–68) | This paper | |
| PGEX-6P-1-GST-RhoB(69–196) | This paper | |
| PGEX-6P-1-GST-RBD | This paper | |
| PGEX-6P-1-GST-PBD | This paper | |
| PGEX-6P-1-GST-LATS1-HM | This paper | |
| PET28a-His-OXSR1 | This paper | |
| PET28a-His-RhoB | This paper | |
| PET28a-His-ARHGAP17 | This paper | |
| PET28a-His-OXSR1-K46R | This paper | |
| pAAV-FLAG-YAP-5SA | This paper | |
| pAAV-FLAG-RhoB-T37E | This paper | |
| pAAV-FLAG-OXSR1-T185E | This paper | |
| pAAV-FLAG-OXSR1-T185A | This paper | |
| pAAV-Vil1-mir30-shsav1 | This paper | |
| PLENTI-HA-OXSR1 | This paper | |
| PLENTI-HA-OXSR1-K46R | This paper | |
| PLENTI-FLAG-OXSR1 | This paper | |
| **Antibodies** | | |
| Mouse anti-YAP/TAZ | Sigma-Aldrich | Cat#WH0010413M1 |
| Rabbit anti-YAP | Abclonal | Cat#A19134 |
| Rabbit anti-phospho-YAP S127) | CST | Cat#13008 |
| Rabbit anti-phospho-YAP (S381) | CST | Cat#13619 |
| Rabbit anti-LATS1 | CST | Cat#3477 |
| Rabbit anti-Phospho-LATS1 (T1079) | CST | Cat#8654 |
| Mouse anti-GAPDH | Abclonal | Cat#A19056 |
| Rabbit anti-phospho-MST1 (T183)/MST2 (T180) | CST | Cat#49332 |
| Rabbit anti-MST1 | CST | Cat#3682 |
| Rabbit anti-MST2 | CST | Cat#3952 |
| Rabbit anti-RhoB | Abclonal | Cat#A22258 |
| Rabbit anti-RAC1 | Abclonal | Cat#A5080 |
| Rabbit anti-OXSR1 | Abclonal | Cat#A15126 |
| Rabbit anti-SPAK | Abclonal | Cat#A2275 |
| Rabbit anti-phospho-SPAK (Ser373)/phospho-OXSR1 (Ser325) | Sigma-Aldrich | RRID: AB_11205577 |
| Rabbit anti-Caspase-3 | Abclonal | Cat#A19654 |
| Mouse anti-GST | GNI | Cat#GNI4110-GT-S |
| Mouse anti-His | GNI | Cat#GNI4110-HS-S |
| Rabbit mouse anti-β-actin | GNI | Cat#GNI4110-BA-M |
| Rabbit anti-HA | GNI | Cat#GNI4110-HA-B |
| Rat anti-Flag | Biolegend | Cat#637303 |

| Reagent/resource | Reference or source | Identifier or catalog number |
|---|---|---|
| Rabbit anti-thiophosphate ester | Abcam | Cat#ab92570 |
| Rabbit anti-p-p38 | CST | Cat#4511 |
| Rabbit anti-p-jnk | CST | Cat#4668 |
| Rabbit anti-p-ERK1/2 | CST | Cat#4370 |
| Rabbit anti-p38 | CST | Cat#9212 |
| Rabbit anti-jnk | CST | Cat#9252 |
| Rabbit anti-ERK1/2 | CST | Cat#9102 |
| Rabbit anti-NLK | Abclonal | Cat#A19270 |
| Rabbit anti-p-TAZ | CST | Cat#59971 |
| Rouse anti-TAZ | proteintech | Cat#66500-1-lg |
| Rabbit anti-SLMAP | Abcam | Cat#ab243383 |
| Rabbit anti-p-taok2 | R&D Systems | Cat#PPS037 |
| Goat anti-Tao2 | Santa Cruz | Cat#SC-47447 |
| Rabbit anti-DCP-1 | CST | Cat#9578S |
| Goat anti-mLRIG1 | R&D Systems | Cat#AF3688 |
| Rabbit anti-Ki67 | Abcam | Cat#ab16667 |
| **Oligonucleotides and other sequence-based reagents** | | |
| Primers for qRT-PCR | | |
| *Ctgf* forward: 5'CATCTTCGGTGG-TACGGTGT | This paper | |
| *Ctgf* reverse: 5'TTCCAGTCGG-TAAGCCGC | This paper | |
| *Cyr61* forward: 5'CGGGTTTCTTTCA-CAAGGCG | This paper | |
| *Cyr61* reverse: 5'TGAAGCGGCTCCCTGTT-TTT | This paper | |
| *Ankrd1* forward: 5'GCCATGCCTT-CAAAATGCCA | This paper | |
| *Ankrd1* reverse: 5'AGAACTGTGCTGGGAA-GACG | This paper | |
| *Gapdh* forward: 5'GAGTCAACG-GATTTGGTCGT | This paper | |
| *Gapdh* reverse: 5'TTGATTTTGGAGG-GATCTCG | This paper | |
| *Arhgap17* forward: 5'GGTCAATATGCCAC-CATTCC | This paper | |
| *Arhgap17* reverse: 5'AAAGACTTCGTGCTGG-GAGA | This paper | |
| *mLgr5* forward: 5'CCTACTCGAAGACT-TACCCAGT | This paper | |

| Reagent/resource | Reference or source | Identifier or catalog number |
|---|---|---|
| *mLgr5* reverse: 5′GCATTGGGGTGAATGA-TAGCA | This paper | |
| *mCD44* forward: 5′TCGATTTGAATG-TAACCTGCCG | This paper | |
| *mCD44* reverse: 5′CAGTCCGGGAGATACTG-TAGC | This paper | |
| *mEphb2* forward: 5′GCGGCTACGACGAGAA-CAT | This paper | |
| *mEphb2* reverse: 5′GGCTAAGTCAAAAT-CAGCCTCA | This paper | |
| *mActinb* forward: 5′GGCTGTATTCCCCTC-CATCG | This paper | |
| *mActinb* reverse: 5′CCAGTTGGTAACAATGC-CATGT | This paper | |
| *mSav1* forward: 5′TGGCTGGGAACGAGTA-GAG | This paper | |
| *mSav1* reverse: 5′AGCATTCCCTGG-TACGTGTC | This paper | |
| gRNA sequences | | |
| sg*MST1*: 5′CACCGGGATCGTTATG-GAGTACTGT 5′AAACACAGTACTCCA-TAACGATCC | This paper | |
| sg*MST2*: 5′CACCGCGATGTTG-GAATCCGACTTG 5′AAACCAAGTCGGATTC-CAACATCG | This paper | |
| sg*Lats1*: 5′CACCGGATTTCATGCC-CACTGCTCG 5′AAACCGAGCAGTGGG-CATGAAATC | This paper | |
| sg*Lats2*: 5′CACCGGCCCCCATCTA-CACGTACAC 5′AAACGTGTACGTGTA-GATGGGGGC | This paper | |
| sg*OXSR1*#1: 5′CACCGCTCG-TAATCGTCCCTGTTGA 5′AAACTCAACAGGGAC-GATTACGAG | This paper | |
| sg*OXSR1*#2: 5′CACCGCGAGTCCTCGGA-CATGACGG 5′AAACCCGTCATGTCC-GAGGACTCG | This paper | |
| shRNA sequences | | |

| Reagent/resource | Reference or source | Identifier or catalog number |
|---|---|---|
| sh*NLK*#1: 5′CCGGTGGGCAACAA-CAGCCATATTTCTCGA-GAAA-TATGGCTGTTGTTGCC-CATTTTTG 5′AATTCAAAAATGGGCAA-CAACAGCCATATTTCTC-GAGAAA-TATGGCTGTTGTTGCCCA | This paper | |
| sh*NLK*#2: 5′CCGGGAAGTTGTTACT-CAGTATTATCTCGAGA-TAATACTGAGTAA-CAACTTC TTTTTG 5′AATTCAAAAA-GAAGTTGTTACTCAGTAT-TATCTCGAGATAATACT-GAGTAACAACTTC | This paper | |
| sh*ARHGAP17*#1: 5′CCGGGGATGGAT-GAAGCTGGAAATAACTC-GAGTTATTTCCAGCTT-CATCCATCTTTTTG 5′AATTCAAAAAGATGGAT-GAAGCTGGAAATAACTC-GAGTTATTTCCAGCTT-CATCCATC | This paper | |
| sh*ARHGAP17*#2: 5′CCGGTGCTGTAG-CAGGTGCTTTAAACTC-GAGTTTAAAGCACCTGC-TACAGCATTTTTG 5′AATTCAAAAATGCTG-TAGCAGGTGCTT-TAAACTCGAGTTTAAAG-CACCTGCTACAGCA | This paper | |
| **Chemicals, enzymes, and other reagents** | | |
| Jasplakinolide | Abcam | Cat#ab141409 |
| Latrunculin B | Abcam | Cat#ab144291 |
| C3 transferase | Cytoskeleton | Cat#CT03-A |
| Staurosporine | MCE | Cat#HY-15141 |
| Phalloidin | MCE | Cat#HY-P0028 |
| Bumetanide | MCE | Cat#HY-17468 |
| Chlorothiazide | MCE | Cat#HY-B0224 |
| CLP-290 | MCE | Cat#HY-103023 |
| WNK463 | MCE | Cat#HY-100626 |
| Rafoxanide | MCE | Cat#HY-17598 |
| Brilliant Blue FCF | Millipore Sigma | Cat#3844-45-9 |
| Dextran sulfate sodium (DSS) (MW 36,000–50,000) | Yeasen | Cat#60316ES60 |
| AOM | Sigma-Aldrich | Cat#A5486 |
| DAPI | Beyotime | Cat#C1002 |
| Hematoxylin-Eosin Staining Kit | Beyotime | Cat#C0105S |
| Phalloidin-iFluor 488 | Abcam | Cat #Ab176753 |

| Reagent/resource | Reference or source | Identifier or catalog number |
|---|---|---|
| **Software** | | |
| BioGRID 5.0 | https://thebiogrid.org/ Oughtred et al, 2021 | |
| Prism 9.0.1 | https://www.graphpad.com/ | |
| Bio-Rad CFX manager | https://www.bioradiations.com/new-and-improved-cfx-manager-software-for-real-time-pcr-data-acquisition-and-analysis/ | |
| Fiji software 1.53t | https://imagej.net/software/fiji/ | |
| Microscope software ZEN (blue edition) | https://www.zeiss.com/microscopy/us/products/software/zeiss-zen-lite.html | |
| **Other** | | |
| Illumina NovaSeq 6000 | Illumina | |

## Cell culture

Human HeLa cells (ATCC CCL-2), HEK293T (ATCC CRL-11268), HEK293A (ATCC CRL-1573), HEK293FT (ThermoFisher R70007), and SW480 (ATCC CCL-228) cells were cultured in Dulbecco's Modified Eagle Medium (DMEM; BBI) supplemented with 10% fetal bovine serum (FBS) and penicillin/streptomycin. All cells were maintained at 37 °C in a humidified incubator with 5% $CO_2$. Mst1/2; MAP4K1/2/3/4/5/6/7 knockout (MM-9KO) HEK293A cells have been described previously (Feng et al, 2016), and were kindly provided by Drs. Kun-Liang Guan (Westlake University) and Fa-Xing Yu (Fudan University).

## Drosophila genetics

Stocks were raised on standard cornmeal-agar medium. Three- to five-day-old adults were used in this study. $w^{1118}$ flies were used as a standard wild-type strain. UAS-Fray (stock ID 99477), UAS-Yki (stock ID 28819), and UAS-Yki-3SA (stock ID 28817) were obtained from Bloomington Drosophila Stock Center (BDSC). fray RNAi stocks were obtained from Bloomington Drosophila Stock Center (stock ID 41587, 42569) and Vienna Drosophila Resource Center (stock ID 106919). escargot-Gal4; UAS-mCD8GFP (esg > GFP) driver/reporter line has been described previously (Micchelli and Perrimon, 2006), and was kindly gifted by Dr. Xianjue Ma (Westlake University).

## Mice

Unless otherwise specified, all mice used in this study were bred on a C57BL/6J background and maintained under standard conditions with a 12-h light/dark cycle at the Xiamen University Laboratory Animal Center (XMULAC). All animal experiments were approved by the Animal Ethics Committee of Xiamen University (acceptance number: XMULAC20230019) and followed the animal welfare guidelines established by XMULAC. $Oxsr1^{fl/fl}$ and $Villin-Cre^{ERT2}$ mice have been described previously (el Marjou et al, 2004), and were purchased from Cyagen Biosciences. To generate Oxsr1 conditional knockout mice, $Villin-Cre^{ERT2}$; $Oxsr1^{fl/fl}$ mice (8–12 weeks old) were intraperitoneal injected with Tamoxifen (120 mg/kg; dissolved in corn oil; MCE #HY-13757A) for 2 weeks.

## Plasmids, antibodies, and chemicals

Plasmids including Flag-OXSR1, HA-MAP4K2, Flag-RhoB, HA-ARHGAP17, and Flag-Rac1 were generated from cDNA clones collected by the School of Life Sciences, Xiamen University. AAV-Vil1-mir30-shsav1 was purchased from PackGene. Flag-YAP, Flag-MST2, Myc-LATS1 constructs have been described previously (Yang et al, 2024). All other related constructs of point mutation, truncation, or different epitope tag were generated from the constructs described above.

The following primary antibodies were used in this study:

For western blot: Mouse anti-YAP/TAZ (Sigma-Aldrich #WH0010413M1, 1:1000), rabbit anti-YAP (Abclonal #A19134, 1:1000), rabbit anti-phospho-YAP (S127) (CST #13008, 1:1000), rabbit anti-phospho-YAP (S381) (CST #13619, 1:1000), rabbit anti-LATS1 (CST #3477, 1:1000), rabbit anti-Phospho-LATS1 (T1079) (CST #8654, 1:1000), mouse anti-GAPDH (Abclonal #A19056, 1:1000), rabbit anti-phospho-MST1 (T183)/MST2 (T180) (CST #49332, 1:1000), rabbit anti-MST1 (CST #3682, 1:1000), rabbit anti-MST2 (CST #3952, 1:1000), rabbit anti-RhoB (Abclonal #A22258, 1:1000), rabbit anti-RAC1 (Abclonal #A5080, 1:1000), rabbit anti-OXSR1 (Abclonal #A15126, 1:1000), rabbit anti-SPAK (Abclonal #A2275, 1:1000), rabbit anti-phospho-SPAK (Ser373)/phospho-OXSR1 (Ser325) (Sigma-Aldrich #AB_11205577, 1:1000), rabbit anti-caspase-3 (Abclonal #A19654, 1:1000), mouse anti-GST (GNI #GNI4110-GT-S, 1:1000), mouse anti-His (GNI #GNI4110-HS-S, 1:1000), rabbit mouse anti-β-actin (GNI #GNI4110-BA-M, 1:10,000), rabbit anti-HA (GNI #GNI4110-HA-B, 1:1000), rat anti-Flag (Biolegend #637303, 1:1000), rabbit anti-thiophosphate ester (Abcam #ab92570, 1:1000), rabbit anti-p-p38 (CST #4511, 1:1000), rabbit anti-p-jnk (CST #4668, 1:1000), rabbit anti-p-ERK1/2 (CST #4370, 1:1000), rabbit anti-p38 (CST #9212, 1:1000), rabbit anti-jnk (CST #9252, 1:1000), rabbit anti-ERK1/2 (CST #9102, 1:1000), rabbit anti-NLK (Abclonal #A19270, 1:1000), rabbit anti-p-TAZ (CST #59971, 1:1000), mouse anti-TAZ (proteintech #66500-1-lg, 1:1000), rabbit anti-SLMAP (abcam #ab243383, 1:1000), rabbit anti-p-taok2 (R&D Systems #PPS037, 1:1000), goat anti-TAOK2 (Santa Cruz #SC-47447, 1:1000).

For immunostaining: Rabbit anti-phospho-SPAK (Ser373)/phospho-OXSR1 (Ser325) (Sigma-Aldrich #AB_11205577, 1:200), rabbit anti-YAP (Abclonal #A19134, 1:100), rat anti-FLAG (Biolegend #637303, 1:250), rabbit anti-HA (GNI #GNI4110-HA-B, 1:250), rabbit anti-DCP-1 (CST #9578S, 1:100), anti-mLRIG1 (R&D Systems #AF3688, 10 μg/mL).

For immunohistochemistry (IHC): Rabbit anti-YAP (Abclonal #A19134, 1:100), rabbit anti-phospho-SPAK (Ser373)/phospho-OXSR1 (Ser325) (Sigma-Aldrich #AB_11205577, 1:200), rabbit anti-Ki67 (Abcam #ab16667, 1:200).

The following chemicals were used in this study:

Jasplakinolide (Abcam #ab141409), Latrunculin B (Abcam #ab144291), C3 transferase (Cytoskeleton #CT03-A), Staurosporine (MCE #HY-15141), Phalloidin (MCE #HY-P0028), Bumetanide (MCE #HY-17468), Chlorothiazide (MCE #HY-B0224), CLP-290 (MCE #HY-103023), WNK463 (MCE #HY-100626), and Rafoxanide (MCE #HY-17598). Brilliant Blue FCF (Millipore Sigma #3844-45-9), Dextran sulfate sodium (DSS) (Yeasen #60316ES60, MW 36,000–50,000), AOM (Sigma-Aldrich #A5486).

## Human tissue specimens

Paraffin-embedded specimens from patients with IBD were obtained via colonoscopy biopsy at the 909th Hospital of Xiamen University. Sample collection was approved by the hospital's ethical committee and institutional review board (acceptance number: 2018-009-01). Written informed consent was obtained from each patient, and all patient data were anonymized. Detailed information of patients is as follows: patient 1 (age 44, male); patient 2 (age 31, female); patient 3 (age 38, male).

## Drosophila DSS feeding and survival experiment

Sex- and age-matched adult flies (3–5 days old) were placed in vials (20–25 flies per vial) containing a piece of Whatman filter paper soaked with 300 μL of 5% sucrose solution containing 5% DSS (Yeasen). Flies were maintained at 29 °C, and the DSS solution was replaced daily to ensure continuous exposure. Fly mortality was recorded every 24 h for up to 10 days, with dead flies removed at each timepoint.

## Smurf assay

Flies were fed with 3% DSS (dissolved in 5% sucrose) for 5 days before being transferred to new culture vials containing 2.5% Brilliant Blue FCF diluted in 5% sucrose for 1 day. Individuals showing blue coloration restricted to the gut were scored as Smurf-negative, while those with widespread blue coloration outside the gut were designated as Smurf-positive.

## Cell transfection, western blot, immunoprecipitation, immunostaining, GST pull-down, in vitro kinase assay and quantitative real-time PCR (RT-qPCR)

Cells were transfected with Effectene transfection reagent (Qiagen) or Lipofectamine 3000 (ThermoFisher) following the manufacturer's recommendations. Western blot, immunoprecipitation, immunostaining, GST pull-down, in vitro kinase assay, and RT-qPCR were performed following standard protocols as described (Shen et al, 2022; Yang et al, 2024). RT-qPCR primers used in this study are as follows:

*Ctgf:*
5'-CATCTTCGGTGGTACGGTGT-3' and 5'-TTCCAGTCGGT AAGCCGC-3';
*Cyr61:*
5'- CGGGTTTCTTTCACAAGGCG-3' and 5'-TGAAGCGGCT CCCTGTTTTT-3';
*Ankrd1:*
5'-GCCATGCCTTCAAAATGCCA-3' and 5'-AGAACTGTGC TGGGAAGACG-3';

*Gapdh:*
5'-GAGTCAACGGATTTGGTCGT-3' and 5'-TTGATTTTGGA GGGATCTCG-3'.
*mLgr5:*
5'-CCTACTCGAAGACTTACCCAGT-3' and 5'-GCATTGGG GTGAATGATAGCA-3'
*mCD44:*
5'-TCGATTTGAATGTAACCTGCCG-3' and 5'-CAGTCCGGG AGATACTGTAGC-3'
*mEphb2:*
5'-GCGGCTACGACGAGAACAT-3' and 5'-GGCTAAGTCAA AATCAGCCTCA-3'
*mActinb:*
5'-GGCTGTATTCCCCTCCATCG-3' and 5'-CCAGTTGGTAA CAATGCCATGT-3'
*ARHGAP17:*
5'-GGTCAATATGCCACCATTCC-3' and 5'-AAAGACTTCGT GCTGGGAGA-3'
*mSav1:*
5'-CTGGCTGGGAACGAGTAGAG-3' and 5'-AGCATTCCCT GGTACGTGTC-3'.

## RNA sequencing

RNA was extracted using Trizol Reagent (Invitrogen) and purified using RNeasy Mini Kit (Qiagen) following the kit manual. Library construction, sequencing, and bioinformatics analysis were performed by MegaGenomics Inc. (Beijing, China). Raw sequencing data in FASTQ format were first processed using fastp (v0.20.1) to remove adapters and low-quality reads. Clean reads were then aligned to the human reference genome (Homo sapiens, Ensembl v96, hg38) using HISAT2 (v2.0.4). The resulting BAM files were assembled into transcripts using StringTie (v1.3.4 d) with the reference GTF file. Transcript annotation comparison was performed using gffcompare (v0.9.8). Transcript abundance was then estimated using StringTie. PKM values were extracted using the Ballgown package. Differential expression gene (DEGs) analysis was conducted using DESeq2 (v1.26.0). DEGs were defined as adjusted $P$ value < 0.05 and $|\log2(\text{FoldChange})| > 0.5$. Three biological replicates were performed for WT and *Oxsr1* KO HEK293T cells.

## Lentivirus packaging and knockout/knockdown cell line generation

Generation of knockout cell lines via plasmid transfection was described previously (Yang et al, 2024). To generate knockout cell lines via lentiviral transduction, gRNAs were subcloned into the LentiCRISPRv2 vector (Addgene #52961) before being transfected into HEK293T cells along with the lentiviral packaging plasmids, psPAX2 (Addgene #12260) and pMD2.G (Addgene #12259). After 48–72 h, the cell culture medium containing the lentiviruses was collected and filtered through a 0.45-μm syringe filter. Target cells were then infected with the lentiviruses in the presence of 8 μg/mL polybrene (Sigma-Aldrich). Puromycin (2 μg/mL) was used to select transduced cells for 3–7 days. Single-cell clones were obtained by limiting dilution or FACS sorting and expanded for validation via western blot and Sanger sequencing.

The gRNA sequences used in this study are listed below.

*Mst1:*
5'-CACCGGGATCGTTATGGAGTACTGT-3'
5'-AAACACAGTACTCCATAACGATCC-3'

*Mst2:*
5'-CACCGCGATGTTGGAATCCGACTTG-3'
5'-AAACCAAGTCGGATTCCAACATCG-3'

*Lats1:*
5'-CACCGGATTTCATGCCCACTGCTCG-3'
5'-AAACCGAGCAGTGGGCATGAAATC-3'

*Lats2:*
5'-CACCGGCCCCCATCTACACGTACAC-3'
5'-AAACGTGTACGTGTAGATGGGGGC-3'

*Oxsr1 #1:*
5'-CACCGCTCGTAATCGTCCCTGTTGA-3'
5'-AAACTCAACAGGGACGATTACGAG-3'

*Oxsr1 #2:*
5'-CACCGCGAGTCCTCGGACATGACGG-3'
5'-AAACCCGTCATGTCCGAGGACTCG-3'.

To generate NLK or ARHGAP17 knock down HEK293T cells, shRNAs were cloned into the PLKO.1 plasmid. Virus was produced by co-transfecting HEK293T cells with the shRNA-expressing PLKO.1 plasmid, psPAX2 (Addgene #12260), and pMD2.G (Addgene #12259) using Lipofectamine 3000 (Thermo-Fisher). Viral supernatants were collected 48–72 h post-transfection, filtered through a 0.45 μm syringe filter, and used to infect target cells in the presence of 8 μg/mL polybrene (Sigma-Aldrich). Puromycin (1–2 μg/mL) was used to select transduced cells.

shRNA sequences used in this study are listed below:

*Nlk #1:*
5'-CCGGTGGGCAACAACAGCCATATTTCTCGAGAAA-TATGGCTGTTGTTGCCCATTTTTG-3'
5'-AATTCAAAAATGGGCAACAACAGCCATATTTCTCGA-GAAATATGGCTGTTGTTGCCCA-3'

*Nlk #2:*
5'-CCGGGAAGTTGTTACTCAGTATTATCTCGAGATAA-TACTGAGTAACAACTTC TTTTTG-3'
5'-AATTCAAAAGAAGTTGTTACTCAGTATTATCTCGA-GATAATACTGAGTAACAACTTC-3'

*Arhgap17 #1:*
5'-CCGGGATGGATGAAGCTGGAAATAACTCGAGT-TATTTCCAGCTTCATCCATCTTTTTG-3'
5'-AATTCAAAAAGATGGATGAAGCTGGAAATAACTC-GAGTTATTTCCAGCTTCATCCATC-3'

*Arhgap17 #2:*
5'-CCGGTGCTGTAGCAGGTGCTTTAAACTCGAGTT-TAAAGCACCTGCTACAGCATTTTTG-3'
5'-AATTCAAAAATGCTGTAGCAGGTGCTTTAAACTC-GAGTTTAAAGCACCTGCTACAGCA-3'.

## AAV production and delivery

FLAG-YAP-5SA, FLAG-RhoB-T37E, FLAG-OXSR1-T185A, and FLAG-OXSR1-T185E were cloned into pAAV vector. Recombinant AAV2/9 viruses were produced as previously described (Grieger et al, 2006) and purified via Iodixanol (MCE #HY-K3015) gradient ultracentrifugation. Each mouse received an intraperitoneal injection of $1 \times 10^{13}$ vg AAV, followed by a second injection of the same dose one week later. Mice were analyzed two weeks after the final injection.

## Osmolarity measurements

Gut osmolarity was measured as previously described (Tropini et al, 2018). Briefly, colonic contents were collected after euthanizing the mice and homogenized in a tube containing a nylon mesh with a 22-micron aperture. The mixture was then centrifuged at $16,000 \times g$ for 40 min at 4 °C. The osmolality of the colonic supernatant was measured using an Vapro 5600 Dew Point Osmometer following the manufacturer's instructions (Wescor).

## Experimental colitis

Colitis was induced by administering 2.5% (w/v) DSS in drinking water for 5 days (injury phase), followed by regular water for 9 additional days (regeneration phase). Pharmacological inhibition of OXSR1 or WNK kinase activity was achieved through oral administration of Rafoxanide (0.2 mg/mL) or WNK463 (0.04 mg/mL), respectively, dissolved in drinking water. Drug treatments began immediately after DSS withdrawal and continued until tissue collection. Control animals received equivalent concentrations of DMSO in drinking water.

## AOM/DSS mouse tumor model

Mice were intraperitoneally injected with 12 mg/kg of body weight AOM resuspended at 20 mg/mL in water, which was further diluted to 1 mg/mL in PBS before being injected into mice. Administration of DMSO, Rafoxanide (0.05 mg/mL dissolved in drinking water) or WNK463 (0.02 mg/ml dissolved in drinking water) was started on the same day and continued for the entire duration of the experiment. 1 week later, 2.5% (w/v) DSS was administered in the drinking water for 6 days followed by 2 weeks of regular water. The same DSS treatment was repeated for two more cycles (three cycles in total), each time with an interval of 2 weeks of regular water in between. Mice were euthanized at the indicated timepoints, and colon samples were collected to evaluate the tumor number.

## Crypt isolation

Crypts were isolated as previously described (Cai et al, 2010). Briefly, colons were harvested, flushed with cold PBS, and cut longitudinally into 2–3 mm pieces, then washed with cold PBS. Tissue fragments were incubated in 4 mM EDTA in PBS for 20 min at 4 °C. After removing the EDTA solution, 3–5 mL of cold PBS was added, and vigorous shaking was used to release the crypts. For western blot analysis, the supernatant containing free crypts was collected and centrifuged at $1000 \times g$ for 3 min. The pellet was washed three times with ice-cold PBS and snap-frozen in liquid nitrogen to break the crypts. For RhoB-GTP enrichment assays, the pellet was resuspended in RIPA lysis buffer (Beyotime #P0013D) without snap-frozen in liquid nitrogen, incubated at 4 °C for 20 min, and then centrifuged at 15,000 g for 15 min. The supernatant was collected for GST-RBD pulldown assays.

## Histology, immunohistochemistry, and immunofluorescence staining on paraffin-embedded mouse tissues

Intestinal tissues were freshly harvested, fixed overnight in 4% paraformaldehyde at 4 °C, and washed with PBS. For cryo-embedding, tissues were then processed with 15% sucrose and 30% sucrose until tissues sank prior to freezing in O.C.T compound (Sakura). 10 μm sections were prepared by cryotome (Leica CM3050S). For paraffin embedding, tissues were dehydrated through a graded ethanol series, cleared in xylene, and embedded in paraffin. Sections (5 μm thick) were prepared for hematoxylin and eosin (H&E) staining, immunohistochemistry (IHC), and immunofluorescence staining (IF). H&E staining was performed using the Hematoxylin-Eosin Staining Kit (Beyotime #C0105S) following the standard protocol. For IHC, sections were deparaffinized in xylene, rehydrated through graded alcohols, and incubated with 3% $H_2O_2$ in PBS for 15 min to quench endogenous peroxidase activity. Antigen retrieval was performed by boiling in 1 mM EDTA (pH 9.0) or 10 mM sodium citrate (pH 6.0) buffer for 20 min. After PBS washes, sections were blocked with 5% goat serum for 2 h at room temperature, followed by overnight incubation at 4 °C with primary antibodies. After washing, sections were incubated with HRP-conjugated secondary antibodies for 2 h at room temperature. Signal was developed using a DAB detection kit (Vazyme #HC301-01), and nuclei were counterstained with hematoxylin (Beyotime #C0105S). Slides were dehydrated, cleared in xylene, mounted in permanent mounting medium, and imaged using a Zeiss Observer7 or Leica M165FC microscope. For IF, sections were blocked with 5% goat serum, incubated overnight at 4 °C with primary antibodies, and followed by incubation with FITC- or Cy3-conjugated secondary antibodies. After PBS washes, nuclei were counterstained with DAPI (2 μg/mL; Beyotime #C1002). Slides were mounted using anti-fade mounting medium (BBI #E675011-0010) and visualized on a Zeiss LSM980 confocal microscope.

## In vitro kinase assay followed by in vitro binding assay

HEK293T cells were transfected with HA-ARHGAP17 plasmid. 48 h later, cells were lysed with lysis buffer (50 mM Tris, pH 7.4, 150 mM NaCl, 1 mM EDTA, 0.5% Triton X-100, protease inhibitor cocktail, 50 mM NaF, 1.5 mM $Na_3VO_4$). Immunoprecipitation was performed using anti-HA agarose beads (Biolegend). Recombinant His-OXSR1/His-OXSR1-K46R and GST-RhoB were purified using Ni-NTA resin and GST beads, respectively. GST-RhoB was eluted using glutathione and subsequently incubated with Ni-NTA resin-bound His-OXSR1/His-OXSR1-K46R in kinase buffer (25 mM HEPES, pH 7.2, 25 mM $MgCl_2$, 50 mM β-glycerol phosphate, 2 mM dithiothreitol, 0.5 mM sodium vanadate, and 5 mM ATP) at 37 °C for 1.5 h. The reaction mixture was centrifuged, and the supernatant was diluted with PBS, then incubated with HA-ARHGAP17-coupled agarose beads for 2 h at 4 °C. After three washes, the beads were boiled in 2×SDS loading buffer for 5 min. The interaction between RhoB and ARHGAP17 was assessed by western blot.

## Actin segmentation by ultracentrifugation

Actin segmentation by ultracentrifugation was performed as previously described (Qiao et al, 2017). Briefly, cells were plated sparsely, treated with phalloidin (250 ng/mL) for 1 h, and lysed in

actin stabilization buffer. Lysates were collected by scraping and centrifuged at 750 g for 5 min to remove insoluble particles. Protein concentration was determined by Bradford assay, and lysates were equalized before fractionation. F-actin and G-actin were separated by ultracentrifugation at $100,000 \times g$ for 1 h at 37 °C. The supernatant (G-actin) was collected, and the F-actin pellet was resuspended in cold distilled water with 2 mM cytochalasin D for 1.5 h on ice. Both fractions were mixed with 2×SDS loading buffer, boiled, and analyzed by western blot.

## CCK-8 cell viability assay

HEK293T cells were seeded in 96-well plates at a density of 2000 cells per well and cultured overnight before being analyzed using the CCK-8 kit following the manufacturer's recommendation (Abclonal #RM02823). The optical density at 450 nm (OD450) was measured with a microplate reader (TECAN).

## Colony formation assay

Colony formation assay was performed as previously described (Franken et al, 2006). Briefly, cells were seeded at a density of 1000 cells per 100 mm dish and cultured for 14 days. The cells were then washed with phosphate-buffered saline (PBS), fixed with 4% paraformaldehyde for 10 min, and stained with a 0.5% crystal violet for 2 min. The colony area per dish was analyzed with ImageJ software and was divided by the dish area to obtain the area percentage.

## Wound-healing assays

Cells were cultured in 60 mm dishes until they formed confluent monolayers ( >90% confluence), then scratched across the surface using a sterile pipette tip. The dishes were gently washed with PBS to remove cellular debris and subsequently maintained under standard culture conditions. Wound area was calculated using the ImageJ software.

## Quantification and statistical analysis

All statistical analyses were done using GraphPad Prism 9.0.1 (GraphPad Software). No statistical method was used to pre-determine sample size. No blinding was performed because none of the analyses reported involved procedures that could be influenced by investigator bias. Sample allocation was random in all experiments. Two-tailed unpaired Student's *t* test, one-way or two-way ANOVA with appropriate post-hoc tests were used for statistical analysis of data. Survival curves were plotted and analyzed by log-rank test. Detailed statistical analysis methods, sample size, or the number of biological replicates is indicated in the figure legends. The *P* values were labeled in the figures. Exact *P* values ranging from 0.9999 to 0.0001 are reported as precise numbers, while those less than 0.0001 are indicated as <0.0001. The *P* values lower than 0.05 were considered statistically significant.

# Data availability

RNA-Seq data have been deposited at NCBI's Gene Expression Omnibus (GEO) database (accession number: GSE296724) and are

publicly available as of the date of publication. Source data have been deposited at BioStudies database (accession number: S-BSST2346).

The source data of this paper are collected in the following database record: biostudies:S-SCDT-10_1038-S44318-026-00738-8.

## Peer review information

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

## Acknowledgements

We thank Drs. Kun-Liang Guan (Westlake University) and Fa-Xing Yu (Fudan University) for the *MM*-9KO HEK293A cells, Dr. Xianjue Ma (Westlake University) for the *esg > GFP* flies, Drs. Qiao Wu and Hangzi Chen (Xiamen University) for the AAV2/9 vector, and Dr. Shuyong Lin (Xiamen University) for the MAPKs-related antibodies. We thank Dr. Xuan Guo (Jinzhou Medical University) for the help with fly import and Wei Han (Xiamen University) for the help with fly maintenance. We thank Bloomington Drosophila Stock Center (BDSC), Vienna Drosophila Resource Center (VDRC), Drosophila Genomics Resource Center (DGRC), and Developmental Studies Hybridoma Bank (DSHB) for fly strains and reagents. This study was supported in part by grants from Natural Science Foundation of Fujian Province (2024J02002 and 2024J08010), National Natural Science Foundation of China (32470750, 32170873, and 32400715), Natural Science Foundation of Xiamen City (3502Z202371005), and Fundamental Research Funds for the Central Universities (2072024).

## Author contributions

**Heming Cao**: Resources; Data curation; Formal analysis; Validation; Investigation; Visualization; Methodology; Writing—original draft; Project administration; Writing—review and editing. **Xiawei Huang**: Data curation; Software; Formal analysis; Supervision; Funding acquisition; Validation; Investigation; Visualization; Methodology; Writing—original draft; Project administration; Writing—review and editing. **Xiaobing Jiang**: Data curation; Formal analysis; Validation; Investigation; Visualization; Methodology. **Jingrong Deng**: Data curation; Formal analysis; Validation; Investigation; Visualization; Methodology. **Jiahui Wang**: Data curation; Formal analysis; Validation; Investigation; Visualization; Methodology. **Chengfang Wu**: Data curation; Formal analysis; Validation; Investigation; Visualization; Methodology. **Minhuang Hu**: Data curation; Formal analysis; Validation; Investigation; Visualization; Methodology. **Bei Zeng**: Data curation; Formal analysis; Validation; Investigation; Visualization; Methodology. **Zhihao Hu**: Data curation; Formal analysis; Validation; Investigation; Visualization; Methodology. **Huimin Pan**: Data curation; Formal analysis; Validation; Investigation; Visualization; Methodology. **Yuxia Yang**: Data curation; Formal analysis; Validation; Investigation; Visualization; Methodology. **Kewei Zheng**: Data curation; Formal analysis; Validation; Investigation; Visualization; Methodology. **Rui Shen**: Data curation; Formal analysis; Validation; Investigation; Visualization; Methodology. **Mingqing Zhang**: Resources; Methodology. **Bo Liu**: Conceptualization; Resources; Data curation; Formal analysis; Supervision; Funding acquisition; Validation; Investigation; Visualization; Methodology; Writing—original draft; Project administration; Writing—review and editing.

Source data underlying figure panels in this paper may have individual authorship assigned. Where available, figure panel/source data authorship is listed in the following database record: biostudies:S-SCDT-10_1038-S44318-026-00738-8.

## Disclosure and competing interests statement

The authors declare no competing interests.

# Expanded View Figures

**Figure EV1.  Wnk–Fray axis regulates intestinal regeneration in *Drosophila*.**

(A, B) Representative images and quantifications of adult wings (A) and eyes (B) from flies expressing Fray or *fray* RNAi under the indicated Gal4 drivers. Data were analyzed using one-way ANOVA followed by Dunnett's multiple comparisons test and are presented as mean ± s.d.; in (A), scale bar: 500 μm; $n = 9$ flies per group; in (B), scale bars: 100 μm; $n = 10$ flies per group. (C, D) Survival analysis of virgin female (C) and male (D) flies expressing a *fray* RNAi line (BDSC #42569, hereafter referred to as *fray* RNAi #2) under control of the *esg*-Gal4 driver. Flies were exposed to 5% DSS, and survival was recorded daily. Survival curves were analyzed using the log-rank (Mantel–Cox) test and are presented as mean ± s.e.m. In (C), $n = 32$ flies for the *esg* > GFP group and $n = 52$ flies for the *esg* > GFP; *fray*.RNAi group, pooled from three independent experiments. In (D), $n = 37$ flies for the *esg* > GFP group and $n = 63$ flies for the *esg* > GFP; *fray*.RNAi group, pooled from three independent experiments. (E) Smurf assays were performed using *fray* RNAi #2 line driven by the *esg*-Gal4 driver. Flies were treated with 3% DSS for 5 days and then fed 0.5% Brilliant Blue in 5% sucrose for 12 h before assessment. Quantification of Smurf-positive flies is shown. Data were analyzed using a two-tailed Student's *t* test and are presented as mean ± s.d.; $n = 3$ independent experiments (10 flies per experiment). (F) Wild-type flies were fed with DMSO, Rafoxanide (Rafo), or WNK463 and concurrently treated with 3% DSS for 2 days to induce gut injury. ISC/EB cells in the midgut were visualized using the *esg* > GFP reporter (green). Cell nuclei were counterstained with DAPI (blue). Gut boundary was marked by white dashed curves. Note that treatment with Rafo or WNK463 suppressed the DSS-induced increase in ISC/EB cell numbers. Data shown are representative images of the anterior (R2) and posterior (R4) regions of the midgut. UI, uninjured. Scale bar: 50 μm. (G) Wild-type flies were treated as described in (F), but without Rafo treatment. Midguts were dissected on day 2 of the regeneration phase, stained with anti-DCP-1 antibody (red) and counterstained with DAPI (blue). White dashed curves mark the gut boundary. Scale bar: 50 μm. Data shown are representative of at least three independent experiments.

▶

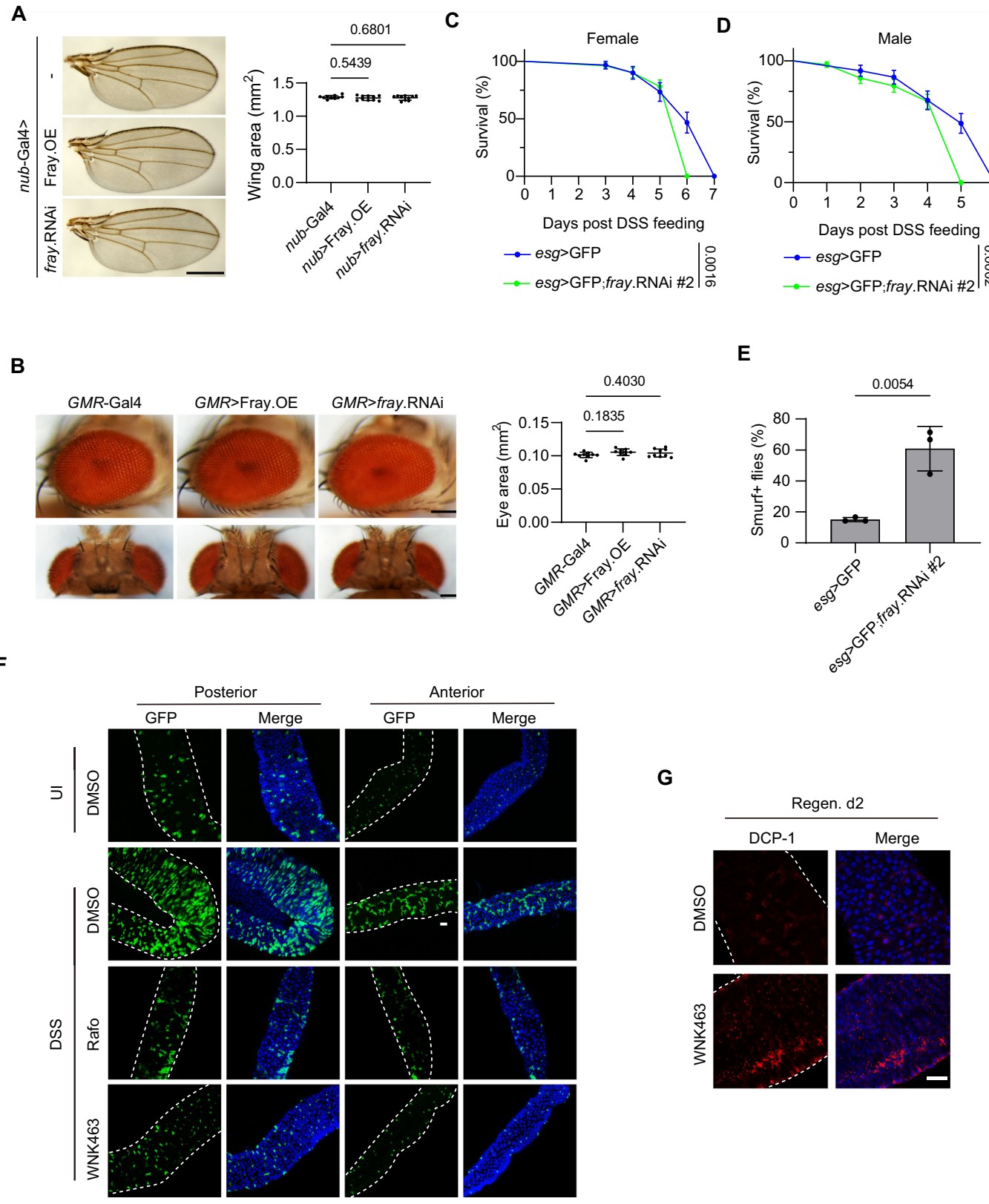

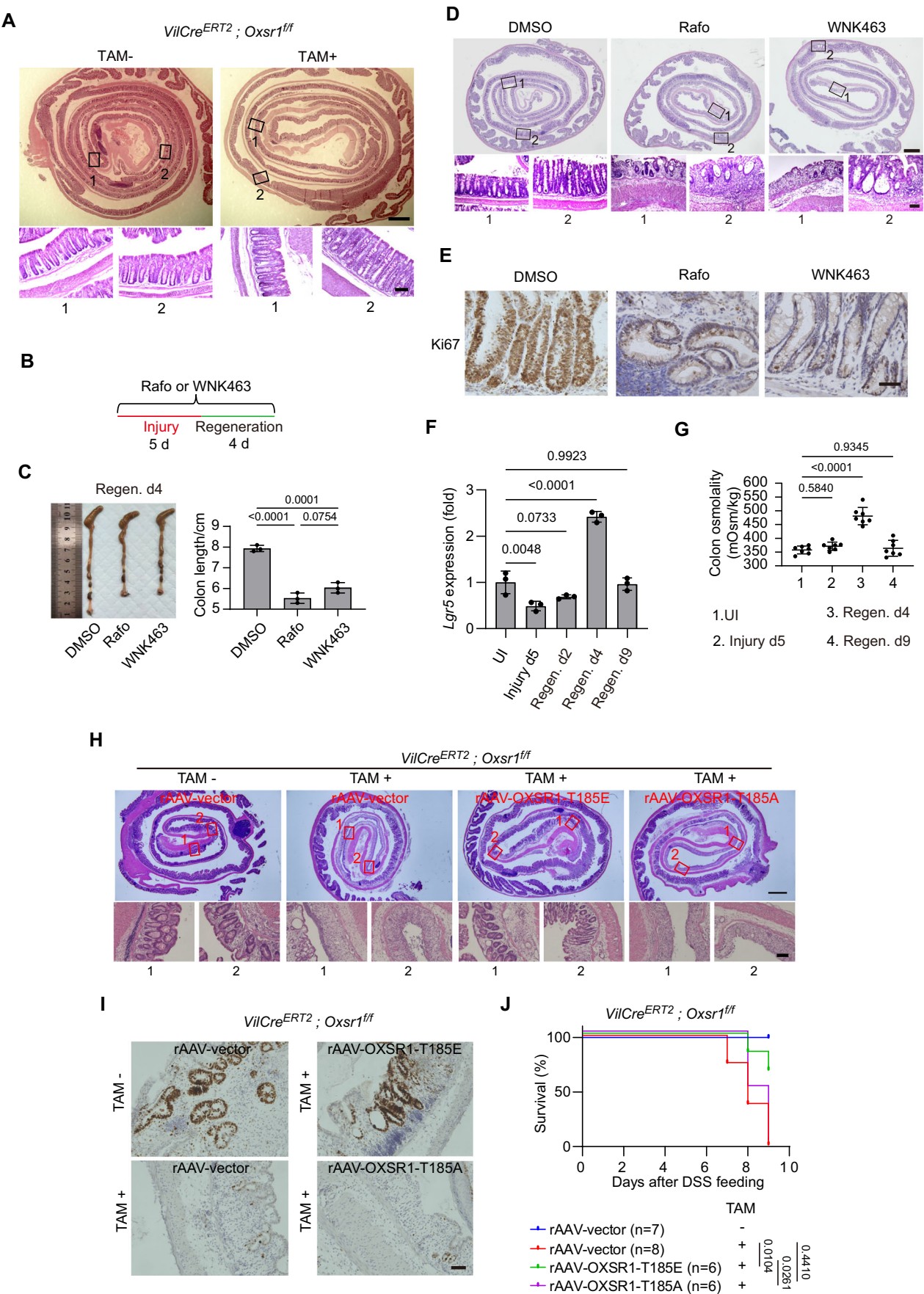

**Figure EV2. WNK-OXSR1 axis regulates intestinal regeneration in mammals.**

(A) Colon tissues were collected from mice with or without tamoxifen treatment for 5 consecutive days. Representative hematoxylin and eosin (H&E) staining images of colon Swiss rolls are shown from at least three independent experiments. Note that conditional knockout of *Oxsr1* in the intestinal epithelium did not result in any discernible abnormalities under steady-state conditions. Scale bars: 500 μm (upper panel) and 100 μm (lower panel). (B) Schematic representation of the chemical treatment regimen in mice. (C–E) Colons from wild-type mice with the indicated chemical treatment were collected on day 4 of the regeneration phase and subjected to similar analyses as in Fig. 2D–F. For (C), data were analyzed using two-tailed Student's *t* test and are presented as mean ± s.d. (*n* = 3 colons). Scale bars in (D): 500 μm (upper panel) and 100 μm (lower panel). Scale bar in (E): 50 μm. (F) Colonic crypts were collected at the indicated stages of injury and regeneration and *Lgr5* mRNA levels were assessed by RT-qPCR. Note that *Lgr5* mRNA levels were reduced during injury and progressively reappeared during regeneration. Data were analyzed using one-way ANOVA followed by Dunnett's multiple comparisons test and are presented as mean ± s.d. (*n* = 3 colons). (G) Colonic contents were collected from mice at the indicated regeneration stages. The osmolality of the colonic supernatant was measured using a Dew Point Osmometer. Note that colonic osmolality was markedly elevated during the early stage of regeneration and returned to homeostatic levels at the late stage of regeneration. Data were analyzed using one-way ANOVA followed by Tukey's multiple comparisons test and are presented as mean ± s.d., *n* = 7 mice. UI, uninjured. (H–J) Recombinant adeno-associated viruses carrying the indicated OXSR1 variants were delivered to *Oxsr1* cKO mice. Colon sections were collected on day 4 of the regeneration phase and subjected to H&E (H) and Ki67 (I) staining. Mouse survival was monitored daily (J). Survival curves were analyzed using the log-rank (Mantel–Cox) test. *n* for each group is shown. Scale bars in (H): 500 μm (upper panels) and 100 μm (lower panels). Scale bar in (I): 50 μm.

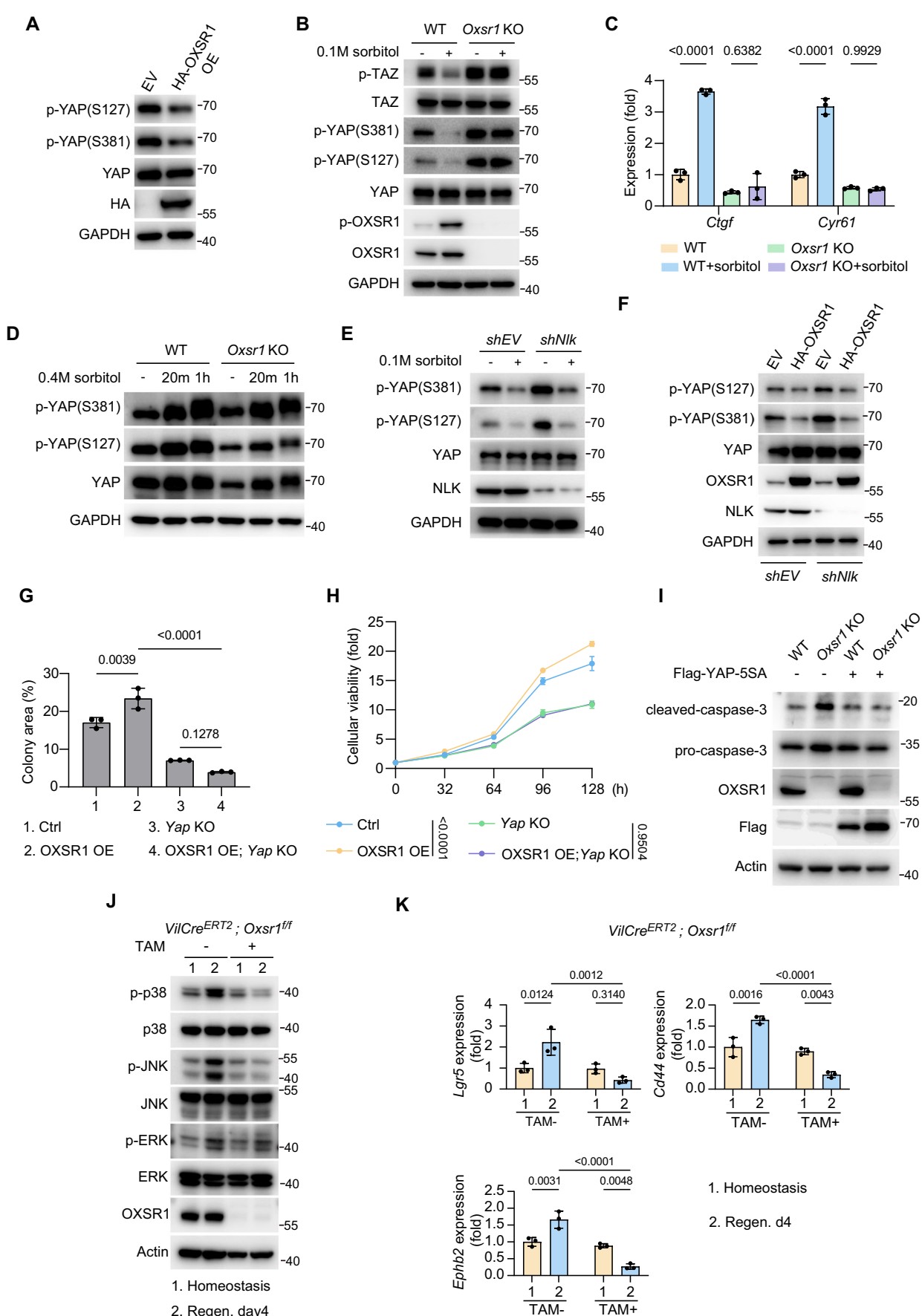

◀ **Figure EV3. OXSR1 regulates cell proliferation via YAP.**

(**A**) Western blot was performed using HEK293T cells that stably express HA-OXSR1. Note the reduced phosphorylation of YAP upon OXSR1 overexpression. EV, empty vector. (**B, C**) Western blot (**B**) and RT-qPCR (**C**) were performed in HeLa cells (**B**) and HEK293T cells (**C**) with the indicated genotypes that were treated with sorbitol (0.1 M, 20 min). Note the reduced YAP/TAZ phosphorylation and enhanced YAP/TAZ target gene expression under these mild hyperosmotic stress conditions. Also note that this phenomenon was missing in *Oxsr1* knockout cells. Data in (**C**) were analyzed using two-way ANOVA followed by Tukey's multiple comparisons test and are presented as mean ± s.d., $n = 3$ independent experiments. (**D**) Western blot was performed in WT or *Oxsr1*-null HEK293T cells that were treated with sorbitol (0.4 M) for the indicated time points. Note the increased YAP phosphorylation under these intense hyperosmotic stress conditions. (**E**) Western blot analysis in HeLa cells expressing the indicated shRNAs, and were treated with sorbitol (0.1 M, 20 min). Note that knockdown of *Nlk* did not affect YAP activation induced by mild osmotic stress. (**F**) Similar to (**A**) except for the additional *Nlk* knockdown cells. Note that the reduced YAP phosphorylation caused by OXSR1 overexpression was not affected by *Nlk* depletion. (**G**) Clonogenic assay of HEK293T cell lines with the indicated genotypes. Note that the increased colony formation observed in OXSR1 overexpressing cells was completely abrogated by *Yap* knockout. Data were analyzed using two-way ANOVA followed by Tukey's multiple comparisons test and are presented as mean ± s.d., $n = 3$ independent experiments. (**H**) Cell viability of the indicated HEK293T cell lines was assessed using the CCK-8 assay. Note that the increase in viability caused by OXSR1 overexpression was completely rescued by *Yap* knockout. Data were analyzed using two-way ANOVA followed by Tukey's multiple comparisons test and are presented as mean ± s.d., $n = 3$ independent experiments. (**I**) Western blot analysis of WT or *Oxsr1* KO HEK293T cells transfected with empty vector or Flag-YAP-5SA. Note that overexpression of the constitutively active YAP-5SA abolished the increased levels of cleaved caspase-3 in *Oxsr1* KO cells. (**J, K**) Colon samples from *Oxsr1* cKO (TAM + ) and control (TAM−) mice were collected and analyzed by western blot (**J**) or RT-qPCR (**K**). Note that the increase in the phosphorylation of p38, JNK, and ERK, and the induction of Wnt target genes *Lgr4*, *Ephb2*, and *Cd44* in control colons during the regeneration phase were completely abolished upon *Oxsr1* knockout. For (**K**), data were analyzed using two-way ANOVA followed by Tukey's multiple comparisons test and are presented as mean ± s.d., $n = 3$ independent experiments. The gel and microscopy images shown are representative of at least two independent experiments.

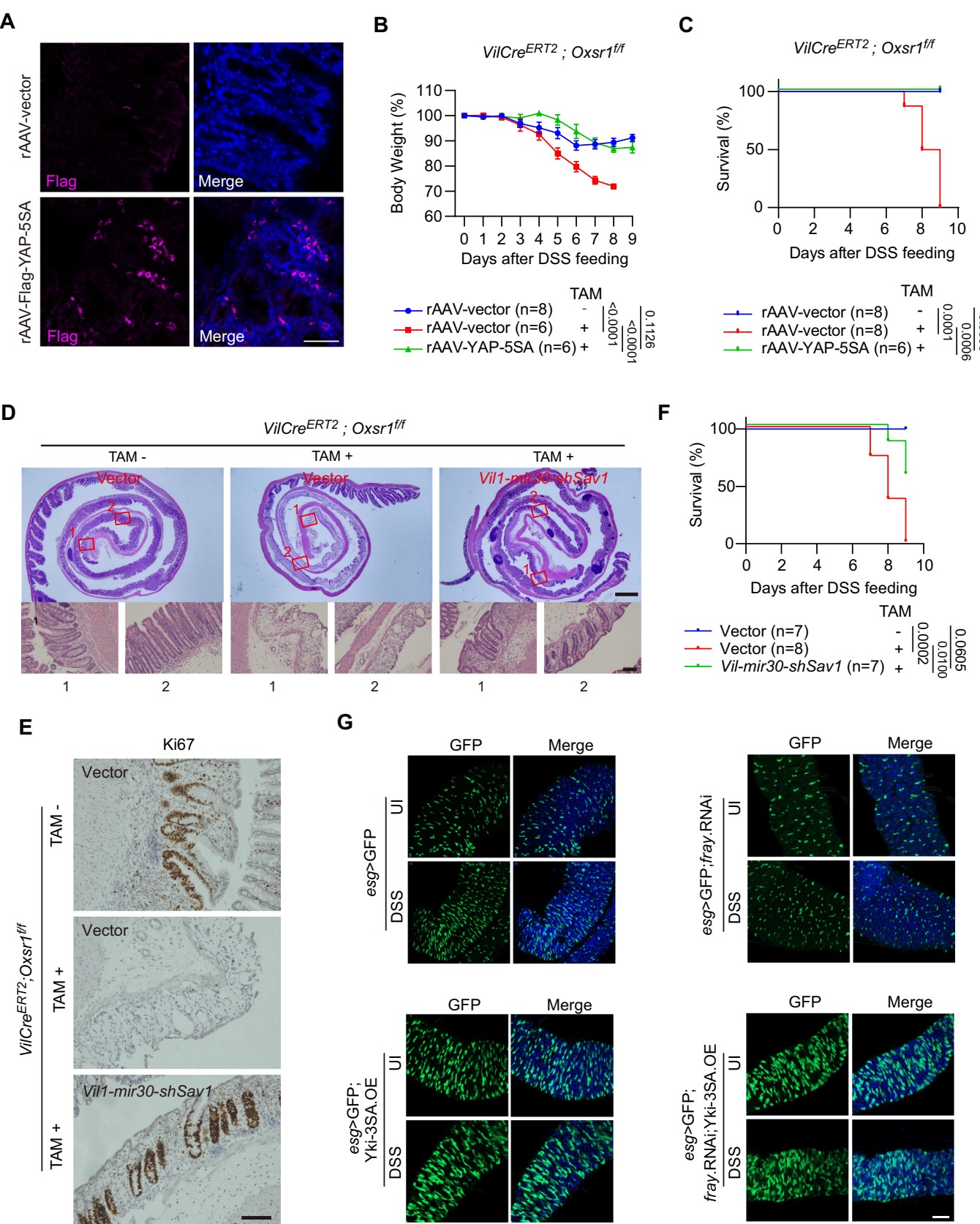

◀ **Figure EV4. Activation of YAP or Yki rescues regeneration defects in *Oxsr1* cKO mice or *fray* knockdown flies.**

(A) Delivery of rAAV-Flag-YAP-5SA to the mouse intestine was confirmed by immunostaining with anti-Flag antibody (pink). Cell nuclei were counterstained with DAPI (blue). Scale bar: 50 μm. (B, C) Body weight (B) and survival (C) of the indicated mice were monitored daily. In accordance with animal welfare guidelines, mice that lost more than 30% of their initial body weight were euthanized. Note that overexpression of YAP-5SA rescued the increased body weight loss and lethality caused by *Oxsr1* deletion. For (B), data were analyzed using two-way ANOVA followed by Tukey's multiple comparisons test, and are presented as mean ± s.d., *n* for each group is shown. For (C), survival curves were analyzed using the log-rank (Mantel–Cox) test. *n* for each group is shown. (D–F) Recombinant adeno-associated virus carrying mir30-based short-hairpin RNA targeting *Sav1* under the control of the *Vil1* promoter was delivered to *Oxsr1* cKO and control mice. Colon sections on day 4 of the regeneration phase were subjected to H&E (D) and Ki67 (E) staining. Survival of the indicated mice was monitored daily (F). Scale bar in (D): 500 μm (upper panel) and 100 μm (lower panel). Sale bar in (E): 50 μm. For (F), survival curves were analyzed using the log-rank (Mantel–Cox) test. *n* for each group is shown. (G) ISC/EB cells in the midguts of the indicated flies were visualized through *esg* > GFP reporter under uninjured (UI) conditions or following DSS treatment (injury day 2). Note that overexpression of a constitutively active form of Yki (Yki-3SA) restored the number of ISC/EB cells following DSS treatment in *fray* knockdown flies. The data shown are representative images of fly midgut (R4 region). Scale bar: 200 μm. Data shown are representative of at least two independent experiments.

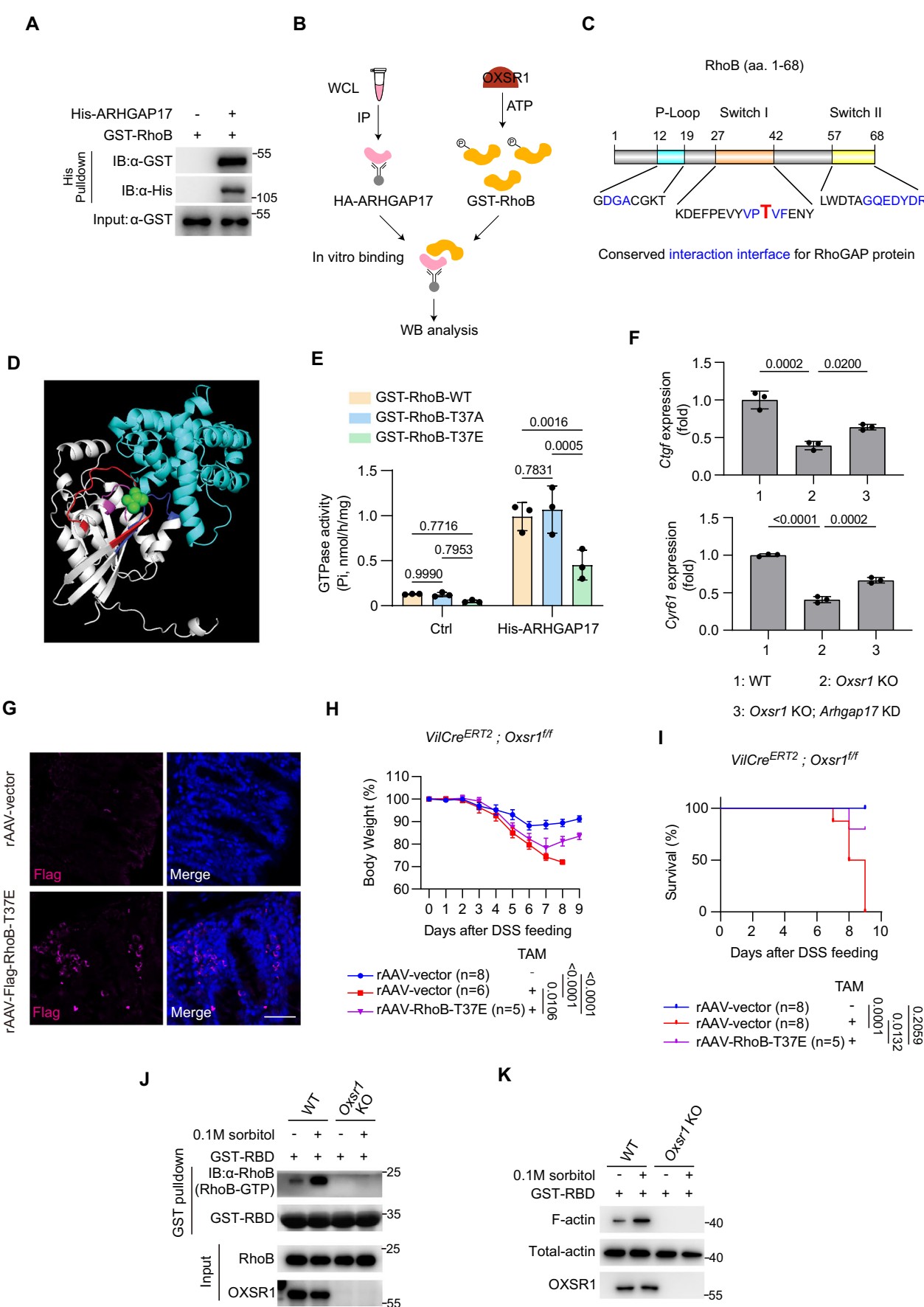

◄    **Figure EV5.   OXSR1 phosphorylates RhoB and inhibits its interaction with ARHGAP17.**

(A) His pull-down assay using bacterially purified His-ARHGAP17 and GST-RhoB revealed a direct interaction between ARHGAP17 and RhoB. (B) Schematic illustration of the in vitro kinase assay followed by an in vitro binding assay. Bacterially purified recombinant His-OXSR1 and GST-RhoB were subjected to in vitro kinase assay, then incubated with HA-ARHGAP17 immunoprecipitated from HEK293T cell lysates. WCL, whole cell lysate. (C) Schematic diagram showing the conserved RhoGAP interaction interface within RhoB (aa 1–68). The T37 residue is highlighted in bold red. (D) Predicted 3D structure of the complex formed by human RhoB and the GAP domain of ARHGAP17 (aa 246–446), as modeled by AlphaFold. RhoB is shown in gray, with key regions highlighted: the P-loop (purple), switch I (red), switch II (blue), and residue T37 (green). The GAP domain of ARHGAP17 is depicted in cyan. (E) In vitro GTPase activity of GST-RhoB-WT, GST-RhoB-T37A, and GST-RhoB-T37E in the absence or presence of purified His-ARHGAP17. Purified RhoB proteins (WT, T37A, or T37E) were incubated with or without His-ARHGAP17, and GTP hydrolysis was quantified using a commercial kit (Beyotime #P2435S). Note that ARHGAP17 markedly enhanced the GTPase activity of RhoB-WT and RhoB-T37A, whereas the phosphomimetic RhoB-T37E exhibited only a modest increase. Data were analyzed using two-way ANOVA followed by Tukey's multiple comparisons test and are presented as mean ± s.d., $n = 3$ independent experiments. (F) RT-qPCR analysis of *Ctgf* and *Cyr61* mRNA levels in HEK293T cells of the indicated genotypes. Note that the reduction in these YAP target genes upon *Oxsr1* knockout was partially rescued by knockdown of *Arhgap17*. Data were analyzed using one-way ANOVA followed by Tukey's multiple comparisons test and are presented as mean ± s.d., $n = 3$ independent experiments. (G) Delivery of rAAV-Flag-RhoB-T37E to the mouse intestine was confirmed by immunostaining with anti-Flag antibody (pink). Cell nuclei were counterstained with DAPI (blue). The control group (top) is intentionally reused and corresponds to the same rAAV-vector control shown in Fig. EV4A. Scale bar: 50 μm. (H, I) Body weight (H) and survival (I) of the indicated mice were monitored daily. In accordance with animal welfare guidelines, mice that lost more than 30% of their initial body weight were euthanized. Note that overexpression of RhoB-T37E rescued the increased body weight loss and lethality caused by *Oxsr1* deletion. For (H), data were analyzed using two-way ANOVA followed by Tukey's multiple comparisons test and are presented as mean ± s.d., *n* for each group is shown. For (I), survival curves were analyzed using the log-rank (Mantel–Cox) test. *n* for each group is shown. (J, K) RhoB-GTP levels (J) and F-actin levels (K) were assessed in wild-type or *Oxsr1* knockout SW480 cells with or without sorbitol treatment (0.1 M, 1 h). Note that sorbitol treatment elevated RhoB-GTP and F-actin levels in wild-type cells, but this response was abolished in *Oxsr1*-null cells. The gel and microscopy images shown are representative of at least two independent experiments.

