## [Peer Review File · The EMBO Journal]

The WNK-OXSR1 osmosensing pathway mediates intestinal regeneration via Hippo-YAP Signaling

Heming Cao, Xiawei Huang, Xiaobing Jiang, Jingrong Deng, Jiahui Wang, Chengfang Wu, Minhuang Hu, Bei Zeng, Zhihao Hu, Huimin Pan, Yuxia Yang, Kewei Zheng, Rui Shen, Mingqing Zhang, and Bo Liu

Corresponding author: Bo Liu (bliu23@xmu.edu.cn)

Review Timeline:

Submission Date:	11th Jun 25
Editorial Decision:	21st Jul 25
Revision Received:	25th Dec 25
Editorial Decision:	21st Jan 26
Revision Received:	6th Feb 26
Accepted:	17th Feb 26

Editor: Ieva Gailite

Transaction Report:

Dear Bo,

Thank you for submitting your manuscript for consideration by the EMBO Journal. We have now received comments from a full set of reviewers, which are included below for your information.

As you can see, the reviewers appreciate the novelty of the findings and find the identification of WNK-OXSR1 as a new regulator of the Hippo pathway of interest. However, they also find that substantial further analysis would be needed to convincingly substantiate this proposed role.

Based on the overall interest expressed in the reviewers' reports and your willingness to engage in a major revision as discussed during the pre-decision consultation, I would like to invite you to revise the manuscript as outlined in your revision plan. I should add that it is The EMBO Journal policy to allow only a single major round of revision and that it is therefore important to resolve the main concerns at this stage.

We generally allow three months as standard revision time, which can be extended to six months in the case of major revisions. Should you foresee a problem in meeting this deadline, please let us know in advance to discuss an extension.

As a matter of policy, competing manuscripts published during this period will not negatively impact on our assessment of the conceptual advance presented by your study. However, please contact me as soon as possible upon publication of any related work to discuss the appropriate course of action.

When preparing your letter of response to the referees' comments, please bear in mind that this will form part of the Review Process File and will therefore be available online to the community. For more details on our Transparent Editorial Process, please visit our website: <https://www.embopress.org/page/journal/14602075/authorguide#transparentprocess>. Please also see the attached instructions for further guidelines on preparation of the revised manuscript.

Please feel free to contact me if you have any further questions regarding the revision. Thank you for the opportunity to consider your work for publication. I look forward to your revision.

With best wishes,

Ieva

- a point-by-point response to the referees' comments, with a detailed description of the changes made (as a word file).
- a word file of the manuscript text.
- individual production quality figure files (one file per figure)

- a complete author checklist, which you can download from our author guidelines (<https://www.embopress.org/page/journal/14602075/authorguide>).

- Expanded View files (replacing Supplementary Information)

We realize that it is difficult to revise to a specific deadline. In the interest of protecting the conceptual advance provided by the work, we recommend a revision within 3 months (19th Oct 2025). Please discuss the revision progress ahead of this time with the editor if you require more time to complete the revisions.

Referee #1:

In this work, the authors demonstrate that osmolarity acts as a damaging signal and plays a role in intestinal regeneration in both *Drosophila* and mice. After tissue damage, changes in osmolarity are detected by WNK-OXSR1 kinases, which then regulate intracellular actin dynamics and suppress the Hippo signaling pathway. As a result, the OXSR1-mediated activation of YAP, a downstream effector of the Hippo pathway, promotes cell proliferation and tissue repair. Overall, this study not only identifies a new regulator and potential drug target for tissue regeneration but also provides new insights into how the Hippo pathway is controlled in vivo in response to tissue damage. However, to fully support the authors' claims, some gaps and inconsistencies need to be addressed.

Major concerns:

1. OXSR1 phosphorylation and activity are frequently used as markers of osmotic stress, which can lead to ambiguity in data interpretation. Other stress signals might activate WNK-OXSR1, and OXSR1 could also influence tissue osmolarity. More accurate methods should be used to measure regional osmolality in the intestines at different time points.
2. The gut or intestinal phenotype should be studied in more detail. Dynamic changes in stem cells and their progeny can be examined using immunostaining or scRNA-seq. Because Wnt signaling is crucial for the development and maintenance of the intestine, the expression of genes targeted by the Wnt pathway should be analyzed. Additionally, the specificity of rescue experiments in both *Drosophila* and mice is a concern, as these are primarily performed using AAV-mediated methods, where active YAP/Yki should be expressed in various cell types, including different mesenchymal cells. Using active YAP is also problematic because it does not accurately mimic physiological or pathological YAP activation. Deleting Lats1, Lats2, or Sav1 may offer a better approach in this context.
3. The effect of high or low osmolality on YAP phosphorylation has been previously examined (Hong et al, EMBO Reports), and no reduction in YAP phosphorylation has been observed under any conditions. The phosphorylation of S128 caused by osmosis might indirectly affect the pYAP(S127) signal in immunoblotting, and this possibility should be investigated using NLK KO cells. Additionally, the potential crosstalk between WNK-OXSR1 and NLK deserves further study.
4. Intestinal regeneration is a dynamic process. As reported previously, YAP activity in the regenerating intestine is induced on Day 2 but downregulated by Day 4 (Cai et al., G&D). However, this study mainly used Day 4 specimens. More time points should be analyzed for YAP activity and OXSR1 activity.
5. The mechanism by which the actin cytoskeleton regulates the Hippo pathway remains unclear. However, the activity of MST1/2 and MAP4K1-7 generally stays unchanged during actin remodeling. This study has shown a significant change in pMST1/2 and MAP4K2 activity in OXSR1 KO cells, suggesting that upstream kinases (TAOK) or phosphatases (STRIPAK) are activated. This possibility should also be tested.

Minor concerns:

1. The impact of osmosis and *Oxsr1* deletion on TAZ phosphorylation and expression should be examined.
2. *Oxsr1* mutant animals exhibit reduced YAP activity under normal growth conditions (Fig 3E), indicating a potential role in organ growth. It is worth investigating whether OXSR1 influences organ size. A double mutant of *Oxsr1* and *Sav1* might provide further insights.

3. A comprehensive analysis of the RNA-seq data should be conducted. Based on the data in Figs. 3A and B, the change in the Hippo pathway is not robust. For genes that are significantly up- or down-regulated in the absence of OXSR1, what processes are involved?
4. For RhoB T37 mutants, it is crucial to examine their basal GTPase activity in the presence or absence of ARHGAP17.
5. The effect of Rac1 on YAP phosphorylation is inconclusive. Rac1 expression is significantly lower (Fig 6E,F).
6. Does ARHGAP17 KO/KD eliminate the effect of OXSR1 on YAP activity?
7. In the absence of OXSR1, does osmotic stress still affect F-actin?
8. For key reagents such as mice, Drosophila, and cell lines, the source publication must be cited.
9. Molecular weight for immunoblots should be added.
10. Upon tissue damage, released molecules can trigger multiple signaling events at the same time. While it may be difficult to exclude factors other than osmosis changes, discussing alternative mechanisms is important.
11. Are SPAK/OXSR1 known targets involved in tissue regeneration? In general, most of them regulate osmolarity, and their roles in tissue homeostasis should be discussed.

Referee #2:

Upon injury, the intestinal epithelium is known to undergo a YAP signaling-mediated regenerative program to repair damaged tissue. However, it remains unclear how epithelial cells sense damage and initiate regeneration. Employing elegant Drosophila and mouse genetic studies as well as comprehensive biochemical experiments, the authors demonstrate that increased osmolarity through WNK-OXSR signaling promotes YAP signaling by activating RhoB and F-actin polymerization. Revealing the critical role of WNK-OXSR signaling in regeneration, this study provides new mechanistic insight into regeneration. This work would likely be of general interest, making significant conceptual advances in the regeneration biology field. Although I understand that most biochemical studies were performed using cell lines (non-gut epithelial cells), a few key aspects such as F-actin and RhoB activation should be confirmed in gut epithelial cells.

In Figs. 2K-2M and S3B, the authors demonstrated that osmolarity closely mirrors OXSR1 phosphorylation dynamics, showing high OXSR1 phosphorylation levels in recovery day 4. However, the authors' term "recovery" is confusing, as it suggests OXSR1 and osmolality involvement in the later recovery phase after regeneration. Recovery should occur after regeneration is completed, but their schematic diagram in Fig. 2A suggests that recovery occurs right after injury. To avoid this confusion (their central hypothesis in the early regeneration phase but not the late recovery phase), I suggest the authors to replace "recovery" with "regeneration" throughout the manuscript.

Linking the WNK-OXSR1 axis to Yap signaling, the authors analyzed F-actin assembly in HeLa cells and HEK293T cells, which are not relevant to their question in gut epithelial cells. Therefore, they should examine this key mechanistic aspect in gut epithelial cells (gut organoid or cancer cell lines). Could the authors analyze F-actin in their mouse models *in vivo*?

Consistent with my comment above, the authors addressed the key RhoB-mediated upstream mechanisms mediated by WNK-OXSR1 axis in non-gut epithelial cells. They should at least confirm altered RhoB activity in gut epithelial cells.

Referee #3:

The authors identify and elucidate the WNK-OXSR1 osmosensing pathway as a critical regulator of intestinal regeneration. Upon injury, WNK-OXSR1-mediated phosphorylation of RhoB at T37 disrupts its interaction with ARHGAP17, leading to increased GTP-bound RhoB. This cascade enhances F-actin polymerization, which subsequently modulates the Hippo phospho-cascade to drive tissue repair via YAP activation. Through integrated functional phenotyping and mechanistic dissection, the study reveals a novel WNK-OXSR1-Hippo-YAP axis and demonstrates its therapeutic potential in mitigating colitis-associated oncogenicity. I have listed my concerns as below.

Major Points:

Major Concerns:

1. Rationale for pathway selection. While the introduction rightly notes osmolarity changes as an underexplored damage signal, it does not sufficiently justify the specific focus on WNK-OXSR1 over other established osmosensors (e.g., p38/MAPK, TRPV4, NFAT5). Given the prominent enrichment of the MAPK pathway in the authors' own KEGG analysis (Fig. 3A) - a pathway directly implicated in osmosensing and intestinal biology - the rationale for prioritizing WNK-OXSR1 requires explicit elaboration, particularly regarding its relative contribution versus parallel osmosensing mechanisms in the regeneration context.
2. RNAseq data interpretation and Hippo pathway. The reliance on RNAseq from OXSR1-KO cell lines (Fig. 3A-C) rather than regenerating intestinal tissue significantly limits physiological relevance, as cell lines cannot recapitulate the complex

multicellular interactions of in vivo regeneration. Furthermore, the KEGG data (Fig. 3A) show the MAPK pathway is more prominently altered than Hippo, and changes in YAP target genes (Fig. 3B) are not the most pronounced. The authors must provide a compelling justification for focusing subsequent mechanistic studies on the Hippo pathway over other significantly altered pathways, especially MAPK. This is crucial given the introduction's emphasis on Hippo's established role in intestinal regeneration and the stated gap in understanding how YAP is activated.

3. Physiological relevance of cell models. Mechanistic experiments (OXSR1-KO, F-actin imaging, RhoB phosphorylation) primarily utilize HEK293 and HeLa cells. The use of human colon epithelial cell lines (e.g., NCM460, HT-29) or, ideally, primary intestinal organoids would considerably strengthen the physiological relevance to the intestinal regeneration processes central to the study's claims.

4. Inconsistency in mechanistic validation: The proposed model (Fig. 2K) highlights OXSR1 phosphorylation/dephosphorylation as the key upstream regulatory event triggered by osmotic changes/injury. However, functional validation relies predominantly on OXSR1 knockout. To convincingly support the model, direct evidence using phospho-mimic (e.g., OXSR1 T185E) or phospho-dead mutants to modulate pathway activity in regeneration assays is essential.

Minor Points:

1. Molecular weight markers must be included for all blots. Phosphorylation data should be quantified relative to corresponding total protein levels, not grayscale ratios, to ensure accurate interpretation.
2. Data in Fig. 1A-1B require validation using at least two independent RNAi lines, consistent with the approach in Fig. 1F.
3. Comparing colony numbers in Fig. S2A is invalid due to significant heterogeneity in colony size. Quantitative analysis should instead measure total area covered or incorporate size-normalized metrics.
4. Phenotypic assessment at "Recovery Day 4" (Fig. 2D-2H) lacks correlation with OXSR1 phosphorylation dynamics. A time-course analysis of phospho-OXSR1 levels post-injury (complementing Fig. 2M) is essential to justify this endpoint.
5. JAS treatment elevates p-YAP levels (indicating inactivation) in WT cells (Fig. 5E), yet upregulates YAP target genes (indicating activation) in Fig. 5F-5G. This contradiction requires resolution.
6. The specific basis for selecting RhoB and Rac1 for analysis in Fig. 6 must be explicitly stated (e.g., prior linkage to osmosensing, Hippo regulation, or preliminary screening data).
7. The citation "Fig. S7c" in line 489 should be corrected to "Fig. S6c."

Response to reviewers' comments (EMBOJ-2025-121597-T):

Dear reviewers and editor,

Thank you very much for your favorable review and your insightful comments to improve our manuscript. We have conducted additional experiments and revised the manuscript accordingly. The following is a point-by-point response to reviewers' comments. For your convenience, we highlight the changes in the revised manuscript in red font. We include most of the new results in the revised manuscript. However, due to space limit, some results and explanations are included only in this rebuttal letter.

We hope you will find the revised manuscript satisfactory.

Referee #1:

In this work, the authors demonstrate that osmolarity acts as a damaging signal and plays a role in intestinal regeneration in both Drosophila and mice. After tissue damage, changes in osmolarity are detected by WNK-OXSRI kinases, which then regulate intracellular actin dynamics and suppress the Hippo signaling pathway. As a result, the OXSRI-mediated activation of YAP, a downstream effector of the Hippo pathway, promotes cell proliferation and tissue repair. Overall, this study not only identifies a new regulator and potential drug target for tissue regeneration but also provides new insights into how the Hippo pathway is controlled in vivo in response to tissue damage. However, to fully support the authors' claims, some gaps and inconsistencies need to be addressed.

We thank the reviewer for recognizing the novelty of our study. Below we address the reviewer's specific points.

Major concerns:

1. OXSRI phosphorylation and activity are frequently used as markers of osmotic stress, which can lead to ambiguity in data interpretation. Other stress signals might activate

WNK-OXSR1, and OXSR1 could also influence tissue osmolarity. More accurate methods should be used to measure regional osmolality in the intestines at different time points.

Thank you for this insightful suggestion. In our study, we do measure intestinal osmolarity at multiple time points post-injury using an osmometer, an established and widely accepted method in the field (Chai, Shen et al., 2024, Eherer & Fordtran, 1992, Shiau, 1987, Shiau, Feldman et al., 1985, Tropini, Moss et al., 2018) (Figure S3B of the initial submission; now Figure EV2F in the revised manuscript). As shown in Figure 2H-2J and Figure EV2F of our revised manuscripts, the temporal pattern of intestinal osmolarity closely parallels the dynamics of OXSR1 phosphorylation, supporting the interpretation that OXSR1 activation is, at least in part, a response to osmotic stress.

Regarding the suggestion to assess regional intestinal osmolality, we agree that this would provide additional spatial resolution. However, we were unable to identify a robust, validated method for accurately measuring localized osmolarity within the intestinal tract in vivo. This limitation likely stems from technical challenges: local osmotic gradients are highly labile and are likely disrupted during tissue handling and sample preparation due to diffusion, dilution, or loss of luminal exudates, thereby compromising the reliability of localized measurements. Given these constraints, we believe that whole-colon osmolarity measurements, as presented in Figure EV2F, provide a reasonable and physiologically meaningful surrogate to capture the overall luminal osmotic changes during intestinal injury and repair. These data are also shown below as **Author Response Figures 1 & 2** for the reviewer's convenience.

Author Response Figure 1. Phosphorylation levels of OXSR1 in the colonic crypts during the indicated stages of regeneration were assessed by immunohistochemistry (H), immunofluorescence (I), and immunoblotting (J). Note the elevated phosphorylation of OXSR1 at the early regeneration stage (Regen. d4), which returned to baseline levels at the late regeneration stage (Regen. d9). UI, uninjured. Scale bars in (H) and (I), 100 μ m.

Author Response Figure 2. Colonic contents were collected from mice at the indicated regeneration stages. The osmolality of the colonic supernatant was measured using a Dew Point Osmometer. Note that colonic osmolality was markedly elevated during the early stage of regeneration and returned to homeostatic levels at the late stage of regeneration. Data were analyzed using one-way ANOVA followed by Tukey's multiple comparisons test and are presented as mean \pm s.d., $n = 7$ mice. UI, uninjured.

2. *The gut or intestinal phenotype should be studied in more detail. Dynamic changes in stem cells and their progeny can be examined using immunostaining or scRNA-seq.*

We thank the reviewer for the insightful suggestion to further characterize the intestinal phenotype and to examine stem cell dynamics in greater detail. As recommended, we expanded our analysis of intestinal stem cell behavior during injury and regeneration.

We initially attempted immunofluorescence staining for Lgr5, but consistent detection could not be achieved due to the lack of a reliable antibody for Lgr5 protein. To overcome this limitation, we used Lrig1, another marker of intestinal stem cells (Yui, Azzolin et al., 2018), to visualize stem cell dynamics. We examined Lrig1 expression across the same time points used in Figures 2H and 2I. The results showed a pronounced loss of Lrig1⁺ stem cells during the injury phase, followed by a gradual re-emergence during regeneration, coinciding with the induction of p-OXSR1. These findings indicate that stem cell restoration temporally aligns with OXSR1 activation. This data

has been incorporated as Figure 2K in the revised manuscript (also show below as **Author Response Figure 3** for the reviewer's convenience).

Author Response Figure 3. Dynamic changes of the Lrig1⁺ stem-cell population in colonic crypts during the indicated stages of injury and regeneration were assessed by immunostaining. Note that Lrig1 signal decreased during injury and progressively reappeared during regeneration. Scale bar, 20 μ m.

Moreover, to complement the phenotypic characterization shown in Figures 2D-2F, we performed Lrig1 immunostaining specifically at Regeneration day 4 in control and *Oxsr1* cKO mice. The results showed that *Oxsr1* cKO intestines exhibited markedly reduced Lrig1⁺ stem cells at this early regenerative stage compared to the control intestines, which further support the conclusion that OXSR1 is required for intestinal regeneration. This new data has been incorporated as Figure 2G in the revised manuscript (also show below as **Author Response Figure 4** for the reviewer's convenience).

Together, these expanded analyses provide a more comprehensive view of the stem cell dynamics throughout the injury and repair phases, and they strengthen the mechanistic link between OXSR1 activity and epithelial regeneration.

Author Response Figure 4. Colons from control or *Oxsr1* cKO mice were collected on day 4 of the regeneration phase following DSS treatment and immunostaining for Lrig1. Note that *Oxsr1* cKO colons exhibited markedly reduced Lrig1 expression compared to controls. Scar bar, 20 μ m.

Because Wnt signaling is crucial for the development and maintenance of the intestine, the expression of genes targeted by the Wnt pathway should be analyzed.

As suggested, we performed RT-qPCR analysis of several representative Wnt target genes (*Lgr5*, *Cd44*, and *Ephb2*) in colons from control and *Oxsr1* cKO mice on day 4 of the regeneration phase following DSS treatment. In control colons, expression of *Lgr5*, *Cd44*, and *Ephb2* was robustly induced, whereas this induction was markedly diminished in *Oxsr1* cKO colons, suggesting that Wnt signaling may also contribute to OXSR1-dependent regenerative responses. We have discussed this in the revised Discussion section (**Lines 670-675**). While these findings point to a potential link between OXSR1 and Wnt pathway activity, fully delineating the underlying mechanisms would require more extensive and targeted studies beyond the scope of the present work and are better suited for future studies. The new RT-qPCR data have been added to the revised manuscript as Figure EV3K (also shown below as **Author Response Figure 5** for the reviewer's convenience).

Author Response Figure 5. Colon sections from *Oxsr1* cKO (TAM+) and control (TAM-) mice were collected on day 4 of the regeneration phase. Expressions of the indicated Wnt target genes were analyzed via RT-qPCR. Data were analyzed using two-way ANOVA followed by Tukey's multiple comparisons test and are presented as mean \pm s.d., $n=3$ independent experiments

Additionally, the specificity of rescue experiments in both *Drosophila* and mice is a concern, as these are primarily performed using AAV-mediated methods, where active YAP/Yki should be expressed in various cell types, including different mesenchymal cells. Using active YAP is also problematic because it does not accurately mimic physiological or pathological YAP activation. Deleting *Lats1*, *Lats2*, or *Sav1* may offer a better approach in this context.

We thank the reviewer for this insightful comment. To address the reviewer's concern, we repeated the rescue experiments using a modified AAV vector carrying the intestine-specific *Vill* promoter and a mir30-based shRNA cassette targeting *Sav1* (*AAV2/9-Vill-mir30-shSav1*). This design enables efficient and selective knockdown of *Sav1* in intestinal epithelial cells but not mesenchymal cells. The results indicated that targeted depletion of *Sav1* recapitulated YAP activation and markedly ameliorated the regeneration defects observed in DSS-treated *Oxsr1* cKO mice, as evidenced by restored crypt architecture, increased Ki67-positive proliferating cells, and improved survival. These new data have been incorporated into the revised manuscript as Figure EV4D-F and are also provided below as **Author Response Figure 6** for the reviewer's convenience.

Author Response Figure 6. *Sav1* was depleted in the intestinal epithelial cells of *Oxsr1* cKO mice using *AAV2/9-Vill-mir30-shSav1* vector. Colon sections on day 4 of the regeneration phase were subjected to H&E (D) and Ki67 (E) staining. The survival of the indicated mice was monitored daily (F). Scale bar in (D), 500 μ m (upper panel) and 100 μ m (lower panel). Sale bar in (E), 50 μ m.

3. *The effect of high or low osmolality on YAP phosphorylation has been previously examined (Hong et al, EMBO Reports), and no reduction in YAP phosphorylation has been observed under any conditions. The phosphorylation of S128 caused by osmosis might indirectly affect the pYAP(S127) signal in immunoblotting, and this possibility should be investigated using NLK KO cells. Additionally, the potential crosstalk between WNK-OXSRI and NLK deserves further study.*

Thank you for bringing up this important point. We would like to clarify that, upon careful re-examination of the data presented in *Hong et al. (EMBO Reports)*, a reduction in p-YAP in response to osmotic stress is indeed observable under serum-starvation conditions, in which the basal p-YAP levels are substantially elevated (see the “-serum” panels in Fig. 1A and Lanes 4-6 in Fig. 1B). Consistently, p-Lats levels are also decreased by osmotic stress under these same conditions (compare Lanes 4-6 in Fig. 1B). Moreover, Hong et al. reported increased nuclear translocation of YAP and enhanced expression of YAP target genes by osmotic stress under serum-starvation conditions, further supporting a functional decrease in YAP phosphorylation (Fig. 2). For the reviewer’s convenience, Fig. 1A & 1B from *Hong et al.* was shown below as **Author Response Figure 7.**

These observations suggest that whether a reduction in p-YAP is detectable under mild osmotic stress depends strongly on the basal phosphorylation state of YAP. When basal p-YAP levels are high, the decrease is observable; when basal levels are already low, the change may fall below the detection threshold. In our study, the reduction of p-YAP in response to osmotic stress was readily detectable, likely because the basal p-YAP levels were higher under our cell culture conditions, which, as is well established, are highly sensitive to variables such as cell density and serum concentration.

Author Response Figure 7. Figures 1A and 1B from *Hong et al (PMID: 27979971)*. Note the decreased levels of p-YAP (top row in A and B), p-YAP (S127) (2nd row in A), or p-Lats (2nd row in B) by osmotic stress under serum-starvation conditions.

To further address the reviewer's concern regarding the potential contribution of NLK to the decreased p-YAP (S127/S381) levels under mild osmotic stress conditions, we examined YAP phosphorylation under 0.1 M sorbitol in wild-type and *Nlk*-knockdown HeLa cells. The results indicated that the reduction in p-YAP (S127/S381) was comparable in WT and *Nlk*-knockdown cells, suggesting that NLK may not be involved in this process. Furthermore, we found that OXSR1-induced reduction of p-YAP (S127/S381) was similarly unaffected by *Nlk* knockdown, suggesting that WNK-OXSR1 and NLK may not crosstalk in this context. These data have been incorporated as Figures EV3E and F in the revised manuscripts and are provided below as **Author Response Figure 8** for the reviewer's convenience.

Author Response Figure 8. (E) Western blot analysis in HeLa cells expressing the indicated shRNAs and were treated with sorbitol (0.1 M, 20 min). Note that knockdown of *Nfk* did not affect YAP activation induced by mild osmotic stress. (F) Western blot was performed using HEK293T cells of the indicated genotypes. Note that the reduced YAP phosphorylation caused by OXSRI overexpression was not affected by *Nfk* depletion.

4. Intestinal regeneration is a dynamic process. As reported previously, YAP activity in the regenerating intestine is induced on Day 2 but downregulated by Day 4 (Cai et al., G&D). However, this study mainly used Day 4 specimens. More time points should be analyzed for YAP activity and OXSRI activity.

We appreciate the reviewer's comment regarding the dynamic nature of intestinal regeneration. As noted, the study by *Cai et al. (Genes & Development)* used Day 2 and Day 4 as representative time points for early and late stages of regeneration, respectively. In our experimental setup, however, we observed a slower repair trajectory, with near-complete regeneration occurring around Day 9. Such differences in timing are not totally unexpected and may reflect variations in injury severity, DSS treatment regimen, gut microbiota composition, or other environmental factors that can influence both the onset and duration of YAP activation.

To directly address the reviewer's concern, we have now included Day 2 specimens in our analysis of OXSRI activity and YAP levels. These additional data are incorporated into the revised Figure 2H-2J (also provided above as **Author Response Figure 1** for the reviewer's convenience).

5. The mechanism by which the actin cytoskeleton regulates the Hippo pathway remains unclear. However, the activity of MST1/2 and MAP4K1-7 generally stays unchanged during actin remodeling. This study has shown a significant change in pMST1/2 and MAP4K2 activity in OXSRI KO cells, suggesting that upstream kinases (TAOK) or phosphatases (STRIPAK) are activated. This possibility should also be tested.

We agree with the reviewer that MST1/2 and MAP4Ks activities are generally reported to remain stable during actin remodeling, although another study has reported that MST kinases can be activated by actin cytoskeleton disruption (Densham, O'Neill et al., 2009). Moreover, a previous study demonstrated that combined deletion of

MAP4K4/6/7 and MST1/2 largely abolishes the phosphorylation of LATS and YAP in response to various signals, including actin remodeling (Meng, Moroishi et al., 2015), highlighting the importance of these kinases in this context.

To address the reviewer’s concern, we assessed TAOK2 phosphorylation in *Oxsr1*-null HeLa cells. The results indicated that TAOK2 phosphorylation remained unchanged in *Oxsr1*-null cells, suggesting that TAOK activation is unlikely to underlie the effects of OXSR1 loss. Moreover, we also sought to determine whether OXSR1 affects the activity of the STRIPAK complex. However, due to the lack of an established assay to directly measure STRIPAK enzymatic activity, we evaluated SLMAP, a key STRIPAK component that links the complex to MST1/2 kinases (Zheng, Liu et al., 2017), as a functional proxy. Specifically, we examined whether OXSR1 alters SLMAP protein abundance or its interaction with MST1/2. The results showed that endogenous SLMAP levels were unchanged in *Oxsr1*-knockout HeLa cells, and transient co-expression of OXSR1 and SLMAP in 293T cells did not affect SLMAP protein levels. Likewise, OXSR1 expression did not influence the interaction between SLMAP and MST2. Together, these findings suggest that OXSR1 may not modulate STRIPAK activity through SLMAP. Nonetheless, whether OXSR1 affects other STRIPAK components remains an open question that will require additional tools, such as validated antibodies, for further investigation. These results have been incorporated into the revised manuscript as **Figure 4M** and are also provided below as **Author Response Figure 9** for the reviewer’s convenience.

Author Response Figure 9. Left, western blot analysis in HeLa cells with the indicated genotypes. Note the unaltered p-TAOK2 or SLMAP abundance in *Oxsr1*-knockout cells. Right, co-IP in 293T cells transiently transfected with the indicated constructs. Note that OXSR1 does not affect SLMAP protein levels or the SLMAP-MST2 interaction.

Minor concerns:

1. *The impact of osmosis and Oxsr1 deletion on TAZ phosphorylation and expression should be examined.*

Thanks for the suggestion. We examined TAZ phosphorylation and expression in *Oxsr1*-knockout and control HeLa cell under basal conditions and following mild hyperosmotic stress (0.1 M sorbitol). Consistent with our observations for YAP, TAZ phosphorylation was elevated in *Oxsr1*-knockout cells. Moreover, mild hyperosmotic stress (0.1 M sorbitol) resulted in reduced p-TAZ levels in an OXSR1-dependent manner. These new data have been incorporated into the revised manuscript as Figure EV3B and are also provided below as **Author Response Figure 10** for the reviewer's convenience.

Author Response Figure 10. Western blot was performed in HeLa cells with the indicated genotypes that were treated with sorbitol (0.1 M, 20 min). Note the reduced YAP/TAZ phosphorylation under this mild hyperosmotic stress conditions. Also note that this phenomenon was missing in *Oxsr1* knockout cells.

2. *Oxsr1* mutant animals exhibit reduced YAP activity under normal growth conditions (Fig 3E), indicating a potential role in organ growth. It is worth investigating whether OXSR1 influences organ size. A double mutant of *Oxsr1* and *Sav1* might provide further insights.

We appreciate the reviewer's insightful comment. We would like to clarify that the data shown in Figure 3E (now Figure 3F in the revised manuscript) were obtained from HEK293T cell line, not from *Oxsr1* mutant animals. In Figure 3I of the revised manuscript, we examined YAP phosphorylation in control and *Oxsr1* cKO colonic crypts. Under homeostatic conditions, neither total YAP levels nor p-YAP levels differed between the two groups (compare lanes 1 and 3), suggesting that, unlike in cultured cells, OXSR1 does not enhance YAP activity in vivo under homeostatic

conditions. Consistent with this conclusion, our data in Figure EV2A show that *Oxsr1* cKO mice exhibit no overt intestinal abnormalities at steady state. These in vivo results strongly suggest that OXSR1 may not influence organ size.

To further test this notion, we turned to *Drosophila*, a well-established animal model for studying organ size control. The results show that manipulation of Fray (the sole fly homolog of OXSR1) expression produced no detectable changes in wing or eye size, corroborating our findings in mice intestine. Together, these results indicate that OXSR1/Fray do not regulate organ size. These data have been incorporated as Figure EV1A-B in the revised manuscript and are also provided below as **Author Response Figure 11** for the reviewer's convenience.

Author Response Figure 11. Representative images and quantifications of adult wings (A) and eyes (B) from flies expressing Fray or *fray* RNAi under the indicated Gal4 drivers. Data were analyzed using one-way ANOVA followed by Dunnett's multiple comparisons test and are presented as mean \pm s.d.; for (A), scale bar: 500 μ m; $n = 9$ flies per group; for (B), scale bars: 100 μ m; $n = 10$ flies per group.

3. A comprehensive analysis of the RNA-seq data should be conducted. Based on the data in Figs. 3A and B, the change in the Hippo pathway is not robust. For genes that are significantly up- or down-regulated in the absence of OXSR1, what processes are involved?

We thank the reviewer for this thoughtful suggestion. To provide a more comprehensive assessment of the transcriptional responses of *Oxsr1* loss, we performed Gene Ontology (GO) enrichment analysis on significantly up- or down-regulated genes. Our GO Biological Process (GO-BP) analysis reveals that genes up-regulated upon *Oxsr1* knockout are involved in diverse processes such as gene transcription, cell adhesion,

nervous system development, and negative regulation of cell proliferation. In contrast, genes down-regulated upon *Oxsr1* knockout are primarily associated with cell division, regulation of apoptosis, and the DNA damage response, processes that align well with known YAP/TAZ-dependent functions. Consistent with these findings, several of the top pathways identified by KEGG analysis in Figure 3A of the initial manuscript are closely linked to Hippo signaling, such as cancer-related pathways (Sanchez-Vega, Mina et al., 2018), HPV infection (He, Mao et al., 2015, Patterson, Cogan et al., 2024), the MAPK signaling pathway (Paul, Hagenbeek et al., 2025, Zhang, Ji et al., 2009), and focal adhesion (Pearson, Huang et al., 2021). Together, these data support the conclusion that Hippo pathway activity is differentially regulated upon loss of OXSR1. The GO-BP results are provided below as **Author Response Figure 12** for the reviewer’s convenience.

Author Response Figure 12. GO-BP enrichment analysis of down-regulated (left) and up-regulated (right) differentially expressed genes (DEGs) in *Oxsr1* knockout cell.

4. For *RhoB* T37 mutants, it is crucial to examine their basal GTPase activity in the presence or absence of *ARHGAP17*.

Thank you for the insightful suggestion. As suggested, we measured the GTPase activity of RhoB-WT and the two RhoB T37 mutants (T37E and T37A) in the presence or absence of bacterially purified ARHGAP17 using a commercial colorimetric GTPase activity kit (Beyotime, Cat# P2435S). The results show that, in the absence of ARHGAP17, all three RhoB variants displayed comparably low basal GTPase activity. Upon addition of ARHGAP17, the GTPase activity of RhoB-WT and the non-phosphorylatable RhoB-T37A mutant was robustly stimulated, whereas stimulation of the phophomimetic RhoB-T37E mutant was much weaker. There results not only

underscore the role of ARHGAP17 as a RhoB GAP by targeting the T37 residue, but also highlight the essential role of T37 phosphorylation in this process. These new data have been incorporated into the revised manuscript as Figure EV5E and are also provided below as **Author Response Figure 13** for the reviewer's convenience.

Author Response Figure 13. Bacterially purified GST-RhoB proteins (WT, T37A, or T37E) were incubated with or without His-ARHGAP17, and GTP hydrolysis was quantified using GTPase activity kit (Beyotime Cat#P2435S). Data were analyzed using two-way ANOVA followed by Tukey's multiple comparisons test and are presented as mean \pm s.d., $n=3$ independent experiments.

5. The effect of Rac1 on YAP phosphorylation is inconclusive. Rac1 expression is significantly lower (Fig 6E,F).

We appreciate the reviewer's comment. To address this concern, we adjusted the amount of transfected plasmids to ensure that Rac1 was expressed at levels comparable to RhoB. Consistent with our original conclusion, only RhoB, rather than Rac1, affected the phosphorylation of either exogenous or endogenous YAP, even when Rac1 was expressed at levels similar to, or higher than, RhoB. We have replaced the original Figures 6E and 6F with the updated data (now Figures 6F and 6G), which are also provided below as **Author Response Figure 14** for the reviewer's convenience.

Author Response Figure 14. (F) Co-IP was performed in HEK293T cells transfected with the indicated plasmids. YAP phosphorylation levels were assessed by western blot. Note the reduced phosphorylation of exogenous YAP upon co-expression with Flag-RhoB, but not with Flag-Rac1. (G) Western blot was performed in HEK293T cells transfected with the indicated plasmids. Note the reduced phosphorylation of endogenous YAP by RhoB, but not Rac1.

6. Does ARHGAP17 KO/KD eliminate the effect of OXSRI on YAP activity?

To address this concern, we generated an *Arhgap17* knockdown (KD) HEK293T cell line and examined whether ARHGAP17 depletion counteracts the effect of *Oxsr1* knockout on YAP activity. RT-qPCR analysis shows that *Arhgap17* KD partially rescued the decreased expression of the YAP target genes *Ctgf* and *Cyr61* in *Oxsr1*-knockout cells, consistent with our model in which ARHGAP17 contributes to the effect of OXSRI on YAP activity. These data have been added as Figure EV5F in the revised manuscript and are also provided below as **Author Response Figure 15**.

Author Response Figure 15. RT-qPCR was performed in cells with the indicated genotypes. Note the decreased expression of *Ctgf* and *Cyr61* in *Oxsr1*-knockout cells were significantly rescued by ARHGAP17 depletion.

7. In the absence of OXSRI, does osmotic stress still affect F-actin?

To address this concern, we visualized F-actin in wild-type and *Oxsr1* KO HeLa cells via phalloidin staining. Our results show that sorbitol treatment markedly enhanced F-actin levels in wild-type cells but had minimal effect in *Oxsr1* KO cells, suggesting that osmotic stress requires OXSRI to promote F-actin. The corresponding images are provided below as **Author Response Figure 16**.

Author Response Figure 16. F-actin was detected in wild-type (WT) or *Oxsr1* knockout (KO) HeLa cells via phalloidin staining. Note that sorbitol treatment (0.1 M, 1 hr) enhanced F-actin levels in WT but not *Oxsr1* KO cells.

8. *For key reagents such as mice, Drosophila, and cell lines, the source publication must be cited.*

We have verified that the source publications for key reagents are now properly cited throughout the manuscript.

9. *Molecular weight for immunoblots should be added.*

Molecular weight markers have been added to all relevant immunoblots in the revised manuscript.

10. *Upon tissue damage, released molecules can trigger multiple signaling events at the same time. While it may be difficult to exclude factors other than osmosis changes, discussing alternative mechanisms is important.*

We appreciate the reviewer's insightful comment. We agree that tissue damage can lead to the release of multiple factors that may simultaneously activate diverse signaling pathways. While our data support an important role for osmotic stress in regeneration, we now discuss the possibility that additional damage-associated signals, such as released soluble factors or mechanical cues, may also participate in pathway activation. This discussion has been added to the revised manuscript (**Lines 659-669**) to

acknowledge these alternative mechanisms and to clarify the limitations of our current experimental system.

11. Are SPAK/OXSRI known targets involved in tissue regeneration? In general, most of them regulate osmolarity, and their roles in tissue homeostasis should be discussed.

We thank the reviewer for this comment. At present, there is no direct evidence linking the WNK-SPAK/OXSRI-N(K)CC axis to tissue regeneration. However, components of this pathway including WNK, SPAK/OXSRI, and their best-characterized targets N(K)CCs, have been implicated in diverse cellular processes such as proliferation, migration, cytoskeleton remodeling, and angiogenesis (Algharabil, Kintner et al., 2012, Dbouk, Weil et al., 2014, Gallolu Kankanamalage, Karra et al., 2018, Jung & Cobb, 2023, Jung, Jaykumar et al., 2022, Shiozaki, Nako et al., 2014, Xie, Yoon et al., 2013, Zhang, Meor Azlan et al., 2023). It is therefore plausible that this signaling axis contributes to the regulation of tissue homeostasis. We have expanded the Discussion to address this point in the revised manuscript (**Lines 681-688**).

Referee #2:

Upon injury, the intestinal epithelium is known to undergo a YAP signaling-mediated regenerative program to repair damaged tissue. However, it remains unclear how epithelial cells sense damage and initiate regeneration. Employing elegant Drosophila and mouse genetic studies as well as comprehensive biochemical experiments, the authors demonstrate that increased osmolarity through WNK-OXSRL signaling promotes YAP signaling by activating RhoB and F-actin polymerization. Revealing the critical role of WNK-OXSRL signaling in regeneration, this study provides new mechanistic insight into regeneration. This work would likely be of general interest, making significant conceptual advances in the regeneration biology field. Although I understand that most biochemical studies were performed using cell lines (non-gut epithelial cells), a few key aspects such as F-actin and RhoB activation should be confirmed in gut epithelial cells.

We are grateful to the reviewer for recognizing the novelty and quality of our study. The specific comments are addressed below.

In Figs. 2K-2M and S3B, the authors demonstrated that osmolarity closely mirrors OXSRL phosphorylation dynamics, showing high OXSRL phosphorylation levels in recovery day 4. However, the authors' term "recovery" is confusing, as it suggests OXSRL and osmolality involvement in the later recovery phase after regeneration. Recovery should occur after regeneration is completed, but their schematic diagram in Fig. 2A suggests that recovery occurs right after injury. To avoid this confusion (their central hypothesis in the early regeneration phase but not the late recovery phase), I suggest the authors to replace "recovery" with "regeneration" throughout the manuscript.

We thank the reviewer for this insightful suggestion. To avoid confusion, we have replaced “recovery” with “regeneration” throughout the manuscript.

Linking the WNK-OXSRL axis to Yap signaling, the authors analyzed F-actin assembly in HeLa cells and HEK293T cells, which are not relevant to their question in gut epithelial cells. Therefore, they should examine this key mechanistic aspect in gut

epithelial cells (gut organoid or cancer cell lines). Could the authors analyze F-actin in their mouse models *in vivo*?

Thank you for this insightful suggestion. As recommended, we examined F-actin levels in a relevant gut epithelial context. We first visualized F-actin by phalloidin staining in SW480 cells, a colorectal cancer cell line. Consistent with our findings in HeLa cells (Figure 5A of the revised manuscript) and HEK293T cells (Figure 5C of the revised manuscript), F-actin levels were substantially reduced in *Oxsr1*-null SW480 cells compared with wild-type controls.

In addition, we analyzed F-actin levels in colonic crypts isolated from wild-type or *Oxsr1* conditional knockout mice via western blot. The results indicated that *Oxsr1* knockout crypts exhibited markedly reduced F-actin levels relative to wild-type, consistent with the critical role for OXSR1 in regulating F-actin assembly. These new data have been incorporated as Figure 5B and 5D in the revised manuscript and are also provided below as **Author Response Figure 17** for the reviewer's convenience.

Author Response Figure 17. (B) Phalloidin staining of F-actin in wild-type or *Oxsr1* knockout SW480 cells. Note that knockout of *Oxsr1* suppresses the abundance of Phalloidin-positive F-actin fibers. (D) Western blot was performed to examine the levels of detergent-insoluble F-actin and total actin in primary colonic crypts of the indicated genotypes collected at regeneration day 4. For each genotype, crypts isolated from 4 independent animals were included. Note that *Oxsr1*-deficient crypts exhibit reduced F-actin levels compared with controls.

Consistent with my comment above, the authors addressed the key *RhoB*-mediated upstream mechanisms mediated by *WNK-OXSR1* axis in non-gut epithelial cells. They should at least confirm altered *RhoB* activity in gut epithelial cells.

We thank the reviewer for this insightful comment. We would like to clarify that we have already assessed RhoB-GTP levels in isolated colonic crypts from *Oxsr1* cKO mice and their control littermates in Figure 7L of our initial submission. Our results showed that basal levels of RhoB-GTP are modestly reduced in *Oxsr1* cKO crypts compared to controls (compare Lanes 3 and 1). Moreover, the injury-induced increase in RhoB-GTP levels observed in regenerating control crypts (compare Lanes 2 and 1) is largely abolished in *Oxsr1* cKO crypts (compare Lanes 4 and 3).

To further address the reviewer's concern, we repeated the RhoB-GTP pulldown assay in WT and *Oxsr1*-knockout SW480 colorectal epithelial cells. Consistent with the results from primary colonic crypts, RhoB-GTP levels are reduced in *Oxsr1*-knockout cells compare to WT controls. These additional data have been included in the revised manuscript as Figure 6C (also show below as **Author Response Figure 18** for the reviewer's convenience).

Author Response Figure 18. GTP-bound RhoB was pulled down from SW480 cell lysates using GST-RBD and analyzed by western blot. Note the reduced RhoB-GTP levels in *Oxsr1*-knockout cells compared with wild-type controls.

Referee #3:

The authors identify and elucidate the WNK-OXSRI osmosensing pathway as a critical regulator of intestinal regeneration. Upon injury, WNK-OXSRI-mediated phosphorylation of RhoB at T37 disrupts its interaction with ARHGAP17, leading to increased GTP-bound RhoB. This cascade enhances F-actin polymerization, which subsequently modulates the Hippo phospho-cascade to drive tissue repair via YAP activation. Through integrated functional phenotyping and mechanistic dissection, the study reveals a novel WNK-OXSRI-Hippo-YAP axis and demonstrates its therapeutic potential in mitigating colitis-associated oncogenicity. I have listed my concerns as below.

We thank the reviewer for recognizing the novelty and quality of our study. The specific comments are addressed below.

Major Points:

Major Concerns: 1. Rationale for pathway selection. While the introduction rightly notes osmolarity changes as an underexplored damage signal, it does not sufficiently justify the specific focus on WNK-OXSRI over other established osmosensors (e.g., p38/MAPK, TRPV4, NFAT5). Given the prominent enrichment of the MAPK pathway in the authors' own KEGG analysis (Fig. 3A) - a pathway directly implicated in osmosensing and intestinal biology - the rationale for prioritizing WNK-OXSRI requires explicit elaboration, particularly regarding its relative contribution versus parallel osmosensing mechanisms in the regeneration context.

Thanks for the insightful comment. A central question of our study is how osmotic changes in the gut are sensed and transduced into regenerative signals following injury. For this purpose, we sought to focus on a pathway that is not only specifically responsive to osmotic stress but also functions as a primary and specific sensor of such changes.

Recent studies have established the WNK-OXSRI axis as a direct sensor of osmotic stress. WNK kinases respond to changes in cell volume and intracellular molecular crowding, biophysical consequences of osmotic fluctuations, by undergoing phase

separation and activation. This, in turn, activates downstream effectors such as OXSRI to mediate adaptive responses (Boyd-Shiwarski, Shiwarski et al., 2022). Importantly, this activation is relatively specific to osmotic stress compared to other pathways. As a result, WNK-OXSRI activity (e.g., phosphorylation levels) serves as a robust and interpretable readout of osmolarity changes. This specificity also enhances the interpretability of genetic and pharmacological perturbation experiments, allowing us to more confidently attribute observed phenotypes to altered osmolarity.

By contrast, other known osmosensing pathways, such as MAPK/p38, can be activated by a wide range of non-osmotic stressors (Cargnello & Roux, 2011). As such, changes in these pathways often reflect a broader cellular stress response, which complicates data interpretation when investigating the specific role of osmotic stress in regeneration.

Furthermore, the WNK-OXSRI pathway is ubiquitously expressed and highly conserved across species, including in the intestinal epithelium, making it a compelling candidate for studying conserved mechanisms of osmotic sensing in tissue regeneration. In contrast, proteins like TRPV4 have more restricted expression patterns and limited functional characterization in the context of intestinal regeneration.

In summary, we selected the WNK-OXSRI pathway because it uniquely fulfills two critical criteria for our study: (1) it acts as a relatively specific and direct sensor of osmotic stress, and (2) it is broadly expressed and evolutionarily conserved, making it well suited for mechanistic investigation across multiple model systems. While we fully agree that exploring other osmosensing pathways and molecules would be valuable, we respectfully suggest that a systematic analysis of those pathways would be best pursued in future studies specifically designed for that purpose.

2. RNAseq data interpretation and Hippo pathway. The reliance on RNAseq from OXSRI-KO cell lines (Fig. 3A-C) rather than regenerating intestinal tissue significantly limits physiological relevance, as cell lines cannot recapitulate the complex multicellular interactions of in vivo regeneration.

Thanks for the comment. We agree with the reviewer that cell lines do not fully capture the complexity of multicellular interactions during in vivo regeneration. However, they provide a well-established and efficient model system for the initial identification of candidate pathways, as widely utilized in the field. Importantly, our subsequent analyses employed in vivo models, including *Drosophila* and mice, to validate the physiological relevance of the pathways identified in the *Oxsr1*-KO cell lines. We believe this stepwise approach, combining in vitro screening with in vivo validation, offers a robust and scientifically sound strategy to elucidate OXSR1 function in tissue regeneration.

Furthermore, the KEGG data (Fig. 3A) show the MAPK pathway is more prominently altered than Hippo, and changes in YAP target genes (Fig. 3B) are not the most pronounced. The authors must provide a compelling justification for focusing subsequent mechanistic studies on the Hippo pathway over other significantly altered pathways, especially MAPK. This is crucial given the introduction's emphasis on Hippo's established role in intestinal regeneration and the stated gap in understanding how YAP is activated.

We appreciate the reviewer's careful assessment and the opportunity to clarify our rationale. While the KEGG analysis (Fig. 3A) indeed highlights MAPK signaling as one of the most prominently altered pathways, our decision to center our mechanistic studies on the Hippo pathway was based on both biological significance and the specific knowledge gap our study aimed to address. The Hippo-YAP pathway plays a well-established role in intestinal regeneration, yet the upstream mechanisms by which injury or osmotic stress activate YAP remain poorly understood. This gap aligns directly with a main objective of our study, namely to elucidate how YAP is activated during regeneration. Thus, even if the transcriptional footprint of Hippo signaling is less prominent than that of MAPK, dissecting the upstream regulation of YAP remains highly relevant to the biology of gut repair.

To address the reviewer's concern regarding MAPK signaling, we analyzed MAPK activation during regeneration in *Oxsr1* cKO and control colonic crypts. We found that phosphorylation of p38, JNK, and ERK is induced during regeneration (e.g.,

regeneration day 4) in control mice, but this induction is markedly reduced in *Oxsr1* cKO crypts. These new results (now included as Figure EV3J, and shown below as **Author Response Figure 19**) suggest that MAPK signaling is activated during regeneration in an OXSR1-dependent manner and may contribute to the regenerative process.

Author Response Figure 19. Colonic crypts from *Oxsr1* cKO (TAM+) and control (TAM-) mice were collected at regeneration day 4 and analyzed by western blot. Note the increased phosphorylation of MAPKs in control crypts but not in *Oxsr1* cKO crypts.

The existing literature supports a context-dependent role for MAPK signaling in intestinal injury and regeneration. MAPK activation can promote gut regeneration in *Drosophila* and mouse models (Jiang, Grenley et al., 2011, Zhang, Bandyopadhyay et al., 2020), whereas inhibition of p38/MAPK signaling has been reported to alleviate intestinal inflammation in IBD patients and mouse colitis models (Assi, Pillai et al., 2006, Hollenbach, Neumann et al., 2004, Hommes, van den Blink et al., 2002). These conflicting observations suggest that MAPK signaling may play a context-dependent, possibly biphasic or cell type-specific role in intestinal regeneration. Moreover, a recent study showed that mice with epithelial-specific deletion of MAP3K2, a key component of the MAPK pathway, exhibited comparable degree of DSS-induced colitis to control mice (Wu, Sun et al., 2021). Collectively, these findings suggest that epithelial MAPK pathway may be sufficient but not strictly required for intestinal regeneration.

Overall, our findings identify Hippo-YAP signaling as a key downstream effector of OXSR1 in intestinal regeneration, while not excluding important contributions from additional pathways such as MAPK. We now address this point explicitly in the revised Discussion section (**Lines 670-675**). Given the complexity of signaling crosstalk and the scope of the current study, we hope the reviewer would agree with us that fully defining the role of MAPK signaling and the mechanism by which OXSR1 regulates MAPK activity would be best pursued in future work.

3. Physiological relevance of cell models. Mechanistic experiments (OXSR1-KO, F-actin imaging, RhoB phosphorylation) primarily utilize HEK293 and HeLa cells. The use of human colon epithelial cell lines (e.g., NCM460, HT-29) or, ideally, primary intestinal organoids would considerably strengthen the physiological relevance to the intestinal regeneration processes central to the study's claims.

We appreciate the reviewer's insightful suggestion. As noted in our responses to points 2 and 3 of Reviewer 2, we expanded our mechanistic analyses to include additional intestinal epithelial models. Specifically, we examined the effects of OXSR1 on F-actin assembly and RhoB-GTP levels in the human colon epithelial cell line SW480 as well as in isolated primary colonic crypts. Consistent with our findings in HeLa and HEK293T cells, OXSR1 deletion markedly reduced F-actin abundance (Figure 5B in the revised manuscript; see also above **Author Response Figure 17**) and RhoB-GTP levels (Figure 6C in the revised manuscript; see also above **Author Response Figure 18**) in SW480 cells compared with wild-type controls. Moreover, in primary crypts, the injury-induced increases in F-actin (Figure 5D in the revised manuscript; see also **Author Response Figure 17**) and RhoB-GTP (Figure 7L) observed in regenerating control crypts were largely abolished in *Oxsr1* cKO crypts. Together, these additional experiments confirm that OXSR1 is essential for regulating RhoB activity and F-actin assembly within intestinal epithelial contexts, thereby substantiating the physiological relevance of our mechanistic findings.

4. Inconsistency in mechanistic validation: The proposed model (Fig. 2K) highlights OXSR1 phosphorylation/dephosphorylation as the key upstream regulatory event triggered by osmotic changes/injury. However, functional validation relies

predominantly on OXSR1 knockout. To convincingly support the model, direct evidence using phospho-mimic (e.g., OXSR1 T185E) or phospho-dead mutants to modulate pathway activity in regeneration assays is essential.

We appreciate the reviewer's insightful comment. We would like to clarify that, in addition to using *Oxsr1* conditional knockout mice, we also employed Rafoxanide (Rafo), a well-characterized chemical inhibitor of OXSR1 kinase activity (AlAmri, Kadri et al., 2017). Rafo treatment significantly impaired intestinal regeneration in both *Drosophila* (Fig. S1B in the initial submission, now Fig. EV1F) and mouse models (Fig. 2H-2J in the initial submission, now Fig. EV2C-E), supporting the functional importance of OXSR1 kinase activity.

Given that OXSR1 phosphorylation is upregulated during intestinal regeneration (Figs. 2K-2M in our initial submission), we anticipate that the phospho-mimetic OXSR1 mutant would behave similarly to wild-type OXSR1 in rescuing the regeneration defects in *Oxsr1* cKO mice. To test this and to directly address the reviewer's concern, we introduced either a phospho-mimetic mutant (OXSR1-T185E) or a phospho-dead mutant (OXSR1-T185A) into *Oxsr1* cKO mice using the AAV2/9 vector. The results showed that expression of OXSR1-T185E, but not OXSR1-T185A, substantially restored intestinal regeneration in *Oxsr1*-deficient mice, as evidenced by enhanced crypt regeneration, increased epithelial proliferation, and improved survival following DSS treatment. These results provide direct functional validation of our proposed model and demonstrate that OXSR1 phosphorylation and kinase activity are essential for intestinal regeneration. The new data have been incorporated as Figures EV2G-I in the revised manuscript and are provided below as **Author Response Figure 20** for the reviewer's convenience.

Author Response Figure 20. Recombinant adeno-associated viruses carrying the indicated OXSR1 variants were delivered into *Oxsr1* cKO mice. Colon sections were collected on day 4 of the regeneration phase and subjected to H&E (G) and Ki67 (H) staining. Mouse survival was monitored daily (I). Survival curves were analyzed using the log-rank (Mantel-Cox) test. *n* for each group is shown. Scale bars in (G): 500 μ m (upper panels) and 100 μ m (lower panels). Scale bar in (H): 50 μ m.

Minor Points:

1. Molecular weight markers must be included for all blots. Phosphorylation data should be quantified relative to corresponding total protein levels, not grayscale ratios, to ensure accurate interpretation.

Thank you for these suggestions. Molecular weight markers have now been added to all relevant immunoblots in the revised manuscript. Regarding the quantification of phosphorylation, we would like to clarify that our analyses were already performed as recommended. Specifically, for each sample, phospho-protein signal intensity was first normalized to the corresponding total protein level, and these values were then further normalized to the control group to generate the fold-change values shown below each blot.

2. Data in Fig. 1A-1B require validation using at least two independent RNAi lines, consistent with the approach in Fig. 1F.

We validated the findings in Figure 1A-1B (and additionally Figure 1C) using a second independent RNAi line (BDSC #42569). The new results have been incorporated into the revised manuscript as Figures EV1C-E and are also provided below as **Author Response Figure 21** for the reviewer's convenience.

Author Response Figure 21. (C-D) Survival analysis of virgin female (C) and male (D) flies expressing *fray* RNAi construct driven by the *esg-Gal4* driver. Flies were exposed to 5% DSS, and survival was recorded daily. Survival curves were analyzed using the log-rank (Mantel-Cox) test. For (C), $n = 32$ flies for the *esg>GFP* group and $n = 52$ flies for the *esg>GFP;fray.RNAi* group, pooled from three independent experiments. For (D), $n = 37$ flies for the *esg>GFP* group and $n = 63$ flies for the *esg>GFP;fray.RNAi* group, pooled from three independent experiments. (E) Quantification of Smurf-positive flies with the indicated genotypes. Data were analyzed using a two-tailed Student's t-test and are presented as mean \pm s.d.; $n = 3$ independent experiments (10 flies per experiment).

3. Comparing colony numbers in Fig. S2A is invalid due to significant heterogeneity in colony size. Quantitative analysis should instead measure total area covered or incorporate size-normalized metrics.

We appreciate the reviewer's suggestion. In accordance with this comment, we have reanalyzed the clonogenic assay data using colony area rather than colony number and have updated the corresponding figures in the revised manuscript.

4. Phenotypic assessment at "Recovery Day 4" (Fig. 2D-2H) lacks correlation with OXSRI phosphorylation dynamics. A time-course analysis of phospho-OXSRI levels post-injury (complementing Fig. 2M) is essential to justify this endpoint.

Thank you for the comment. We would like to clarify that the experiments in these figures were designed to assess the functional requirement of OXSRI during intestinal regeneration by comparing *Oxsrl* conditional knockout mice (TAM+) with control littermates (TAM-). Because *Oxsrl* is genetically ablated in TAM+ mice, OXSRI protein, and therefore its phosphorylation, is not detectable in these samples, making a

phosphorylation time-course analysis in this context infeasible. Furthermore, as shown in Figure 2C, due to the severity of the regeneration defect, *Oxsr1* conditional knockout mice do not survive beyond Recovery Day 4 (equivalent to Day 9 after DSS treatment: 5 days of DSS followed by 4 days of recovery), which defines the latest viable time point for comparative phenotypic analysis. For reference, the dynamics of OXSR1 phosphorylation in wild-type mice are presented in Figures 2K-2M of the initial submission (now Figures 2H-2J in the revised manuscript).

5. JAS treatment elevates p-YAP levels (indicating inactivation) in WT cells (Fig. 5E), yet upregulates YAP target genes (indicating activation) in Fig. 5F-5G. This contradiction requires resolution.

We thank the reviewer for the thoughtful comment. Previous studies have established a positive relationship between F-actin levels and YAP activity: stabilization of F-actin with JAS reduces p-YAP and increases YAP target gene expression (Reddy, Deguchi et al., 2013), whereas disruption of F-actin with LatB increases p-YAP and drives YAP cytoplasmic localization (Kim, Kim et al., 2013, Zhao, Li et al., 2012). Consistent with these reports, in Figure 5H of our initial submission (now Figure 5I in the revised manuscript), LatB treatment enhanced p-YAP levels. Additionally, in this revision we examined YAP target gene expression following LatB treatment and found it reduced (Figure 5J; also shown below as **Author Response Figure 22**).

By contrast, although JAS treatment increased gene expression in WT cells, we observed a modest increase in YAP phosphorylation, as noted by the reviewer. While the basis for this discrepancy is not fully clear, one plausible explanation is that YAP phosphorylation may fluctuate due to feedback regulations, whereas YAP target gene expression measured by RT-qPCR reflects cumulative transcriptional output over time. Thus, modest or fluctuating changes in p-YAP may not strictly correlate with steady-state target gene expression levels under all conditions.

More importantly, the purpose of the experiments in Figure 5E-5G in the initial submission (now Figures 5G and 5H) was not to examine the absolute effects of JAS on YAP phosphorylation and activity in wild-type cells, but rather to determine whether

OXSRI regulates YAP activity in an F-actin-dependent manner. Thus, the key comparisons are between wild-type and *Oxsr1*-knockout cells under each treatment condition. In these comparisons, the enhanced p-YAP levels and reduced YAP target gene expression in *Oxsr1*-knockout cells (lanes 2 vs. 1 in Figure 5G; bars 2 vs. 1 in Figures 5H) were largely suppressed when F-actin was stabilized by JAS (lanes 4 vs. 3 in Figure 5G; bars 4 vs. 3 in Figures 5H). Conversely, in Figure 5H of the initial submission (now Figure 5I), the reduction in p-YAP caused by OXSRI overexpression (lanes 2 vs. 1) was largely inhibited when F-actin was disrupted by LatB treatment (lanes 4 vs. 3). Consistently, newly added data in this revision (Figure 5J; also shown below as **Author Response Figure 22**) show that the increase in YAP target gene expression caused by OXSRI overexpression is similarly abolished by LatB treatment. Together, we hope the reviewer agrees that, despite the noted discrepancy, the overall interpretation remains unchanged and our core conclusion, OXSRI regulates YAP activity through its influence on F-actin, remains well supported by the data.

Author response Figure 22. HEK293T cells stably expressing empty vector (EV) or OXSRI1 were treated with DMSO or Latrunculin B (LatB, 500 ng/mL, 1.5 h). The expression of the YAP target genes *Ctgf* and *Cyr61* was analyzed by RT-qPCR. Notably, the elevated expression of *Ctgf* and *Cyr61* caused by OXSRI1 overexpression was blocked by LatB treatment. Data were analyzed using two-way ANOVA followed by Sidak's multiple comparisons test and are presented as mean \pm s.d., $n = 3$ independent experiments.

6. The specific basis for selecting *RhoB* and *Rac1* for analysis in Fig. 6 must be explicitly stated (e.g., prior linkage to osmosensing, Hippo regulation, or preliminary screening data).

Thanks for this comment. To investigate the mechanisms by which OXSRI1 promotes F-actin assembly, we searched the BioGRID database for known OXSRI1 interactors with established roles in actin cytoskeletal regulation. Among the identified candidates,

RhoB and Rac1 were the only small GTPases with both well-documented functions in regulating F-actin dynamics and evidence of direct interaction with OXSR1 (Tapon & Hall, 1997). Based on this, we selected RhoB and Rac1 for further analysis. For reference, this rationale was described in Lines 387-392 of our initial submission (now **Lines 471-478** in the revised manuscript).

7. The citation "Fig. S7c" in line 489 should be corrected to "Fig. S6c."

We thank the reviewer for pointing this out. The citation has been corrected from “Fig. S7C” to “Fig. EV5C” in the revised manuscript, in accordance with the journal’s guidelines.

References

AlAmri MA, Kadri H, Alderwick LJ, Simpkins NS, Mehellou Y (2017) Rafoxanide and Closantel Inhibit SPAK and OSR1 Kinases by Binding to a Highly Conserved Allosteric Site on Their C-terminal Domains. *ChemMedChem* 12: 639-645

Algharabil J, Kintner DB, Wang Q, Begum G, Clark PA, Yang SS, Lin SH, Kahle KT, Kuo JS, Sun D (2012) Inhibition of Na(+)-K(+)-2Cl(-) cotransporter isoform 1 accelerates temozolomide-mediated apoptosis in glioblastoma cancer cells. *Cell Physiol Biochem* 30: 33-48

Assi K, Pillai R, Gomez-Munoz A, Owen D, Salh B (2006) The specific JNK inhibitor SP600125 targets tumour necrosis factor-alpha production and epithelial cell apoptosis in acute murine colitis. *Immunology* 118: 112-21

Boyd-Shiwarski CR, Shiwarski DJ, Griffiths SE, Beacham RT, Norrell L, Morrison DE, Wang J, Mann J, Tennant W, Anderson EN, Franks J, Calderon M, Connolly KA, Cheema MU, Weaver CJ, Nkashama LJ, Weckerly CC, Querry KE, Pandey UB, Donnelly CJ et al. (2022) WNK kinases sense molecular crowding and rescue cell volume via phase separation. *Cell* 185: 4488-4506 e20

Cargnello M, Roux PP (2011) Activation and function of the MAPKs and their substrates, the MAPK-activated protein kinases. *Microbiol Mol Biol Rev* 75: 50-83

Chai T, Shen J, Sheng Y, Huang Y, Liang W, Zhang Z, Zhao R, Shang H, Cheng W, Zhang H, Chen X, Huang X, Zhang Y, Liu J, Yang H, Wang L, Pan S, Chen Y, Han L, Qiu Q et al. (2024) Effects of flora deficiency on the structure and function of the large intestine. *iScience* 27: 108941

Dbouk HA, Weil LM, Perera GK, Dellinger MT, Pearson G, Brekken RA, Cobb MH (2014) Actions of the protein kinase WNK1 on endothelial cells are differentially mediated by its substrate kinases OSR1 and SPAK. *Proceedings of the National Academy of Sciences of the United States of America* 111: 15999-6004

Densham RM, O'Neill E, Munro J, Konig I, Anderson K, Kolch W, Olson MF (2009) MST kinases monitor actin cytoskeletal integrity and signal via c-Jun N-terminal kinase stress-activated kinase to regulate p21Waf1/Cip1 stability. *Molecular and cellular biology* 29: 6380-90

Eherer AJ, Fordtran JS (1992) Fecal osmotic gap and pH in experimental diarrhea of various causes. *Gastroenterology* 103: 545-51

Gallolu Kankanamalage S, Karra AS, Cobb MH (2018) WNK pathways in cancer signaling networks. *Cell Commun Signal* 16: 72

He C, Mao D, Hua G, Lv X, Chen X, Angeletti PC, Dong J, Remmenga SW, Rodabaugh KJ, Zhou J, Lambert PF, Yang P, Davis JS, Wang C (2015) The Hippo/YAP pathway interacts with EGFR signaling and HPV oncoproteins to regulate cervical cancer progression. *EMBO molecular medicine* 7: 1426-49

Hollenbach E, Neumann M, Vieth M, Roessner A, Malfertheiner P, Naumann M (2004) Inhibition of p38 MAP kinase- and RICK/NF-kappaB-signaling suppresses inflammatory bowel disease. *FASEB J* 18: 1550-2

Hommel D, van den Blink B, Plasse T, Bartelsman J, Xu C, Macpherson B, Tytgat G, Peppelenbosch M, Van Deventer S (2002) Inhibition of stress-activated MAP kinases induces clinical improvement in moderate to severe Crohn's disease. *Gastroenterology* 122: 7-14

Jiang H, Grenley MO, Bravo MJ, Blumhagen RZ, Edgar BA (2011) EGFR/Ras/MAPK signaling mediates adult midgut epithelial homeostasis and regeneration in *Drosophila*. *Cell Stem Cell* 8: 84-95

Jung JU, Cobb MH (2023) WNK1 controls endosomal trafficking through TRIM27-dependent regulation of actin assembly. *Proceedings of the National Academy of Sciences of the United States of America* 120: e2300310120

Jung JU, Jaykumar AB, Cobb MH (2022) WNK1 in Malignant Behaviors: A Potential Target for Cancer? *Frontiers in cell and developmental biology* 10: 935318

Kim M, Kim M, Lee S, Kuninaka S, Saya H, Lee H, Lee S, Lim DS (2013) cAMP/PKA signalling reinforces the LATS-YAP pathway to fully suppress YAP in response to actin cytoskeletal changes. *The EMBO journal* 32: 1543-55

Meng Z, Moroishi T, Mottier-Pavie V, Plouffe SW, Hansen CG, Hong AW, Park HW, Mo JS, Lu W, Lu S, Flores F, Yu FX, Halder G, Guan KL (2015) MAP4K family kinases act in parallel to MST1/2 to activate LATS1/2 in the Hippo pathway. *Nature communications* 6: 8357

Patterson MR, Cogan JA, Cassidy R, Theobald DA, Wang M, Scarth JA, Anene CA, Whitehouse A, Morgan EL, Macdonald A (2024) The Hippo pathway transcription factors YAP and TAZ play HPV-type dependent roles in cervical cancer. *Nature communications* 15: 5809

Paul S, Hagenbeek TJ, Tremblay J, Kameswaran V, Ong C, Liu C, Guarnaccia AD, Mondo JA, Hsu PL, Kljavin NM, Czech B, Smola J, Nguyen DAH, Lacap JA, Pham TH, Liang Y, Blake RA, Gerosa L, Grimmer M, Xie S et al. (2025) Cooperation between the Hippo and MAPK pathway activation drives acquired resistance to TEAD inhibition. *Nature communications* 16: 1743

Pearson JD, Huang K, Pacal M, McCurdy SR, Lu S, Aubry A, Yu T, Wadosky KM, Zhang L, Wang T, Gregorieff A, Ahmad M, Dimaras H, Langille E, Cole SPC, Monnier PP, Lok BH, Tsao MS, Akeno N, Schramek D et al. (2021) Binary pan-cancer classes with distinct vulnerabilities defined by pro- or anti-cancer YAP/TEAD activity. *Cancer Cell* 39: 1115-1134 e12

Reddy P, Deguchi M, Cheng Y, Hsueh AJ (2013) Actin cytoskeleton regulates Hippo signaling. *PLoS one* 8: e73763

Sanchez-Vega F, Mina M, Armenia J, Chatila WK, Luna A, La KC, Dimitriadoy S, Liu DL, Kantheti HS, Saghafeinia S, Chakravarty D, Daian F, Gao Q, Bailey MH, Liang WW, Foltz SM, Shmulevich I, Ding L, Heins Z, Ochoa A et al. (2018) Oncogenic Signaling Pathways in The Cancer Genome Atlas. *Cell* 173: 321-337 e10

Shiau YF (1987) Clinical and laboratory approaches to evaluate diarrheal disorders. *Crit Rev Clin Lab Sci* 25: 43-69

Shiau YF, Feldman GM, Resnick MA, Coff PM (1985) Stool electrolyte and osmolality measurements in the evaluation of diarrheal disorders. *Ann Intern Med* 102: 773-5

Shiozaki A, Nako Y, Ichikawa D, Konishi H, Komatsu S, Kubota T, Fujiwara H, Okamoto K, Kishimoto M, Marunaka Y, Otsuji E (2014) Role of the Na (+)/K (+)/2Cl(-) cotransporter NKCC1 in cell cycle progression in human esophageal squamous cell carcinoma. *World J Gastroenterol* 20: 6844-59

Tapon N, Hall A (1997) Rho, Rac and Cdc42 GTPases regulate the organization of the actin cytoskeleton. *Current opinion in cell biology* 9: 86-92

Tropini C, Moss EL, Merrill BD, Ng KM, Higginbottom SK, Casavant EP, Gonzalez CG, Fremin B, Bouley DM, Elias JE, Bhatt AS, Huang KC, Sonnenburg JL (2018) Transient Osmotic Perturbation Causes Long-Term Alteration to the Gut Microbiota. *Cell* 173: 1742-1754 e17

Wu N, Sun H, Zhao X, Zhang Y, Tan J, Qi Y, Wang Q, Ng M, Liu Z, He L, Niu X, Chen L, Liu Z, Li HB, Zeng YA, Roulis M, Liu D, Cheng J, Zhou B, Ng LG et al. (2021) MAP3K2-regulated intestinal stromal cells define a distinct stem cell niche. *Nature* 592: 606-610

Xie J, Yoon J, Yang SS, Lin SH, Huang CL (2013) WNK1 protein kinase regulates embryonic cardiovascular development through the OSR1 signaling cascade. *The Journal of biological chemistry* 288: 8566-8574

Yui S, Azzolin L, Maimets M, Pedersen MT, Fordham RP, Hansen SL, Larsen HL, Guiu J, Alves MRP, Rundsten CF, Johansen JV, Li Y, Madsen CD, Nakamura T, Watanabe M, Nielsen OH, Schweiger PJ, Piccolo S, Jensen KB (2018) YAP/TAZ-Dependent Reprogramming of Colonic Epithelium Links ECM Remodeling to Tissue Regeneration. *Cell Stem Cell* 22: 35-49 e7

Zhang J, Ji JY, Yu M, Overholtzer M, Smolen GA, Wang R, Brugge JS, Dyson NJ, Haber DA (2009) YAP-dependent induction of amphiregulin identifies a non-cell-autonomous component of the Hippo pathway. *Nature cell biology* 11: 1444-50

Zhang S, Meor Azlan NF, Josiah SS, Zhou J, Zhou X, Jie L, Zhang Y, Dai C, Liang D, Li P, Li Z, Wang Z, Wang Y, Ding K, Wang Y, Zhang J (2023) The role of SLC12A family of cation-chloride cotransporters and drug discovery methodologies. *Journal of pharmaceutical analysis* 13: 1471-1495

Zhang X, Bandyopadhyay S, Araujo LP, Tong K, Flores J, Laubitz D, Zhao Y, Yap G, Wang J, Zou Q, Ferraris R, Zhang L, Hu W, Bonder EM, Kiela PR, Coffey R, Verzi MP, Ivanov, II, Gao N (2020) Elevating EGFR-MAPK program by a nonconventional Cdc42 enhances intestinal epithelial survival and regeneration. *JCI Insight* 5

Zhao B, Li L, Wang L, Wang CY, Yu J, Guan KL (2012) Cell detachment activates the Hippo pathway via cytoskeleton reorganization to induce anoikis. *Genes & development* 26: 54-68

Zheng Y, Liu B, Wang L, Lei H, Pulgar Prieto KD, Pan D (2017) Homeostatic Control of Hpo/MST Kinase Activity through Autophosphorylation-Dependent Recruitment of the STRIPAK PP2A Phosphatase Complex. *Cell reports* 21: 3612-3623

Dear Bo,

Thank you for submitting the revised version of your manuscript to The EMBO Journal. The study has now been seen by two of the original referees, who appreciate the revisions, but also find that several of their initial points were not sufficiently addressed or clarified. Since the raised concerns appear addressable, I would like to invite you to address the remaining referee comments in the final version.

Additionally, there are some editorial aspects that would need to be addressed in the final revision:

1. Please check that the funding information is correct and identical both in the manuscript and our online system. Currently, Fundamental Research Funds for the Central Universities (2072024) are missing in our online system.
2. CRediT has replaced the traditional author contributions section because it offers a systematic, machine-readable author contributions format that allows for more effective research assessment. Please remove the Author Contributions from the manuscript and use the free text boxes beneath each contributing author's name in our online submission system to add specific details on the author's contribution. More information is available in our guide to authors.
3. Please move Figure Legends and Expanded View Figure Legends after the 'References'.
4. Please update references according to The EMBO Journal style - where there are more than 10 authors on a paper, the first 10 should be listed, followed by 'et al.'
5. In the Data Availability section, please add a resolvable link to the transcriptomics dataset. More information about the format of this section can be found here: <https://link.springer.com/partners/embo-press/editorial-policies#Data%20availability%20statement>.
6. During our standard numerical data check, we noticed a couple of cases of numerical repetitions in the source data files. I appreciate that this could be the case due to the calculations or measurement approaches that were used. I have attached the corresponding files with the detected duplications labelled in colour. Please take a look and correct if needed. A brief explanation would be very helpful.
7. In our standard textual plagiarism check, we noted a paragraph that is almost identical to that from an earlier publication from your team (attached, rows 98-107). Please check and paraphrase.
8. Our data editors have flagged the following issues in figure legends that need correcting:
 - Please note that the exact p values are not provided in the legends of figures 2D, EV2 C, D; EV3 C, G, K; EV4 B, EV5 F, H.
 - Please indicate the statistical test used for data analysis in the legends of figures 3A, B.
 - Please define the error bars in the legends of figures 1B, EV1 C, D.
9. Papers published in The EMBO Journal are accompanied online by a 'Synopsis' to enhance discoverability of the manuscript. It consists of A) a short (1-2 sentences) summary of the findings and their significance, B) 3-4 bullet points highlighting key results and C) a synopsis image that is 550x300-600 pixels large (width x height, jpeg or png format). You can either show a model or key data in the synopsis image. Please note that the image size is rather small and that text needs to be readable at the final size. Please send us this information together with the revised manuscript.

We generally allow three months as standard revision time. Should you foresee a problem in meeting this deadline, please let us know in advance to discuss an extension.

Please feel free to contact me if you have any questions regarding this final revision. Thank you again for giving us the chance to consider your manuscript for The EMBO Journal. I look forward to receiving the revised version.

With best wishes,

Ieva

Ieva Gailite, PhD
Senior Scientific Editor
The EMBO Journal
Meyerohofstrasse 1
D-69117 Heidelberg
Tel: +4962218891309
i.gailite@embojournal.org

We realize that it is difficult to revise to a specific deadline. In the interest of protecting the conceptual advance provided by the work, we recommend a revision within 3 months (21st Apr 2026). Please discuss the revision progress ahead of this time with the editor if you require more time to complete the revisions.

Referee #1:

The authors have effectively addressed most of my concerns. For responses to major concerns 1-3, they might consider additional revisions before publication. Some experiments, like regional osmosis (#1), are technically complex; it would be helpful to mention this or include it as a limitation at the end of the main text. Regarding stem cell markers (#2), Lrig1 may not be appropriate since it marks cells different from Lgr5+ stem cells; alternatives such as Olfm4 staining or single-cell transcriptome analysis could be used. The discrepancies with previous literature (#3) should be explicitly listed as discussed.

Referee #3:

Author justify focusing on WNK-OXSR1 due to its specificity as an osmosensor, unlike the "broad stress response" of p38/MAPK. However, intestinal injury is a broad stressor involving inflammation, cytokines, and bacterial products. Why is a pathway selective for pure osmotic change more physiologically relevant for in vivo regeneration than a pathway like MAPK that integrates multiple injury-related signals?

MAPK is the most prominently altered pathway in your OXSR1-KO screen but chose Hippo due to a "knowledge gap." This is a valid starting point but creates a circularity: they find how osmolarity activates YAP, so they selectively pursue the Hippo data from your screen. Can they provide positive evidence that the Hippo pathway alteration is more critical than the MAPK alteration for the *OXSR1-dependent* phenotype? The new p-ERK data (Fig. S3D) shows it's still induced in cKO, suggesting MAPK activation may be OXSR1-independent. A key experiment would be testing if concurrent inhibition of MAPK and OXSR1 produces an additive defect, which would undermine the claim that Hippo is the primary effector.

The T185E/T185A rescue experiment is excellent and directly supports relevant model. However, this validation occurs at the very end of the pathway. Do these phospho-mutants also recapitulate the proposed mechanistic steps? Specifically, does expressing OXSR1-T185E, but not T185A, in OXSR1-KO cells restore, RhoB-T37 phosphorylation, RhoB-GTP loading, and F-actin polymerization? Without this link, the mutant data only confirms OXSR1 kinase activity is important, not that it functions specifically through the RhoB/actin/Hippo axis you delineate.

The addition of SW480 and primary crypt data strengthens the case. However, were the key mechanistic experiments in SW480 cells and primary crypts (RhoB-GTP, F-actin) performed under conditions of osmotic stress or injury mimicry? The rebuttal states findings are "consistent" but does not specify the stimulus. To claim relevance to an osmosensing pathway, it is crucial to demonstrate that the OXSR1-RhoB-actin cascade is activated by osmotic change in these intestinal models.

Referee #1:

The authors have effectively addressed most of my concerns.

We are glad that most of the concerns have been effectively addressed. Below we address the remaining points.

For responses to major concerns 1-3, they might consider additional revisions before publication. Some experiments, like regional osmosis (#1), are technically complex; it would be helpful to mention this or include it as a limitation at the end of the main text.

Thanks for the suggestion. We have mentioned this in the revised Discussion section (**Lines 681-684**) to clarify the limitations of our current experimental system.

*Regarding stem cell markers (#2), *Lrig1* may not be appropriate since it marks cells different from *Lgr5*⁺ stem cells; alternatives such as *Olfm4* staining or single-cell transcriptome analysis could be used.*

Thanks for the suggestion. We would like to clarify that we did attempt immunostaining for *Olfm4* in the previous revision before using *Lrig1* as a stem cell marker in the mouse colon. However, we were unable to detect any *Olfm4* signal in mouse colonic samples, which are the primary tissues analyzed in this study. This result is consistent with previous reports showing that *Olfm4* is not expressed in the mouse colon, although it is expressed in the mouse small intestine and in the human colon (Schuijers *et al*, 2014; Shi *et al*, 2024; van der Flier *et al*, 2009).

To further address the reviewer's concern, we alternatively assessed intestinal stem cell dynamics by isolating colonic crypts from wild-type mice under homeostatic conditions as well as at multiple time points during the regeneration phase. We then analyzed the expression of the canonical intestinal stem cell marker *Lgr5* via RT-qPCR. These experiments revealed a marked reduction in *Lgr5* levels during the injury phase, followed by a gradual restoration during regeneration, which temporally correlates with *OXSRI* activation. This data has now been incorporated into the revised manuscript as **Figure EV2F** (also show below as **Author Response Figure 1** for convenience).

Author Response Figure 1. Colonic crypts were collected at the indicated stages of injury and regeneration and *Lgr5* mRNA levels were assessed by RT-qPCR. Note that *Lgr5* mRNA levels were reduced during injury and progressively reappeared during regeneration. Data were analyzed using one-way ANOVA followed by Dunnett's multiple comparisons test and are presented as mean \pm s.d. ($n = 3$ colons).

The discrepancies with previous literature (#3) should be explicitly listed as discussed.

We have discussed this in the revised Discussion section (**Lines 772-790**).

Referee #3:

Author justify focusing on WNK-OXSRI due to its specificity as an osmosensor, unlike the "broad stress response" of p38/MAPK. However, intestinal injury is a broad stressor involving inflammation, cytokines, and bacterial products. Why is a pathway selective for pure osmotic change more physiologically relevant for in vivo regeneration than a pathway like MAPK that integrates multiple injury-related signals?

We thank the reviewer for this follow-up question. We fully agree that intestinal injury in vivo represents a complex stress involving multiple factors, such as inflammation, cytokine signaling, microbial cues, and tissue damage. Indeed, pathways such as MAPK/p38 play essential roles in integrating these diverse injury-related signals during regeneration.

Our rationale for prioritizing the WNK-OXSRI pathway is not that osmotic stress occurs in isolation, but rather that osmotic changes constitute an early, fundamental, and unavoidable biophysical consequence of epithelial barrier disruption, occurring concurrently with, and potentially upstream of, many inflammatory and stress responses. Barrier damage leads to rapid alterations in luminal and interstitial osmolarity, directly impacting epithelial cell volume and intracellular molecular crowding. The WNK-OXSRI axis is uniquely positioned to sense these immediate biophysical changes and convert them into intracellular signaling responses.

While MAPK/p38 signaling is undoubtedly physiologically relevant, it integrates a broad spectrum of stressors, including cytokines, reactive oxygen species, mechanical stress, and microbial pattern molecules, rather than osmotic cues alone. This pleiotropy obscures

the specific contribution of osmolarity within the complex in vivo injury environment. In contrast, the specificity of WNK-OXSRI pathway allows for the isolation of osmotic stress, enabling us to interrogate it as a distinct regenerative input.

Importantly, our focus on WNK-OXSRI does not imply that it operates independently of other pathways. Rather, we view osmotic sensing via WNK-OXSRI as a complementary, and potentially upstream, mechanism that may shape or modulate downstream stress-integrating pathways, including MAPK signaling, during regeneration. In this context, studying a pathway selective for osmotic changes provides mechanistic clarity and enables us to define how a specific biophysical signal contributes to the overall regenerative program.

In summary, WNK-OXSRI was selected for its unique capacity to largely decouple osmotic changes, an inherent but underexplored feature of intestinal injury, from the multifaceted in vivo regenerative environment.

*MAPK is the most prominently altered pathway in your OXSRI-KO screen but chose Hippo due to a "knowledge gap." This is a valid starting point but creates a circularity: they find how osmolarity activates YAP, so they selectively pursue the Hippo data from your screen. Can they provide positive evidence that the Hippo pathway alteration is more critical than the MAPK alteration for the *OXSRI-dependent* phenotype? The new p-ERK data (Fig. S3D) shows it's still induced in cKO, suggesting MAPK activation may be OXSRI-independent. A key experiment would be testing if concurrent inhibition of MAPK and OXSRI produces an additive defect, which would undermine the claim that Hippo is the primary effector.*

Thanks for the comments. We do not claim that Hippo signaling is more important than MAPK signaling in intestinal regeneration, nor that it is the sole downstream effector of OXSRI. Rather, our intention was to identify and mechanistically characterize one direct and physiologically relevant OXSRI-dependent axis, rather than to establish a strict hierarchy among multiple injury-responsive signaling cascades that govern regeneration.

Hippo-YAP signaling represents a pathway with a well-established role in intestinal regeneration, yet its upstream activation mechanisms in response to injury remain poorly defined. Our study addresses this gap by providing direct evidence that OXSRI modulates YAP activity, and positioning Hippo-YAP axis as a functionally relevant downstream effector of OXSRI.

Regarding the reviewer's concern that MAPK activation may be OXSRI-independent based on p-ERK induction, we note that quantification of p-ERK relative to total ERK levels in Fig. EV3J (also shown below as **Author Response Figure 2**) demonstrates that regeneration-associated ERK activation is largely abolished in *Oxsr1*-cKO mice. This suggests that, similar to JNK and p38, ERK activation during regeneration is predominantly OXSRI-dependent.

We agree that testing concurrent inhibition of MAPK signaling and OXSR1 could, in principle, provide insight into pathway interactions. However, given the dependency of MAPK activation on OXSR1, we consider it unlikely that such an experiment would reveal an additive defect. Moreover, MAPK signaling exerts biphasic, context-dependent effects on intestinal repair. Depending on the timing and context, MAPK signaling can play positive (Jiang *et al*, 2011; Zhang *et al*, 2020), negative (Assi *et al*, 2006; Chen *et al*, 2022; Hollenbach *et al*, 2004; Hommes *et al*, 2002) or minimal (Wu *et al*, 2021) roles in intestinal epithelial repair. These pleiotropic effects would complicate interpretation of combined inhibition experiments and make it difficult to distinguish additive, epistatic, or compensatory effects *in vivo*. We therefore believe that a detailed dissection of MAPK signaling in OXSR1- (or osmotic stress-) mediated regeneration is beyond the scope of the current study and better suited for future work.

In summary, our study does not posit Hippo signaling as more important than MAPK signaling, but rather identifies Hippo-YAP as a direct and mechanistically tractable OXSR1-dependent pathway that links osmotic stress sensing to epithelial regeneration. We have clarified this point explicitly in the Discussion section (**Lines 691-696**).

Author Response Figure 2. Colonic crypts from *Oxsr1* cKO (TAM+) and control (TAM-) mice were collected at regeneration day 4 and analyzed by western blot. p-ERK levels relative to total ERK were quantified using ImageJ. For each genotype, values from regenerative samples were normalized to the corresponding homeostasis samples to calculate fold induction. Note the increased phosphorylation of all three MAPKs in control crypts but not in *Oxsr1* cKO crypts.

The T185E/T185A rescue experiment is excellent and directly supports relevant model. However, this validation occurs at the very end of the pathway. Do these phospho-mutants also recapitulate the proposed mechanistic steps? Specifically, does expressing OXSR1-T185E, but not T185A, in OXSR1-KO cells restore, RhoB-T37 phosphorylation, RhoB-GTP loading, and F-actin polymerization? Without this link, the mutant data only confirms OXSR1 kinase activity is important, not that it functions specifically through the RhoB/actin/Hippo axis you delineate.

Thanks for the comments. As suggested, we reintroduced OXSR1-T185E or OXSR1-T185A into *Oxsr1*-null cells and assessed RhoB-GTP and F-actin levels. The results showed that reintroducing OXSR1-T185E mutant, but not OXSR1-T185A mutant, rescued the diminished RhoB-GTP and F-actin levels observed in *Oxsr1*-null cells, indicating that OXSR1 kinase activity is required for these upstream signaling events. The F-actin and RhoB-GTP data have been added as **Fig. 5G** and **Fig. 6F**, respectively, in the revised manuscript. They are also shown below as **Author Response Figure 3** for convenience.

Regarding RhoB phosphorylation, we previously demonstrated that OXSR1-mediated phosphorylation of RhoB depends on its kinase activity using a kinase-dead mutant (OXSR1-K46R), which failed to induce RhoB phosphorylation (Fig. 7B).

Author Response Figure 3. (G) Western blot was performed to examine the levels of detergent-insoluble F-actin and total actin in HEK293T cells of the indicated genotypes. Note that the diminished F-actin levels in *Oxsr1*-null HEK293T cells were rescued by reconstitution with OXSR1-WT or OXSR1-T185E, but not by OXSR1-T185A. (F) RhoB-GTP levels were assessed in cells with the indicated genotypes. Note that reintroducing OXSR1-T185E mutant, but not OXSR1-T185A mutant, rescued the diminished RhoB-GTP levels in *Oxsr1*-null HEK293T cells. S.E., short-exposure; L.E., long-exposure.

The addition of SW480 and primary crypt data strengthens the case. However, were the key mechanistic experiments in SW480 cells and primary crypts (RhoB-GTP, F-actin) performed under conditions of osmotic stress or injury mimicry? The rebuttal states findings are "consistent" but does not specify the stimulus. To claim relevance to an osmosensing pathway, it is crucial to demonstrate that the OXSR1-RhoB-actin cascade is activated by osmotic change in these intestinal models.

Thanks for the suggestion. The experiments in primary crypts (Figs. 5D and 7L) were performed under injury conditions (regen. day 4), as explicitly stated in the corresponding figure legends.

In the previous revision, the experiments in SW480 cells were performed under isotonic conditions. As suggested by the reviewer, we have now repeated these experiments under osmotic stress conditions. The results showed that sorbitol treatment induced a marked increase in both RhoB-GTP and F-actin levels in wild-type SW480 intestinal epithelial cells, but these effects were abolished in *Oxsr1*-null SW480 cells, highlighting the crucial role of the OXSR1-RhoB-actin cascade in sensing osmolarity disturbances. These new data have been added as **Figs. EV5J and EV5K** in the revised manuscript and are also shown below as **Author Response Figure 4** for convenience.

Author Response Figure 4. RhoB-GTP levels (J) and F-actin levels (K) were assessed in wild-type or *Oxsr1* knockout SW480 cells with or without sorbitol treatment (0.1 M, 1 h). Note that sorbitol treatment elevated RhoB-GTP and F-actin levels in wild-type cells, but this response was abolished in *Oxsr1*-null cells.

References

Assi K, Pillai R, Gomez-Munoz A, Owen D, Salh B (2006) The specific JNK inhibitor SP600125 targets tumour necrosis factor- α production and epithelial cell apoptosis in acute murine colitis. *Immunology* 118: 112-121

Chen W, Liang R, Yi Y, Zhu J, Zhang J (2022) P38 α deficiency in macrophages ameliorates murine experimental colitis by regulating inflammation and immune process. *Pathol Res Pract* 233: 153881

Hollenbach E, Neumann M, Vieth M, Roessner A, Malfertheiner P, Naumann M (2004) Inhibition of p38 MAP kinase- and RICK/NF- κ B-signaling suppresses inflammatory bowel disease. *FASEB J* 18: 1550-1552

Hommes D, van den Blink B, Plasse T, Bartelsman J, Xu C, Macpherson B, Tytgat G, Peppelenbosch M, Van Deventer S (2002) Inhibition of stress-activated MAP kinases induces clinical improvement in moderate to severe Crohn's disease. *Gastroenterology* 122: 7-14

Jiang H, Grenley MO, Bravo MJ, Blumhagen RZ, Edgar BA (2011) EGFR/Ras/MAPK signaling mediates adult midgut epithelial homeostasis and regeneration in *Drosophila*. *Cell Stem Cell* 8: 84-95

Schuijers J, van der Flier LG, van Es J, Clevers H (2014) Robust cre-mediated recombination in small intestinal stem cells utilizing the olfm4 locus. *Stem Cell Reports* 3: 234-241

Shi G, Li Y, Shen H, He Q, Zhu P (2024) Intestinal stem cells in intestinal homeostasis and colorectal tumorigenesis. *Life Med* 3: lnae042

van der Flier LG, Haegebarth A, Stange DE, van de Wetering M, Clevers H (2009) OLFM4 is a robust marker for stem cells in human intestine and marks a subset of colorectal cancer cells. *Gastroenterology* 137: 15-17

Wu N, Sun H, Zhao X, Zhang Y, Tan J, Qi Y, Wang Q, Ng M, Liu Z, He L *et al* (2021) MAP3K2-regulated intestinal stromal cells define a distinct stem cell niche. *Nature* 592: 606-610

Zhang X, Bandyopadhyay S, Araujo LP, Tong K, Flores J, Laubitz D, Zhao Y, Yap G, Wang J, Zou Q *et al* (2020) Elevating EGFR-MAPK program by a nonconventional Cdc42 enhances intestinal epithelial survival and regeneration. *JCI Insight* 5

Dear Bo,

Thank you for addressing the remaining requests in the revised manuscript. I am now pleased to inform you that your manuscript has been accepted for publication. Congratulations with a nice study!

Before we forward your manuscript to our publishers, we would like to propose some edits in the manuscript abstract and synopsis. I have also written a short blurb that will accompany the title of your manuscript in our online system. Please take a look at the proposed text changes in the attached text file and let me know if any corrections are needed.

Please note that it is The EMBO Journal policy for the transcript of the editorial process (containing referee reports and your response letters) to be published as an online supplement to each paper. If you should prefer removal of any referee-only figures included in the point-by-point response(s), e.g. because they may still be used for future publication or because they have been reproduced from published work by others, please do let us know immediately via response email.

More information is available here: <https://link.springer.com/partners/embo-press/editorial-policies#Peer%20review>

You may qualify for financial assistance for your publication charges - either via a Springer Nature fully open access agreement or an EMBO initiative. Check your eligibility: <https://link.springer.com/journal/44318/how-to-publish-with-us>

If you have any questions, please do not hesitate to contact the Editorial Office. Thank you for this contribution to The EMBO Journal!

Wishing you a happy Year of the Horse,

leva

leva Gailite, PhD
Senior Scientific Editor
The EMBO Journal
Meyerhofstrasse 1
D-69117 Heidelberg
Tel: +4962218891309
i.gailite@embojournal.org